# Evolution of crystallographic preferred orientations of ice sheared to high strains by equal-channel angular pressing

Qinyu Wang[1,2,3], Sheng Fan[3], Daniel H. Richards[4], Rachel Worthington[3], David J. Prior[3], and Chao Qi[1,2]

[1]Center for High Pressure Science and Technology Advanced Research, Beijing, 100193, China
[2]Key Laboratory of Earth and Planetary Physics, Institute of Geology and Geophysics, Chinese Academy of Sciences, Beijing, 100029, China
[3]Department of Geology, University of Otago, Dunedin, 9016, New Zealand
[4]Institute for Marine and Antarctic Studies, University of Tasmania, Hobart, 7005, Australia

**Correspondence:** Chao Qi (chao.qi@hpstar.ac.cn)

**Abstract.** Plastic deformation of polycrystalline ice Ih induces crystallographic preferred orientations (CPOs), which give rise to anisotropy in the viscosity of ice, thereby exerting a strong influence on the flow of glaciers and ice sheets. The development of CPOs is governed by the relative importance of two pivotal recrystallization mechanisms: subgrain/lattice rotation and strain-induced grain boundary migration (GBM). To examine the impact of strain on the relative importance of these two mechanisms, synthetic ice (doped with ∼1 vol.% graphite) was deformed using the equal-channel angular pressing technique, enabling multiple passes to accumulate substantial shear strains. Nominal shear strains up to 6.2, equivalent to a nominal von Mises strain of $\varepsilon' \approx 3.6$, were achieved in samples at a temperature of $-5°C$. Cryo-electron backscatter diffraction analysis reveals a primary cluster of crystal $c$ axes perpendicular to the shear plane in all samples, accompanied by a secondary cluster of $c$ axes at an oblique angle to the primary cluster antithetic to the shear direction. With increasing strain, the primary $c$-axis cluster strengthens, while the secondary cluster weakens. The angle between the clusters remains within the range of $45°$ to $60°$. The $c$-axis clusters are elongated perpendicular to the shear direction, with this elongation intensifying as strain increases. Subsequent annealing of the highest-strain sample reveals the same CPO patterns as observed prior to annealing, albeit slightly weaker. A synthesis of various experimental data suggests that the CPO pattern, including the orientation of the secondary cluster, results from a balance of two competing mechanisms: lattice rotation due to dislocation slip, which fortifies the primary cluster while rotating and weakening the secondary one, and grain growth by strain-induced GBM, which reinforces both clusters while rotating the secondary cluster in the opposite direction. As strain increases, GBM contributes progressively less. This investigation supports the previous hypothesis that a single cluster of $c$ axes could be generated in high-strain experiments, while further refining our comprehension of CPO development in ice.

## 1 Introduction

Ice Ih constitutes glaciers and ice sheets on Earth, polar ice caps on Mars, icy shells on icy satellites, and a major part of many dwarf planets and asteroids in the solar system. The rheological behavior of ice Ih is critical to the ice-sheet dynamics on Earth (e.g., Gow and Williamson, 1976; Schoof and Hewitt, 2013; Bons et al., 2016), viscous relaxation of topography on

Mars and Ceres (e.g., Pathare et al., 2005; Bland et al., 2016), and convection and thermal evolution in the icy crust on Europa (e.g., Showman and Han, 2005; Barr and McKinnon, 2007). During plastic deformation of ice, crystallographic preferred orientations (CPOs) are induced. These CPOs translate the kinematics into anisotropy in the microstructure, as ice is a highly anisotropic material. When deformed by dislocation glide, a single crystal of ice is several orders of magnitude weaker for slip on the basal plane, (0001), than on others (Duval et al., 1983). Once a CPO is formed in polycrystalline ice, it can lead to elastic and viscous anisotropies, making shear parallel to the aligned basal planes easier and deformation in other orientations more difficult (Duval et al., 1983; Azuma, 1995). When the stress field that drives ice flow changes, the rate of change of the flow rate depends on the existing CPO and its subsequent evolution under the new stress configuration (Hudleston, 2015; Gerber et al., 2023). Thus, the mechanical response of terrestrial and planetary ice bodies to climate, tidal, gravitational and/or geological forcing depends, in part, on the evolution of their CPOs (e.g., Alley, 1992; Duval et al., 2010; Hudleston, 2015). Moreover, CPOs observed in natural ice samples (Jackson and Kamb, 1997; Jackson, 1999; Faria et al., 2014; Thomas et al., 2021; Monz et al., 2021) and derived from seismic data (e.g., Lutz et al., 2020, 2022) or radio-echo sounding (e.g., Jordan et al., 2019; Ershadi et al., 2022; Zeising et al., 2023) can provide valuable insights into the conditions and history of ice deformation. This is similar to how CPOs for quartz (Schmid and Casey, 1986; Law, 2014) and olivine (Karato et al., 2008) are used to understand deformation in the Earth's crust and mantle, respectively.

Many experimental studies focusing on the evolution of CPO in ice have found a transition of the crystallographic fabric with changing stress, temperature and strain (e.g., Qi et al., 2017, 2019; Fan et al., 2020, 2021a). The CPO of ice is typically characterized by the alignment of the $\langle 0001 \rangle$ axes ($c$ axes, which are normal to the basal planes). The orientation of other crystal axes, such as the $\langle 11\bar{2}0 \rangle$ axes ($a$ axes) and the poles to $\{10\bar{1}0\}$ planes (poles to $m$ planes), could also serve as important kinematic indicators (Schmid and Casey, 1986). This transition is attributed to a change in the dominant mechanism for CPO formation from grain boundary migration (GBM) to lattice rotation and subgrain rotation (polygonization) (Alley, 1992; Qi et al., 2017; Fan et al., 2020). The two mechanisms are also referred to as migration recrystallization and rotation recrystallization (Poirier, 1985, pp. 179–185). GBM is usually driven by the difference in dislocation density of grains on both sides of the grain boundary, causing the grain boundary to migrate from the low-density side to the high-density side, thereby consuming grains with high density of dislocations (Urai et al., 1986). The dislocation density accumulated within a grain is predominantly influenced by its orientation relative to the applied deviatoric stress. Consequently, as grains with lower dislocation densities grow at the expense of those with higher densities, there is a corresponding increase in the proportion of $c$ axes oriented along a specific direction. For instance, in compression tests, grains with their basal planes oriented at 45° to the compression axis are inferred to have lower dislocation densities. When GBM dominates CPO formation, $c$ axes are observed to present a conical distribution at 45° to the compression axis (Jacka and Maccagnan, 1984; Jacka and Li, 2000; Qi et al., 2017). Lattice rotation gradually rotates the orientations of slip planes in a grain, to accommodate the bulk deformation. Subgrain rotation produces new grains with orientations similar to those of their parent grains, but with a deviation (Guillope and Poirier, 1979; Poirier, 1985; Urai et al., 1986), thereby leading to a diffused concentration in the $c$-axis distribution (Alley, 1992). For instance, in uniaxial compression, when rotational recrystallization is the dominant mechanism, the $c$ axes tend to form a single cluster parallel to the applied compression axis (Qi et al., 2017). Since high-strain deformation in ice sheets and

glaciers is dominantly simple shear (Cuffey and Paterson, 2010), understanding the CPO evolution under shear is critical. The highest shear strain achieved in a laboratory setting, $\gamma = 2.6$, was reported by Qi et al. (2019) at $-30°C$. Previous experiments at similar temperatures did not exceed shear strains of 0.12 (Wilson and Peternell, 2012). Natural ice samples from shear zones have strong single (Thomas et al., 2021) or double $c$-axis clusters (Jackson and Kamb, 1997; Jackson, 1999; Thomas et al., 2021) with the primary cluster perpendicular to the shear plane. Where shear strain is quantified, the double clusters switch to a single cluster at $\gamma > 5$ (Hudleston, 1977). Shear experiments on polycrystalline ice have found that the fabric transition from double-clustered to single-clustered $c$ axes occurs with increasing strain (Kamb, 1972; Bouchez and Duval, 1982; Li et al., 2000; Wilson and Peternell, 2012; Qi et al., 2019). However, even at shear strains of $\gamma > 2$, the $c$-axis fabric has not evolved to an absolute single cluster. Instead the $c$ axes form a diffused cluster at $\gamma = 2.6$ at $-30°C$, and a very weak secondary cluster at $\gamma = 2.2$ at $-20°C$ (Qi et al., 2019). Meanwhile, numerical simulations incorporating dynamic recrystallization processes have found a single, primary cluster occurring at smaller shear strains than experiments (Llorens et al., 2017; Piazolo et al., 2019; Richards et al., 2021). This discrepancy leads to an uncertainty as to whether the CPO in natural ice can be explained by the formation mechanisms proposed from experimental observations. Thus, new laboratory experiments to higher strains, closer to those found in naturally deformed ice, are needed.

In this contribution, we adapt the equal-channel angular pressing (ECAP) method to polycrystalline ice. ECAP, also known as equal-channel angular extrusion, is a technique for generating severe plastic deformation, resulting in highly strained microstructures with ultra-fine grain sizes and strong fabric through repeated deformation of the sample. ECAP was initially developed in the 1980s by V.M. Segal and colleagues (Segal et al., 1981). ECAP has been extensively used to investigate the microstructural evolution during severe plastic deformation of metals and alloys (e.g., Iwahashi et al., 1998; Zhao et al., 2004; Kawasaki et al., 2009), and thus can be an ideal tool to study the CPO evolution of ice sheared to high strains (Iliescu and Baker, 2008). By conducting experiments at a warm temperature of $-5°C$, inhibiting grain growth with a small fraction of graphite particles (e.g., Song et al., 2005; Azuma et al., 2012; Saruya et al., 2019) and confining the sample with a small back pressure, we are able to press the samples through ECAP up to 6 passes, allowing us to shear ice to nominal shear strains of $\sim6.2$. The objective of this paper is to explore the effects of shear strain on the CPOs and microstructures of ice, especially when ice is deformed to shear strains higher than any previous experiments. The results could help our understanding the physical processes that control the development of CPO in highly deformed natural ice.

## 2 Methods

### 2.1 Sample preparation

To prepare polycrystalline ice samples with a controlled initial microstructure, we adopted the flooding and freezing procedure developed by Durham et al. (1983) and Stern et al. (1997). The sample preparation procedure is the same as in Qi et al. (2019), except that graphite powders with an average particle size of 2.6 μm were added to the seed ice powders to inhibit ice grain growth during the deformation experiments. Samples fabricated this way have a graphite fraction of 1.8–3.6 wt.% (corresponding to 0.8–1.5 vol.%, for more details please refer to Table 1). Previous experimental investigations have found that

such small fractions of particles have no measurable effect on ice rheology, but result in smaller grain sizes than in particle-free ice (e.g., Jacka et al., 2003; Qi et al., 2018; Saruya et al., 2019).

## 2.2 ECAP die

The core of the ECAP die is two channels with equal cross-sectional areas that intersect at a specific angle, as shown schematically in Figure 1. The die consists of two symmetrical stainless-steel parts, each as half of the channel. Seven bolts alongside the channel fasten the two halves together, making a round and curved channel in the die. The diameters of both the channel and ice samples are 25 mm. The channels are mirror-finished, and are coated with a layer of solid soap before each experiment to minimize friction during the experimental runs. The channel geometry is defined by two angles: the channel angle $\Phi$ represents the angle at which the two channels intersect and is a critical factor influencing the shear strain, and the curvature angle $\Psi$ defines the angle at the outer curvature arc where the two segments intersect (see Fig. 1(a)). This angle is typically used to accommodate the friction boundary conditions and eliminate the "dead zone" of no deformation during the process. The sample is pressed through the die using a plunger under an applied pressure. The sample emerges from the die as shown on the right in Figure 1(c). Ideally, for steady and frictionless ECAP, the sample is sheared along the slip-line (line AO in Fig. 1(a)) in uniform, simple shear geometry. When a sample has fully passed through a corner, we refer to this as completing one pass of deformation, termed as "1 pass". Notably, the cross-sectional dimensions of the sample remain unchanged after passing through the channel as shown on the right in Figure 1(c). This allows the sample to be pressed repeatedly through the channel, achieving a high shear strain. After the sample has fully passed through the corner n times, it is termed as "$N$ passes". The strain imposed in each pass of ECAP is dependent upon the angles $\Phi$ and $\Psi$. It can be shown from first principles that the nominal equivalent strain after $N$ passes, $\varepsilon'_N$, is given by a relationship of the form (Iwahashi et al., 1996)

$$\varepsilon'_N = \frac{N}{\sqrt{3}} \left( 2 \cot \left( \frac{\Phi}{2} + \frac{\Psi}{2} \right) + \Psi \operatorname{cosec} \left( \frac{\Phi}{2} + \frac{\Psi}{2} \right) \right). \tag{1}$$

For our design, $\Phi = 120°$ and $\Psi = 60°$, resulting in a nominal equivalent strain of $\varepsilon'_1 \approx 0.6$ (a nominal shear strain of $\gamma'_1 = \sqrt{3}\varepsilon'_1 \approx 1.0$) in the sample per pass. We note that the ECAP samples are deformed differently from traditional experiments in which the whole sample deformed simultaneously (e.g., Kamb, 1972; Qi et al., 2019). Thus, we refer the calculated strain as the nominal strain, denoted by a prime. $\varepsilon'$ is the nominal equivalent strain and $\gamma'$ is the nominal shear strain.

## 2.3 Deformation experiments

The frame of the deformation apparatus is built with aluminum profiles (see Figs. 1(b) and (d)). The ECAP die sits on a wide aluminum profile within a cooler box insulated with wool. Four thermoelectric coolers (TEC) were attached to the flat surface of the ECAP die, with the cooling side attached to the die and the heating side attached to a water box. Both contact surfaces of coolers were covered with high-performance thermally conductive grease for efficient heat conduction. The water box acted as a heat sink and was linked to a water chiller, circulating 5°C water. Four thermisters (PT100) on the die monitored temperatures at locations illustrated in Figures 1(c) and (d). A controller regulated the power of the TEC via proportional-integral-derivative method based on the target temperature, which is set to −5°C for all experiments.

After cooling the die, a sample (at $-30°$C) was inserted in the channel, and a heat-insulating polymer plunger was inserted on top. An aluminum profile rested on the plunger and can slide vertically along two fixed aluminum pillars attached to the bottom profile. Once the sample temperature had equilibrated with the die (typically after 20 min), a load was applied to the sample using dead weights hung on the aluminum profile. To avoid sudden stress increases that could possibly fracture the sample, the load was increased gradually, starting from 2.5 kg (weight of the plunger and profile) and reaching 22.5 kg in increments of 5 kg every 15 minutes. Further, the load was increased from 22.5 kg to a maximum of 42.5 kg in 5 kg increments every 10 minutes. Typically, under the same load, the sample passes through the ECAP faster during the $(i+1)$-th pass compared to the $i$-th pass. To prevent cracking due to excessive speed, the maximum applied load is slightly reduced as the number of passes increases (a reduction of 5 to 10 kg, refer to Table 1 and Supplementary Fig. S1 for details).

To provide a small back pressure to the sample, thereby preventing it from fracturing within the channel and ensuring more uniform strain distribution, two double O-ring stainless-steel plugs were inserted into the channel from the outlet. The displacement of the sample was measured by a linear variable differential transformer (LVDT) as illustrated in Figure 1. Temperature and displacement data were recorded every 10 seconds throughout each experiment. When the sample tail reaches the corner of the channel, the plunger can't push it further. So, we insert a spare piece of ice (which won't be analyzed) to push the sample through the corner, completing a 1-pass experiment.

Upon passing through the channel, the sample's diameter remains unchanged, but the head and tail surfaces are no longer perpendicular to the cylindrical axis (see Figs. 2 and 3). In preparation for additional passes, the head and tail surfaces were restored to the sample's cylindrical shape with a reduced length compared to the original sample (Fig. 2). Because the ECAP die is made of two halves, as a sample passes through, the channel leaves a trace on the outer surface of the sample, corresponding to the divide of the two halves. Using the trace as a marker, the sample can be reinserted into the channel with the same orientation relative to the corner and deformed again. For samples becoming shorter than 50 mm, an additional piece of ice was added to achieve the required length, but this extra ice was not considered in the analysis (Fig. 2). This process allows for multiple passes through the channel, accumulating high strains. After reaching the target number of passes, the samples were wrapped in aluminum foil and stored in liquid nitrogen. Note that during an experiment, the head part of the sample is exposed to air, while the tail part is deformed in the corner, causing more sublimation in the head part. To avoid sublimation bias, we used the middle part near the tail for microstructural analysis.

To investigate the effect of annealing on pre-existing CPO, one sample (ECAP_38_6) was annealed at $-3.5°$C for 24 days after the ECAP deformation experiment. The annealing experiment was done using a similar apparatus described in Fan et al. (2023).

## 2.4 Analysis of microstructure

Samples were prepared for cryogenic electron backscatter diffraction (EBSD) analysis in a scanning-electron microscope (SEM) at the University of Otago, following the procedure described in Prior et al. (2015). During preparation, a sample was either kept in a cryogenic dewar (at $\lesssim -190°$C) or in an insulated transfer box (at $\lesssim -120°$C). At these low temperatures, the defect activity rate within the crystal is extremely low, minimizing changes to the sample's microstructure. The sample was

first cut in half along the profile plane, which is parallel to the cylinder axis and perpendicular to the shear plane, and then cut again parallel to the initial cut to obtain a 5-mm-thick section of the profile plane (see Fig. 3(c)). This process was conducted using a band saw in a cold room maintained at $-10°C$. The cutting was done within 15 minutes, and the thermal effects were deemed acceptable (Prior et al., 2015; Fan et al., 2021b). Sample mounting to copper ingots to go into the SEM was done the same way as described in Prior et al. (2015). A small piece of the sample was also cut, melted and weighed to examine the exact graphite fraction.

To obtain a flat surface for EBSD analysis, the section surface was polished on sandpapers atop a flat metal plate at $-80$ to $-50°C$, with a grit size of 400. The sample, mounted on the ingot, was transferred into the SEM and mounted on a cryostage, maintained $< -90°C$, through a nitrogen-filled glove box (Prior et al., 2015). A pressure cycle was performed in the SEM chamber to remove frost on the sample surface (Prior et al., 2015).

Orientation and element data were collected sequentially on a Zeiss Sigma VP field-emission-gun SEM equipped with a Symmetry EBSD camera and a X-max energy dispersive spectroscopy (EDS) detector from Oxford Instruments, respectively. 2–5 Pa of nitrogen gas pressure were maintained in the chamber to minimize charging. Raw diffraction data were acquired and processed using the AZTEC software package. For each sample, several orientation maps with a step size of 15 μm were collected from selected regions of the section. Carbon elemental data were then obtained from EDS for these same regions, with step sizes ranging from 3.2 to 6.7 μm. The EDS data were collected separately from the EBSD because the operating voltage for EDS (below 15 kV) needs to be lower than that used for EBSD (30 kV) to obtain high-quality data for graphite. Subsequently, an orientation map with a step size of 30 μm was acquired for the majority of the section.

Orientation data from EBSD with a 15-μm step size were combined with element data from EDS to identify unindexed points. The combined data were then used to analyze grain size, aspect ratio, and shape preferred orientations. Since graphite, the secondary phase in the samples, cannot be reliably detected in EBSD, we used the results from EDS to locate the graphite. This data helped define pixels corresponding to graphite and facilitated the interpolation of unindexed points in the 15-μm step size orientation data. As illustrated in Figure 4(a), the brightness in the EDS map represents different graphite signal intensities (from 1 to 6), with higher brightness indicating a higher content. Note that due to the relatively low spatial resolution of EDS, individual graphite particles cannot be identified, and only graphite-rich areas are revealed. The proportion of pixels occupied by the graphite signal significantly exceeds the actual graphite content of 1 vol.%. A threshold of intensity $\geqslant 2$ was then applied to the data, removing very weak signals that may result from noise (Fig. 4(b)). This is because the activation volume for X-rays is large. In an ice + graphite sample, X-rays are sampled from a volume of ice several micrometers in diameter. Areas with low carbon signals may represent a small number of graphite particles in a larger volume of ice. To match the dimensions of the EBSD data, the pixel size of the EDS map was adjusted to match the step size of EBSD. Then these pixels were attributed to the graphite phase in the EBSD data. By combining the EBSD data from ice and graphite, a data set with two phases was obtained (Fig. 4(e)). Then the orientation map was filled via interpolation. The ice grains were digitally "reconstructed" from the processed data using the MTEX algorithm (Bachmann et al., 2010). In this process, misorientation between neighbouring pixels exceeding 10° were used to identify grain boundaries, while misorientation exceeding 2° were used to identify subgrain boundaries. Grain size was determined as the equivalent diameter of a circle with the area of each grain in cross section. Note

that grain size determined this way represents the size of a 2-D cross section of a 3-D grain. In the analysis of the average grain size for a map, grains containing no more than 5 pixels or lying on the edge of the map were excluded.

## 2.5 Analysis of crystallographic orientations

Orientation distributions were generated from the complete set of raw EBSD data with a step size of 30 μm using the MTEX toolbox in MATLAB (Bachmann et al., 2010; Mainprice et al., 2015). To quantify the strength of the CPOs, both the J-index (Bunge, 1982) and the M-index (Skemer et al., 2005) were used. The J-index, based on a calculated orientation distribution function, increases from 1 (random) to infinity (single crystal). The M-index, which is based on the distribution of random-pair misorientation axes, increases from 0 (random) to 1 (single crystal).

Since the CPOs of sheared ice are often characterized by double clusters of $c$ axes (e.g., Kamb, 1972; Bouchez and Duval, 1982; Jackson, 1999; Qi et al., 2019), an angle, $\varphi$, was used to quantify the relative orientation between the two clusters. We adopted a method similar to that of Figure 2 in Qi et al. (2019), based on an approach used by Bouchez and Duval (1982). As illustrated in Supplementary Figure S2(b), pole figures were generated using a lower hemisphere equal-area projection, with the shear plane (green circle) oriented perpendicular to the page. In the stereonets, angles ranging from 0 to 180° were defined on the shear plane. At a given angle, two semicircles with 5° between them (orange circle) were drawn perpendicular to the page. The number of data points falling between these semicircles was counted, normalized, and plotted as the frequency for each angle in a histogram. The angle $\varphi$ was defined as the angle between the two peaks in the histogram (Supplementary Fig. S2(c)).

## 2.6 Simulations from SpecCAF model

The observed CPOs were compared to predictions from the spectral continuum anisotropic fabric evolution (SpecCAF) model (Richards et al., 2021). The SpecCAF model cannot directly simulate some microstructural changes, such as grain-scale deformation, like other models, e.g., ELLE (Jessell et al., 2001). Instead, SpecCAF simulates the evolution of CPO, which is achieved through modelled processes involving grain rotation, diffusion that simulates rotational recrystallization, and an orientation-dependent source term that emulates migration recrystallization (Richards et al., 2021). Nevertheless, the results from the SpecCAF model agree well with the CPOs reported in deformation experiments performed in Craw et al. (2018), Qi et al. (2019) and Fan et al. (2020). Consequently, the model can be used as a proxy to compare the samples in this study to how the CPO of pure ice would evolve at very high strains.

The numerical model adapted from Richards et al. (2021) produces simulated pole figures representing the distribution of $c$-axis orientations, and the angle, $\varphi$, between clusters. Different values of $\varphi$ were simulated based on the numerical model with a variety of $\beta$. $\beta = k\beta_0$, where $\beta$ controls the relative magnitude of the effect of migration recrystallization on the modelled CPO, $0 \leq k < 1$, and $\beta_0$ represents the value found for natural ice at $T = -5°C$. Different values of $k$ (0.2, 0.4, 0.5, 0.6, 0.8 and 1) were used to assess the relative contributions of GBM and lattice rotation to CPO.

## 3 Results

### 3.1 Starting material

The ice microstructure of the starting materials was similar in character to that described in Qi et al. (2017). The microstructure of an undeformed sample is illustrated in Figure 5. The undeformed sample has a roughly homogeneous ice microstructure. Graphite particles form graphite clusters and graphite-rich regions approximately uniformly distributed between ice grains. The mean ice grain size is 150 μm, which is approximately 80 μm smaller than pure polycrystalline ice made in a similar method (e.g., Qi et al., 2017). The initial crystallographic orientation is approximately random with a M-index of 0.002 and a J-index of 1.08. There are very few intragranular distortions, as shown by very few subgrain boundaries within the grains.

### 3.2 Mechanical data

The plot of displacement and temperature versus time for 6 passes of sample ECAP_38 is illustrated in Figure 6, while plots for other samples are illustrated in Supplementary Figure S1. The temperature, monitored at the corner of the channel, was stable throughout an experiment run. The temperature variation for the 1st to 5th pass is within 0.2°C, with an average temperature of approximately −5°C, but the 6th pass shows a larger perturbation and a slightly lower average temperature ($\sim -5.5$°C), likely due to variations in laboratory temperature and humidity. After the progressive initial loading process, all experiment runs were conducted at a constant load. The displacement typically increased rapidly with time in the first few hours due to compaction and bending of the sample's head (refer to Fig. 1 for the "head" of the sample). Then displacement increased roughly linearly with time as the sample deforms at a nearly steady state, and increased rapidly again in the last few hours caused by the loss of back pressure, where the plug is pushed out of the channel by the sample.

Since load and displacement data from ECAP experiments are not well-suited for rigorous analysis of mechanical behavior (see Valiev and Langdon, 2006, for a review), this paper will not include an in-depth analysis of the mechanical properties, consistent with the limitations of most ECAP deformation studies. However, as ECAP is a relatively new technique for ice, the deformation pathway the sample experiences during pressing is described here. For each pass, the deformation starts at the head of the sample, with the shear stress in the head gradually rising to a maximum, resulting in a nominal equivalent strain of approximately 0.6 in the deformed part. Meanwhile, the rest of the sample is annealed at $\lesssim -5$°C. Subsequently, the deformed head slides through the outlet tube, where the shear stress drops to zero. Simultaneously, the ice behind the head arrives at the corner, leading to a gradual increase in shear stress to its maximum value and obtaining a nominal shear strain of approximately 1.0, while the rest, including both deformed head and undeformed parts, continues to anneal at $\lesssim -5$°C. This sequence is repeated until the tail of the sample passes the corner. A point in the sample only experiences shear stress when passing through the corner. During the rest of the time, the sample anneals.

Figure 6(c) shows the straining-annealing sequence for the 6-pass sample. In each pass, the head undergoes strain at a rate of $1.5 \times 10^{-6}$ (1st pass) to $3 \times 10^{-6}$ (6th pass) s$^{-1}$ for $\sim$30% time ($\sim$100 h for 1st pass and $\sim$50 h for 6th pass) and is then annealed for $\sim$70% time ($\sim$270 h for 1st pass and $\sim$120 h for 6th pass). Ice in the middle of the sample undergoes annealing for $\sim$35% time ($\sim$130 h for 1st pass and $\sim$60 h for 6th pass) before straining and $\sim$35% time ($\sim$130 h for 1st pass and $\sim$60 h for 6th

pass) after straining. The tail anneals for ∼70% time (∼270 h for 1st pass and ∼120 h for 6th pass) before straining. Again, we note that our samples were deformed differently from those reported in previous studies, in which the whole sample deformed simultaneously (e.g., Kamb, 1972; Qi et al., 2019), rather than having a deformation front passing through the sample. Thus, we refer to our equivalent and shear strains as the nominal equivalent and shear strains, denoted by $\varepsilon'$ and $\gamma'$.

## 3.3   Crystallographic preferred orientations

In this subsection, CPOs in all samples deformed to different strains (different numbers of ECAP passes) are described, as illustrated in Figure 7. The CPOs of all samples feature two clusters of $c$ axes. The primary cluster consistently aligns perpendicular to the imposed shear plane at all strains, while the secondary cluster is situated in the profile plane antithetic to the shear direction. The CPO strength, as revealed by J- and M-indexes, generally increases with increasing strain, as illustrated in Figure 8. In all samples, the elongation of $c$-axes clusters is observed sub-perpendicular to the shear direction. The primary cluster becomes more elongated with increasing strain, while the secondary cluster becomes less elongated, as illustrated in Figure 9.

The angle between clusters varies between 45° and 60° for all samples, being roughly constant with increasing strain. Except for ECAP_33_1P ($\varepsilon'_1 \approx 0.6$), the sample with the lowest strain, the secondary clusters in all samples are weaker (indicated by a lower value of multiples of uniform density, MUD) compared to the primary clusters. A comparison between the strength of the two clusters suggests that the secondary cluster generally weakens with increasing strain. A more quantitative comparison is given in Figure 8(c). The ratio of number of data points in the primary and the secondary clusters increases with increasing strain.

In all samples, the CPOs are characterized by prominent girdles of $[11\bar{2}0]$ axes ($a$ axes) and $[10\bar{1}0]$ axes (poles to $m$ planes) within the shear plane, along with weaker girdles of $[11\bar{2}0]$ and $[10\bar{1}0]$ axes perpendicular to the secondary $c$-axis clusters. In the two samples with $\varepsilon' \geqslant 3.0$, $a$ axes strongly concentrate parallel to the shear direction, while in the sample with lower strains, the distribution of $a$ axes is more uniform in the girdles with some maxima parallel to the vorticity axis. The distributions of the poles to $m$ plane are similar to the distributions of $a$ axes in all samples.

## 3.4   Microstructure

The microstructures, grain-size distributions and shape distributions for all samples are presented in Figure 10. Note that the analyses of grain size are based on larger areas (4–6 times) than those presented in the maps. The presence of graphite particles in the samples induces many difficulties in the analyses of microstructure. To prevent graphite from covering the sample surface, the surface cannot be polished down to grit sizes finer than 400. Moreover, graphite conducts heat when scanned by the electron beam, which accelerates sublimation of ice next the graphite and results in areas of unindexed points larger than the graphite particle. Although we processed the microstructure incorporating the EDS analysis, the grain boundaries reconstructed from orientation data are affected by the identified graphite phase. All samples are characterized by lobate grain boundaries and irregular grain shapes. Many grains are elongated, especially in the high-strain samples. Subgrain boundaries are observed in all samples, and are more frequent in the samples with higher strains.

In all samples, the distributions of grain size are skewed, with a peak at finer grain sizes and a long tail extending to coarser grain sizes. In samples deformed by 1–3 passes ($\varepsilon' \leqslant 1.8$), grain sizes vary greatly, with many grains larger than 500 μm and many smaller than 100 μm. In samples deformed by 4–6 passes ($\varepsilon' \geqslant 2.4$), the deviation in grain size is smaller, the grain-size distribution is better fit by a log-normal distribution (orange curves in Fig. 10(b)), and the mean grain size is similar to the starting grain size. In all samples, grains are elongated after deformation and form clear shape preferred orientations (SPOs). The distribution of aspect ratios varied very little in samples deformed by 2–6 passes. All samples have a SPO with a long axis at $\sim 10°$ to the shear plane, except for the 1-pass sample, where the angle is $26°$.

## 3.5 Numerical modelling

The experimental CPOs are compared to numerical modelling results from the SpecCAF model (Richards et al., 2021). The only inputs required by this model are deformation, temperature, and initial CPO. As the experiments are performed in simple shear, the model is also run in this deformation condition, though the model does not account for the complexities introduced by the ECAP process. The predicted angle between the $c$-axis clusters $\varphi$ from the modelling is plotted alongside the experimental results in Figure 11, showing that this angle is highly dependent on the amount of GBM in the model (controlled by $\beta$). For high values of $\beta$, $\varphi$ stays roughly constant around $70°$ even at large strains. As GBM is reduced in the model, the angle between the two clusters $\varphi$ decreases, and decreases with increasing strain. It is found that a value of $\beta = 0.6\beta_0$ most closely matches the experimental results in this paper. The modelled CPO pole figures are also shown in Figure 12 for this value.

## 4 Discussion

CPOs of all deformed samples are characterized by two $c$-axis clusters. The angle between the two clusters stabilizes between $45°$ and $60°$ as strain increases. Here, we note that all pole figures are drawn based on all orientation data, instead of one point per grain. This is due to the fact that EBSD data for pole figures were collected at a large step size (30 μm), in order to cover a large sample area in a limited time. At such a large step size, there are roughly $4 - 6$ points across a grain, which is insufficient to do grain reconstruction. However, based on grain sizes reconstructed with a 15-μm step size (Figs. 10(a) and (b)), grain-size distributions in high-strain samples ($4 - 6$ passes) can be well fit by a log-normal distribution, suggesting there is no dramatic variations in grain size. These pole figures are unlikely to be dominated by a few very large grains. Therefore, we are confident that pole figures drawn using all orientation data represent the CPOs in these samples well.

In this discussion, we first need to discuss the influences of experimental settings introduced in this study. ECAP deformation utilized in our study differs from simultaneous deformation applied to the entire sample in compression, shear and torsion experiments (e.g., Kamb, 1972; Durham et al., 1992; Li et al., 2000; Journaux et al., 2019; Qi et al., 2019). In our experiments, apart from the single-pass sample (ECAP_33_1p), which underwent a single deformation and annealing, the multi-pass samples experienced cyclic deformation and annealing. Additionally, to slow down grain growth, we introduced $\sim 1$ vol.% graphite particles into the sample. The potential influence of a second phase on the evolution of CPO must be taken into account. We will focus on the influences of annealing and graphite in Sections 4.1 and 4.2, and analyze the elongated $c$-axis clusters in

Section 4.3. A careful comparison of the CPO observed in laboratory experiments, field samples, and models will be performed in Sections 4.4, 4.5, and 4.6, respectively. Finally, based on these analyses, we will refine the model of CPO development under simple shear deformation in Section 4.7 and discuss implications for natural ice in Section 4.8.

## 4.1 Influence of annealing

As described in Section 3.2, during an ECAP experiment, a sample was annealed before and/or after passing through the corner. Here, we look at the influence of annealing on CPO. The sample annealed at $-3.5°$C for 24 days has the same CPO patterns as the original sample ECAP_38_6p before annealing, but the strength of the secondary cluster increases, as illustrated in Figure 7. The CPO also slightly weakens after annealing, as illustrated in Figure 8. This observation is in good agreement with previous studies. Wilson (1982) performed annealing experiments at $-1°$C on samples previously deformed at $-10$ and $-1°$C.

He reported that the annealing did not significantly change the CPO patterns in the $-1°$C samples, but modified the patterns in the $-10°$C samples. Hidas et al. (2017) carried out annealing experiments at $-5$ and $-2°$C up to a maximum of 78 h on samples deformed at $-7°$C. The authors found that grain growth proceeds at the expense of domains with high intragranular misorientations, consuming first the most misorientated parts of primary grains. Journaux et al. (2019) carried out an annealing experiment at $-7°$C for 72 h on a sample sheared in torsion at $-7°$C. These authors reported a much larger grain size in the

annealed sample, suggesting strong GBM; however, they found that the pre-existing CPO pattern was not modified by the annealing. Fan et al. (2023) performed annealing experiments at $0°$C on natural samples from the Priestley Glacier, Antarctica, and reported that CPO was modified only when abnormal grain growth occurred after at least 72 hours.

Based on the above discussion, we suggest that annealing will not alter the patterns of CPO obtained after deformation, but can slightly weaken the strength of CPO. During annealing, dislocation density and strain energy are reduced by recrystalliza-

340 tion (Poirier, 1985). At the same nominal strain, a sample that just experienced annealing may have a lower dislocation density compared to a sample continuously deformed without annealing. Consequently, in subsequent deformation, additional strain should be required to compensate for the reduced dislocation density and strain energy resulted from annealing, reaching the threshold to initiate dynamic recrystallization (Bacca et al., 2015). Thus, given the same nominal strain, the effect of strain on microstructure in our ice samples could be weaker than that in ice samples deformed continuously. In other words, the nominal

strains calculated from Eq. 1 in our experiments could be an overestimation for CPO evolution, which could explain the lower CPO strength compared with the samples in Qi et al. (2019).

## 4.2 Influence of second phase

Due to rapid grain growth, we could not perform an ECAP experiment on particle-free ice without fracturing it. So, a small fraction of graphite ($\sim$1 vol.%) was added to inhibit grain growth. Here we investigate the impact of doping $\sim$1 vol.% graphite

on the evolution of CPO. Cyprych et al. (2016) and Wilson et al. (2019) conducted axial compression experiments on pure $D_2O$ ice and $D_2O$ ice doped with 20 or 40 vol.% graphite ($<$150 or 150–355 μm in diameter), or 20 or 40 vol.% calcite ($<$ 150 or 150–355 μm in diameter) at temperatures $10°$C below the melting point (actual at $-7°$C) and ambient pressures. All two-phase samples exhibited ice CPO patterns that were remarkably similar to those of single-phase samples, albeit with a

weaker CPO intensity compared to single-phase samples. Cyprych et al. (2016) and Wilson et al. (2019) suggested that the mechanisms influencing CPO development in two-phase samples and ice-only samples are consistent, involving GBM and nucleation, where GBM may be hindered by the second phase. Song et al. (2005) conducted axial compression experiments on pure water ice and ice doped with 1 wt.% ($\sim$0.43 vol.%) soil particles (sieved to $50\pm10$ μm) at $-10°$C. They found a more random orientation of $c$ axes, that is, weaker fabric, after a strain of $\sim$0.1 in ice with particles than in particle-free ice. In our samples, the fraction of the second phase is much lower than the samples of Cyprych et al. (2016) and Wilson et al. (2019), and the strain is much larger than the samples of Song et al. (2005). Extrapolating from their results, we suggest the graphite doped in our sample may reduce the intensity of CPO, but cannot alter the CPO patterns.

Based on the discussions in Sections 4.1 and 4.2, we suggest that the patterns of CPO observed in our experiments can be compared with those reported in previous studies. The CPO evolution with increasing strain could be slower than those in previous studies, as the nominal strains of our samples are possibly overestimated. Consequently, in the following discussions, we focus on CPO patterns.

### 4.3 The elongated $c$-axis clusters

In our experiments, both primary and secondary clusters are elongated in the direction sub-perpendicular to the shear direction. These observations are consistent with studies on ice samples deformed primarily by simple shear in laboratory settings, for which CPOs with a large number of grains have been measured (Kamb, 1972; Bouchez and Duval, 1982; Li et al., 2000; Qi et al., 2019; Journaux et al., 2019). Li et al. (2000) attributed elongated clusters to extensional deformation in the shear plane, perpendicular to the shear direction, induced by the flattening of the sample during shear deformation. However, elongation was also observed in samples sheared by torsion, in which no axial load was applied and the deformation kinematics were ensured to be simple shear (Bouchez and Duval, 1982; Journaux et al., 2019). Meanwhile, natural ice samples from shear zones provide diverse results. The CPOs reported in Hudleston (1977) for the marginal shear zone of Barnes Ice Cap are all characterized by round $c$-axis clusters. The CPOs reported in Jackson (1999) and Jackson and Kamb (1997) for the marginal shear zone of Ice Stream B, however, have elongated $c$-axis clusters perpendicular to the shear direction. Recently, studies employing modern EBSD techniques provide much more data for CPOs in natural ice samples. Monz et al. (2021) found an elongated primary $c$-axis cluster in a shear-dominated region of Storglaciären Glacier, while Thomas et al. (2021) found a roughly round primary $c$-axis cluster in the shear margin of Priestley Glacier. Furthermore, the elongated cluster is also observed in numerical simple-shear simulations by Llorens et al. (2016, 2017) and Richards et al. (2021), in which no flattening strain is allowed. Richards et al. (2021) found that $c$-axis clusters were elongated primarily due to lattice rotation. Thus, we hypothesize that the elongation of clusters results from rotation recrystallization. At larger strains, most grains have their $c$ axes aligned sub-normal to the shear plane. When rotation recrystallization occurs, the orientations of the nuclei are different from the host grains, resulting in a dispersion of orientations (see Fig. 12(c)). Lattice rotation tends to rotate the $c$ axes of the nuclei back to the normal to the shear plane. The rotation process is easier when the rotation axis is in the same orientation as the vorticity axis of deformation, that is, the rotation is in the shear direction normal to the shear plane (Rathmann et al., 2021). Comparatively, the rotation process is slower when the rotation is normal to the shear direction. This process results in $c$ axes spreading normal to the shear

direction. The round clusters observed in glaciers could be explained by the fact that the stress and temperature conditions favor migration recrystallization over rotation recrystallization. We hope to further investigate the cause of the elongated clusters in the future.

## 4.4 The evolution of the $c$-axis clusters: comparison with experiments

Figure 11 illustrates the evolution of the angle between clusters, $\varphi$, as strain increases, comparing with data from prior experimental studies, numerical models, and naturally deformed ice. This figure was plotted in the same manner as Figure 8 in Qi et al. (2019), focusing on a temperature of $-5\pm4°$C. The values of $\varphi$ are scattered between 43 and 81° for all experimental samples with double $c$-axis clusters.

At warm temperatures ($-5\pm4°$C), Li et al. (2000) remains the only study that reported a single $c$-axis cluster at a shear strain $< 2.5$. However, Wilson and Peternell (2012) present more complex patterns of CPOs, which are characterized by double-cluster patterns at a shear strain of $\sim$1 at $-2°$C, despite conducting their experiments using the same apparatus and kinematic constraints as Li et al. (2000). Two recent studies both reported double-cluster CPOs at warm temperatures (Qi et al., 2019; Journaux et al., 2019). Qi et al. (2019) hypothesized that the key element in generating a single cluster is high shear strain. In this study, the CPOs were characterized by double $c$-axis clusters below a nominal shear strain of $\sim$6.2, but the secondary cluster became very weak at this maximum strain. It is worth noting that under the same strain, the strength of the CPO in the sample deformed by ECAP is weaker than that in samples deformed continuously due to the effects of annealing discussed in Section 4.1. This result supports the earlier hypothesis that the contribution of GBM to CPO formation decreases with increasing strain.

The evolution of $\varphi$ with strain is quite interesting. Based on the data in literature, one could tentatively suggest that $\varphi$ decreases with increasing strain (Qi et al., 2019). However, considering the results of this study, $\varphi$ decreases to $\sim$50° and remains roughly constant until the secondary cluster weakens to almost invisible. This observation suggests that $\varphi$ may not necessarily change, instead the secondary cluster retains its orientation, and weakens with increasing strain. Thus, the model of CPO development in Qi et al. (2019) needs to be refined.

## 4.5 The evolution of the $c$-axis clusters: comparison with natural samples

In Qi et al. (2019), natural CPOs from a glacial shear zone reported by Hudleston (1977) were compared with the laboratory CPOs in simple shear. The key agreements between natural and laboratory CPOs were that (1) the primary cluster maintains a sub-perpendicular orientation to the shear plane at all shear strains; (2) the angle between two clusters diminishes as strain increases; and (3) CPOs feature a solitary $c$-axis cluster at high strains. Here, we review these with new results from natural and laboratory samples from this study. Gerbi et al. (2021) investigated the microstructure of the lateral shear margins of a temperate glacier, Jarvis Glacier. Despite the fabric being relatively weak due to high water content, and short flow distance, $c$ axes are slightly more concentrated in regions closer to the margin where strain is larger. However, without azimuth angles, it is not possible to determine the orientation of these samples relative to the shear plane. Monz et al. (2021) used a composite section method to obtain a representative, bulk CPO on a coarse-grained glacial ice sample from Storglaciären, Sweden. The

results yielded a pronounced $c$-axis cluster sub-perpendicular to the shear plane. Thomas et al. (2021) analyzed an ice core from the floating lateral shear margin of Priestley Glacier, Antarctica. They reported that most samples have a single cluster sub-perpendicular to shear, with some samples having a secondary cluster. The agreements in CPOs between natural and laboratory samples hold true for the orientation of the primary cluster and for the requirement of a single cluster. In these natural samples (Hudleston, 1977; Monz et al., 2021; Thomas et al., 2021), the smallest value of $\varphi$ in samples with double clusters is 35° from the core at a depth of 2.4 m (Thomas et al., 2021). Due to its shallow depth, the microstructure of this sample was not developed completely due to plastic deformation, but may be affected by annealing and other processes. Besides this sample, the next smallest $\varphi$ is 40° in a low strain sample from the shear zone (Hudleston, 1977). The absence of low angles (<40°) between clusters in natural samples is in good agreement with laboratory data from this study, and is possibly because the secondary cluster weakens with increasing strain, rather than moving towards the primary cluster. Thus, both natural observations and laboratory experiments suggest that the secondary cluster retains in its orientation and weakens until it disappears with increasing strain.

## 4.6 The evolution of the $c$-axis clusters: comparison with numerical models

The evolution of the $c$-axis clusters was compared between CPOs from experiments and a viscoplastic fast-Fourier-transform (VPFFT) model (Llorens et al., 2017) in Qi et al. (2019). The numerical models fit some experimental observations but clearly need more work. The SpecCAF model, incorporating recrystallization, lattice rotation and grain rotation processes, yielded excellent quantitative agreement in the CPOs from experimental observations (Richards et al., 2021). Applying this model to simple shear yields roughly constant values of $\varphi$ with increasing strain at shear strains $> 1$, when the effect of GBM is similar to or smaller than the value estimated for natural ice ($\varphi \approx 74°$, 71°, 65° and 60° for $\beta = 1$, 0.8, 0.6 and $0.5\beta_0$, respectively; see Richards et al. (2021)). This result is in qualitative agreement with the experiments that the secondary cluster retains its orientation and weakens with increasing strain. When the effect of GBM is reduced to much weaker than natural ice ($\beta = 0.2\beta_0$), $\varphi$ decreases to <40° with increasing strain and the fabric becomes a single $c$-axis cluster at a shear strain < 1 (the dark blue curve in Fig. 11). This model suggests that lattice/subgrain rotation tends to move the secondary cluster towards the primary cluster, while GBM competes with rotation. The resulting angle between clusters is balanced by the two processes.

## 4.7 A refined model of CPO development in simple shear

Our earlier contribution (Qi et al., 2019) proposed that the CPO development of ice with increasing strain during simple shear was due to a transition in the dominant recrystallization process from lattice/subgrain rotation and GBM. Here, we refine this model based on new experimental data, as illustrated in Figure 12.

GBM is typically driven by differences in dislocation density between grains on either side of the grain boundary, which is often associated with strain inhomogeneity across the grains (Urai et al., 1986). Grains with low resolved shear stresses, or Schmid factors, on the basal plane (also referred to as poorly oriented for easy slip), have to deform through slip on non-basal slip systems. For ice, these non-basal-slip dislocations are more difficult to glide (Duval et al., 1983), and there will be more than one interacting slip systems. Therefore, grains with basal planes poorly oriented for easy (basal) slip are inferred to have

higher internal distortion, leading to higher dislocation density (Vaughan et al., 2017). Grains with higher-Schmid factors on
the basal plane tend to grow by consuming grains with lower-Schmid factors (Fig. 12(c)). In simple shear, basal planes parallel
to the shear plane and normal to the shear direction are the two orientations with high Schmid factors. Thus, GBM generates
two $c$-axis clusters, perpendicular to the shear plane (primary) and parallel to the shear direction (secondary).

Lattice rotation with basal glide produces a $c$-axis cluster perpendicular to the shear plane (primary). With increasing strain,
this cluster strengthens rapidly, while $c$ axes in other orientations are rotated towards this cluster. Subgrain rotation can trigger
a nucleation process, leading to the generation of new small grains, which is commonly referred to as subgrain rotational
recrystallization (Guillope and Poirier, 1979; Urai et al., 1986; Bestmann and Prior, 2003). Orientations of crystallographic
axes in the recrystallized grains generally are randomly diffused equivalents of the stronger CPOs in the host grains (Jiang
et al., 2000; Bestmann and Prior, 2003; Craw et al., 2018). We also hypothesize that in simple shear the dispersed orientations
are easier to rotate back if the rotation axis is normal to the shear direction in the shear plane, and thus leading to a preferential
dispersal of the $c$ axes. Qi et al. (2019) also suggested the elongation of clusters may also relate to GBS-aided rotation around
the vorticity axis synthetic to the shear direction. That hypothesis has not been tested yet.

Figure 12(d) is a refinement based on the schematic diagram proposed by Qi et al. (2019) illustrating the development of
CPO patterns with shear strain, incorporating our experimental data. Due to the combined effect of lattice rotation and GBM,
the initial CPO patterns feature two $c$-axis clusters normal to the shear plane and parallel to the shear direction. With increasing
strain, the secondary cluster is weakened and rotated towards the primary cluster due to lattice rotation. During the deformation
process, GBM will continue to promote the growth of grains with the $c$ axes normal to the shear plane and parallel to the shear
direction. Subgrain rotation recrystallization plays a critical role in CPO formation by supplying grains with varied orientations
compared to the primary cluster, thus providing the necessary grains for GBM. The orientation of secondary clusters is a result
of the competition between lattice rotation and GBM. In the earlier model, the secondary cluster was thought to move towards
the primary cluster continuously with increasing strain. A single cluster can thus develop as the secondary cluster merges
into the primary cluster. In our refined model, given a steady-state deformation, the secondary cluster will be stationary with
increasing strain, once a "balance" between lattice rotation and GBM is reached. With increasing strain, instead of moving, the
secondary cluster progressively weakens, leading to the predominance of the primary. Consequently, there are fewer grains in
low-Schmid-factor orientations, which will reduce the number of grains with high dislocation density, effectively reducing the
driving force for GBM. Thus, with increasing strain, the contribution from GBM reduces, and the CPO will eventually become
a single $c$-axis cluster fabric.

## 4.8 Implications to natural ice

As ice is a highly anisotropic material, once formed, the CPO has a significant influence on the mechanical strength of ice, and
thus, models of the flow of natural ice often rely on applying an anisotropy factor to the laboratory-derived flow laws (Pimienta
and Duval, 1987). Although the models using an anisotropy factor obtained from historic observations generally predict the
correct magnitude of glacial flow rates, Azuma's CPO-only flow law (Azuma, 1994) best describes the strength evolution as
strain increases (Fan et al., 2021a). The anisotropic factor in Azuma's CPO-only flow law is not based on phenomenological

data but rather is calculated from the orientation data of $c$ axes. Thus, this study and many previous studies focusing on the CPO development in ice aim to understand the physical processes that control the evolution of $c$-axis orientation during deformation, and ultimately, to better constrain the anisotropy factor and predict the glacial flow rates, especially for ice-stream margins, where shear deformation is severe.

One important result from the observations of this study is that the secondary $c$-axis cluster retains its orientation and weakens with increasing strain. This result changes our previous intuitive hypothesis and provides different values of anisotropy factors at strains when the $c$-axis fabric is evolving from double clusters to a single cluster. Such fabric evolution could occur at regions not too far away from the dome, where the shear deformation just starts, and possibly at the upstream regions of ice-stream margins, where the shear plane changes from horizontal to vertical, and the fabric has to evolve accordingly. However, it is necessary to note that the laboratory observed microstructures and their evolutions are obtained at strain rates and stresses larger than those in natural ice bodies. The contribution from rotation recrystallization in natural ice could be weaker, as stresses are smaller. Moreover, natural ice is usually impure. Insoluble particles and air bubbles could accumulate along grain boundaries and reduce grain boundary mobility and inhibit GBM, which possibly also occurred in the experiments of this study. The effects of the two mechanisms can only be qualitatively discussed for natural conditions. The microstructural processes observed in laboratory experiments provides good constraints on models and simulations (e.g., Richards et al., 2021; Hunter et al., 2023), which could allow for the extrapolation of laboratory results to natural ice.

## 5 Conclusions

Utilizing the ECAP technique, we achieved a nominal equivalent strain of ~3.6 (a nominal shear strain of ~6.2) in polycrystalline ice doped with ~1 vol.% graphite deformed at $-5°$C in roughly simple shear. The cyclic annealing introduced by ECAP deformation, along with the presence of a small amount of graphite, may reduce the strength of the CPO, but will not alter the patterns of the CPO. All samples develop a primary $c$-axis cluster perpendicular to the shear plane and a secondary $c$-axis cluster in the profile plane antithetic to the imposed shear direction. The orientation of the primary cluster does not change as a function of strain. The secondary cluster roughly retains its orientation but weakens as strain increases. Annealing of the 6-pass sample at $-3.5°$C for 24 days revealed the same CPO patterns as before annealing. The strength of this CPO became slightly weaker and the secondary cluster became slightly stronger after annealing. A combination of our data and published literature data, and comparisons with numerical models reveal the key processes that control the evolution of CPOs in ice during shear. The CPO patterns results from a balance of two competing mechanisms: lattice rotation due to dislocation slip, strengthening the primary cluster and rotating and weakening the secondary one, and growth of grains by strain-induced GBM, strengthening both clusters and rotating the secondary cluster back. GBM contributes less as shear strain increases.

*Data availability.* Data are available on an online data repository: Data of "Evolution of crystallographic preferred orientations of ice sheared to high strains by equal-channel angular pressing". figshare. Dataset. https://doi.org/10.6084/m9.figshare.23807400.v2

*Author contributions.* CQ and DJP designed the research. CQ designed the deformation equipment. QW performed experiments. QW, SF and RW performed analyses. DHR performed simulations. All authors participated in the interpretation of results. QW and CQ wrote the first draft, and all authors edited the manuscript.

*Competing interests.* The authors declare that they have no conflict of interest.

*Disclaimer.* TEXT

*Acknowledgements.* We thank Jianhua Rao for his help with designing the ECAP die and Kaitlin Keegan for her editorial work. We sincerely appreciate the comments from Christopher Gerbi and an anonymous reviewer, whose comments greatly improved this manuscript. This work was supported by a NSFC grant 41972232 (to CQ), the Key Research Program of the Institute of Geology and Geophysics, CAS, IGGCAS-201905 (to CQ), and Marsden Fund of the Royal Society of New Zealand UOO052 (to DJP).

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

**Table 1.** Summary of experiments

| Sample | Graphite fraction | Passes | Load (kg) | Length (mm) | $\varepsilon'$ [1] | Part for analysis | points of EBSD data | area (mm$^2$) | $\varphi$ [2] |
|---|---|---|---|---|---|---|---|---|---|
| ECAP_33 | 2.1 wt.% (0.9 vol.%) | 1 | 42.5 | 110 | 0.6 | ECAP_33_1P | 144045 | 14.85×8.73 | 50° |
| ECAP_19 | 3.6 wt.% (1.5 vol.%) | 1 | 42.5 | 98 | 0.6 | | | | |
| | | 2 | 37.5 | 100[3] | 1.2 | ECAP_19_2P | 99099 | 12.87×6.93 | 55° |
| | | 3 | 37.5 | 50 | 1.8 | ECAP_19_3P | 104550 | 12.75×7.38 | 50° |
| ECAP_21 | 1.8 wt.% (0.8 vol.%) | 1 | 42.5 | 110 | 0.6 | | | | |
| | | 2 | 37.5 | 105 | 1.2 | | | | |
| | | 3 | 37.5 | 93 | 1.8 | | | | |
| | | 4 | 32.5 | 85 | 2.4 | ECAP_21_4P | 108035 | 15.81×6.15 | 45° |
| ECAP_34 | 2.6 wt.% (1.1 vol.%) | 1 | 42.5 | 107 | 0.6 | | | | |
| | | 2 | 32.5 | 99 | 1.2 | | | | |
| | | 3 | 32.5 | 95 | 1.8 | | | | |
| | | 4 | 32.5 | 85 | 2.4 | | | | |
| | | 5 | 32.5 | 65 | 3.0 | ECAP_34_5P | 112892 | 13.86×7.97 | 60° |
| ECAP_38 | 2.2 wt.% (0.9 vol.%) | 1 | 42.5 | 105 | 0.6 | | | | |
| | | 2 | 32.5 | 100 | 1.2 | | | | |
| | | 3 | 32.5 | 95 | 1.8 | | | | |
| | | 4 | 32.5 | 95 | 2.4 | | | | |
| | | 5 | 32.5 | 92 | 3.0 | | | | |
| | | 6 | 32.5 | 94 | 3.6 | ECAP_38_6P | 161100 | 16.11×9.00 | 55° |

[1] $\varepsilon'$ is nominal equivalent strain.

[2] $\varphi$ is an angle between the two clusters.

[3] An extra piece was added, so that the length is longer than or similar to that of the previous pass. This also applies to sample ECAP_38.

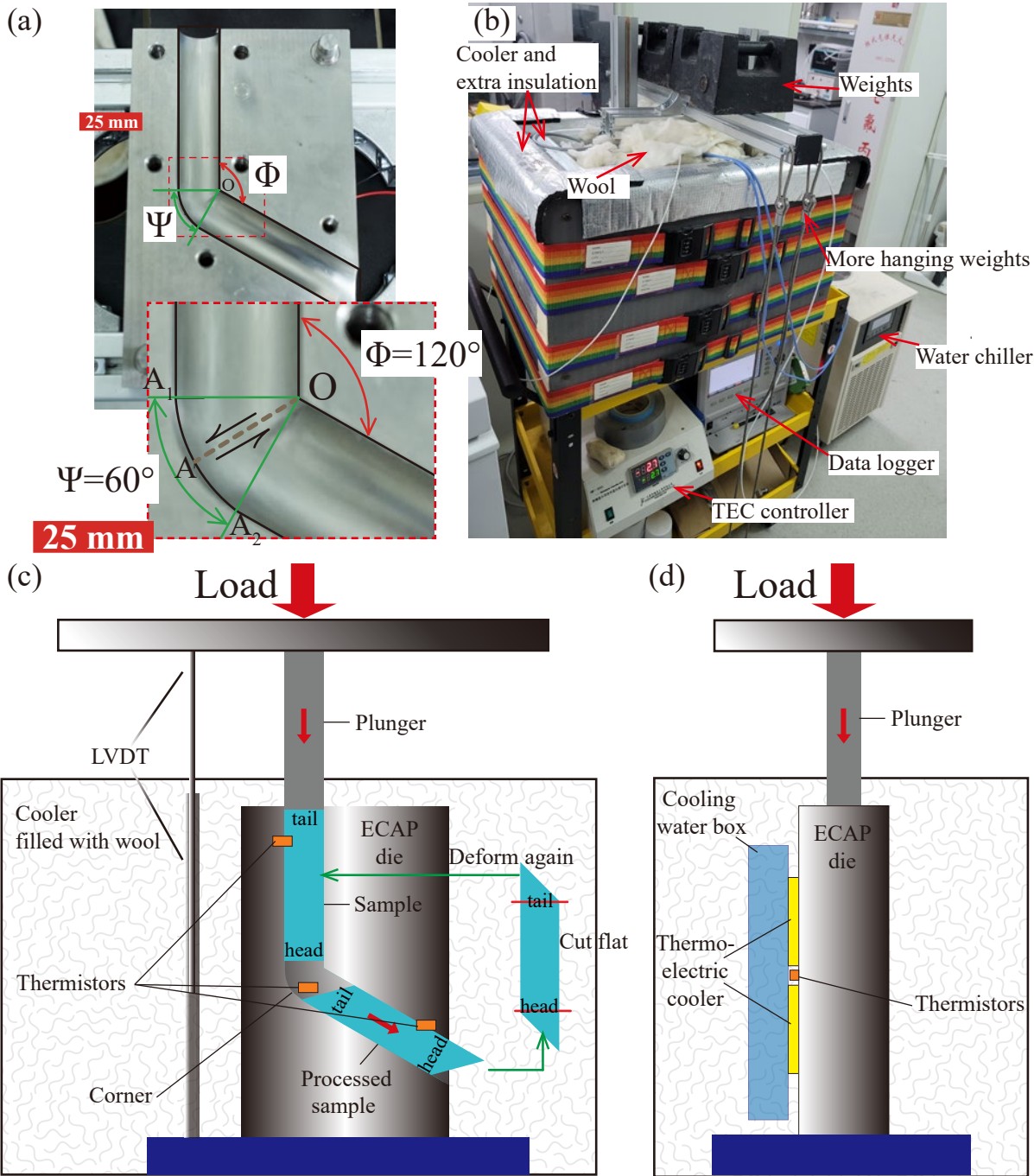

**Figure 1.** Photos and drawings of the ECAP apparatus. (a) Photos focusing on the channel of the ECAP die. In the top photo, the channel angle, $\Phi$, and the curvature angle, $\Psi$, are illustrated. The bottom image is a zoomed-in view of the red box in the top image, providing a close-up view of the shear plane and shear direction. (b) Photo of the apparatus running an experiment. TEC: thermoelectric coolers. (c) and (d) Drawings of the ECAP apparatus from front and side views. Thermistors are attached to the die on the outer surface. The plunger cannot bend at the corner, such that a spare piece of ice is added to the tail of sample to push the sample through the corner. Then the spare piece is removed. All passes are done without changing the orientation of the sample. LVDT: linear variable displacement transducer.

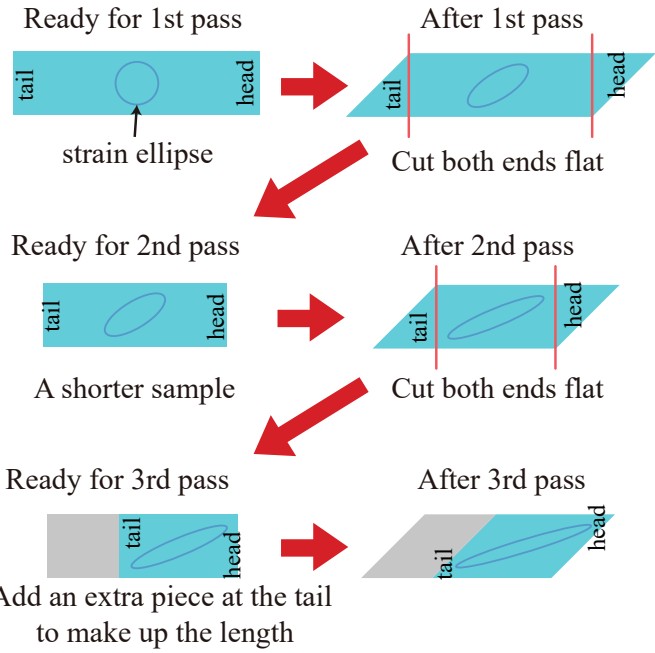

**Figure 2.** Drawings illustrating the accumulation of shear strain in a sample deformed from the first to the third pass in side view. For the preparation of each pass, both ends of the sample must be cut flat and perpendicular to the cylindrical axis. For subsequent passes, the process is repeated. It is important to note that the addition of an extra section is not exclusive to the third pass; rather, an additional section is added to the sample to make up the length, once the sample length falls below 50 mm. The circle in the sample is a strain ellipse, describing the theoretical strain accumulated by each pass.

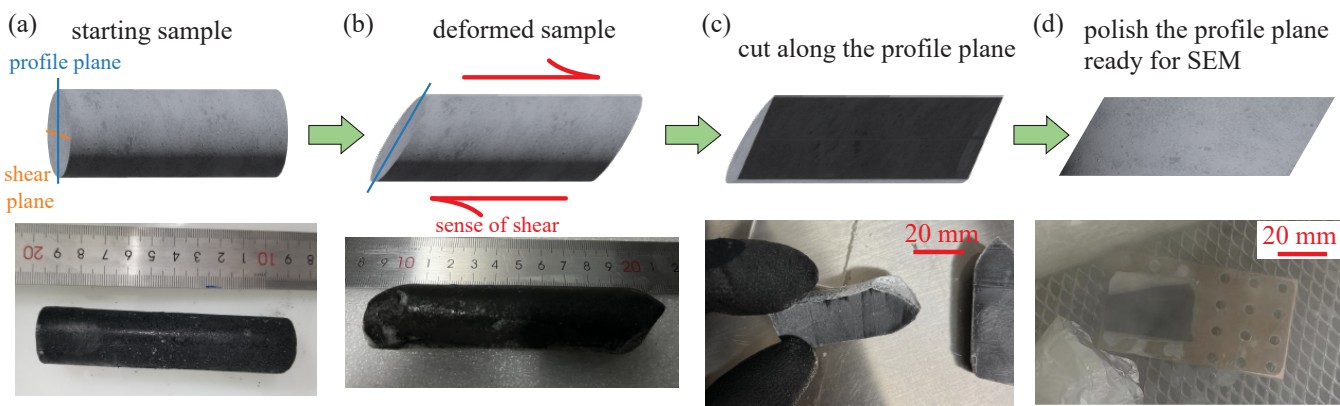

**Figure 3.** Illustration for a sample deformed and prepared for microstructural analysis. (a) Starting sample before deformation. (b) Sample deformed by ECAP. (c) Sample cut along profile plane. (d) Sample polished and mounted on a copper ingot for analysis. Note that the black color stems from graphite powder added to the ice samples. SEM: Scanning Electron Microscope.

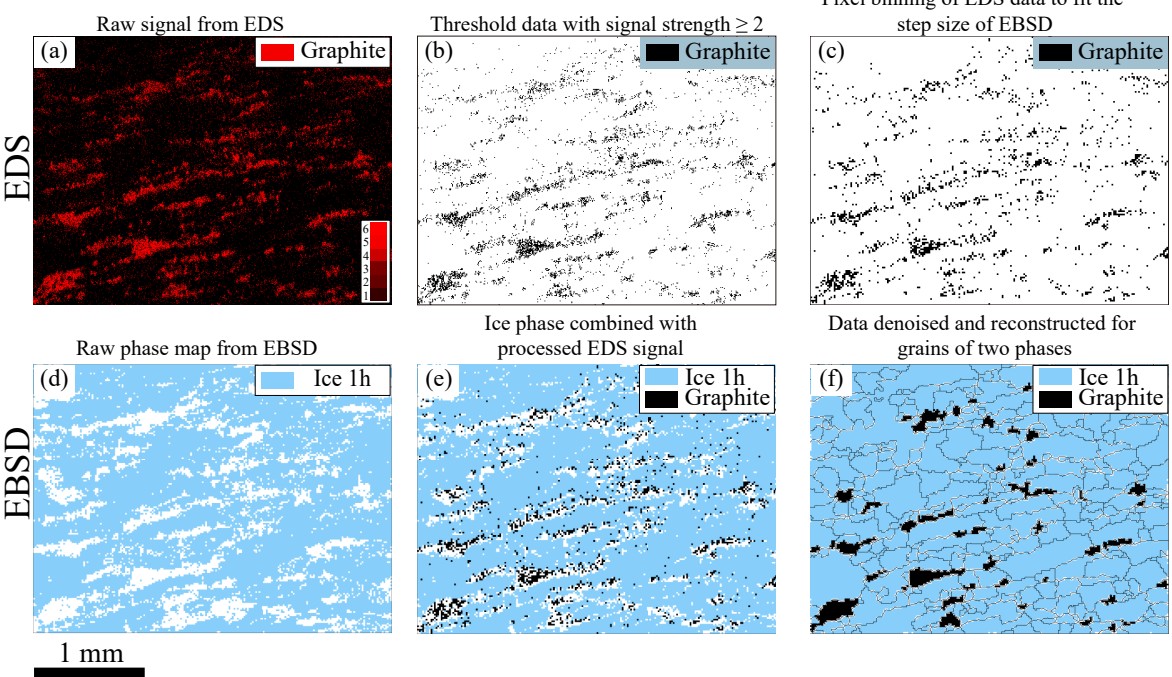

**Figure 4.** Illustration for the combination of EDS and EBSD data to locate the graphite phase. The example shown here is sample ECAP_38_6P. (a) Raw data for carbon obtained from EDS. (b) Data with signal strength $\geqslant$ 2. (c) Pixel binning process so that the pixel size is the same as the 15-μm step size of the EBSD data. (e) Created "EBSD" data of graphite, informed by the graphite coordinates in (c), and combined with the ice phase in (d). (f) Denoised data with grain boundaries tracked by the MTEX toolbox. Note that by this method, the identified graphite phase represents the upper limit of the area fraction of the graphite.

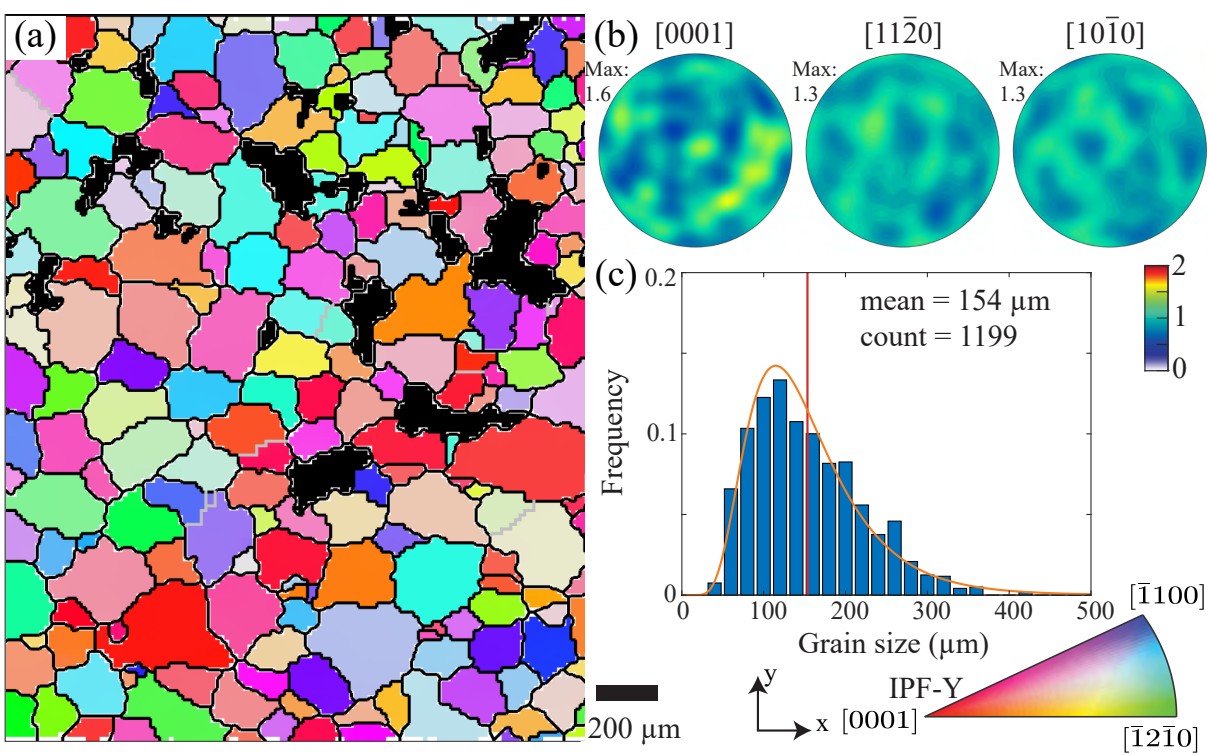

**Figure 5.** Microstructural analyses of an undeformed ice sample. (a) The orientation map is colored by IPF-Y, which uses the color map to indicate the specific crystallographic axis that is parallel to the y-axis. The step size is 15 μm. Grain boundaries, characterized by a misorientation of $\geqslant 10°$, are in black, and sub-grain boundaries, characterized by a misorientation of $\geqslant 2°$, are in gray. Graphite is in black. (b) Stereonets for distributions of [0001], [11$\bar{2}$0] and [10$\bar{1}$0] axes. Data are based on all orientation data and colored by multiples of uniform distribution (MUD), as shown in the color bar. All stereonets are equal-area lower-hemisphere projections. (c) Histogram for the grain-size distribution. The red line marks the arithmetic mean, and the orange curve is a log-normal fit for the distribution. The grain-size data are calculated from a larger area (approximately four times the area of panel (a)), 1199 grains in the entire map.

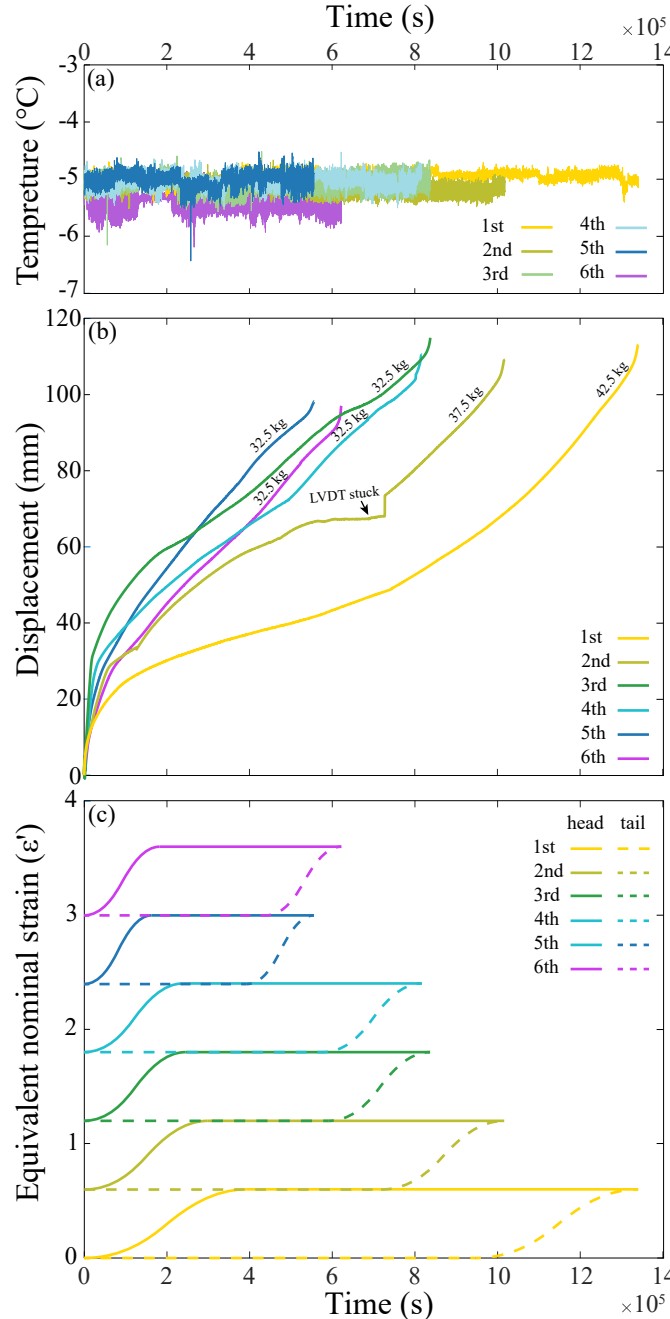

**Figure 6.** Plots of (a) temperature, (b) displacement and (c) nominal equivalent strain versus time for the 6 passes of sample ECAP_38. (a) The temperature is measured on the outer surface of the die, right at the corner. (b) Warmer to cooler colors mark the 1st to the 6th pass, respectively. The load for each pass is noted next to the curve. (c) Equivalent strain is estimated based on a finite element analysis of ECAP with $\Phi = 120°$ (Wei et al., 2009).

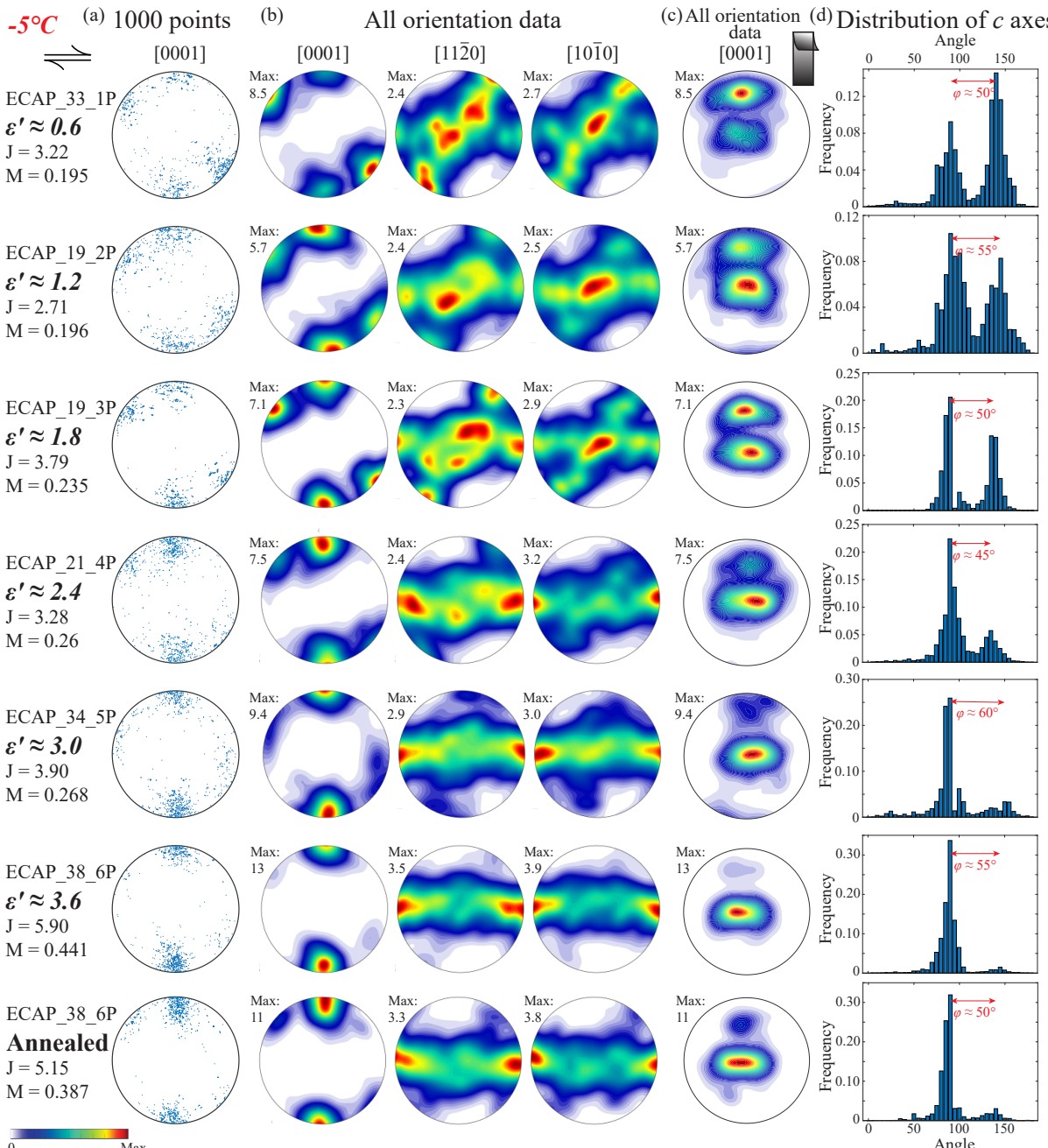

**Figure 7.** Analysis of crystallographic orientations on the profile plane of all deformed samples. The shear direction is top to the right, as illustrated by the black arrows. Step size is 30 μm. J- and M-indexes are calculated based on all orientation data. (a) Distributions of orientations of [0001] axes from 1000 randomly selected points. (b) Distributions of orientations of [0001], [11$\bar{2}$0] and [10$\bar{1}$0] axes contoured on the basis of all orientation data. The contours on the stereonets are colored by MUD, values of which range from 0 to the maximum value indicated on top left of each stereonet. (c) Distributions of orientations of [0001] axes. The shear plane rotated to be parallel to the paper, and the shear direction is topside up. (d) Distributions of the [0001] axes on the great circle normal to the profile plane. The angle between the two clusters, $\varphi$, is presented on each histogram. For sample ECAP_38_6p, $\varphi$ is measured by taking the very weak, disappearing secondary cluster into consideration.

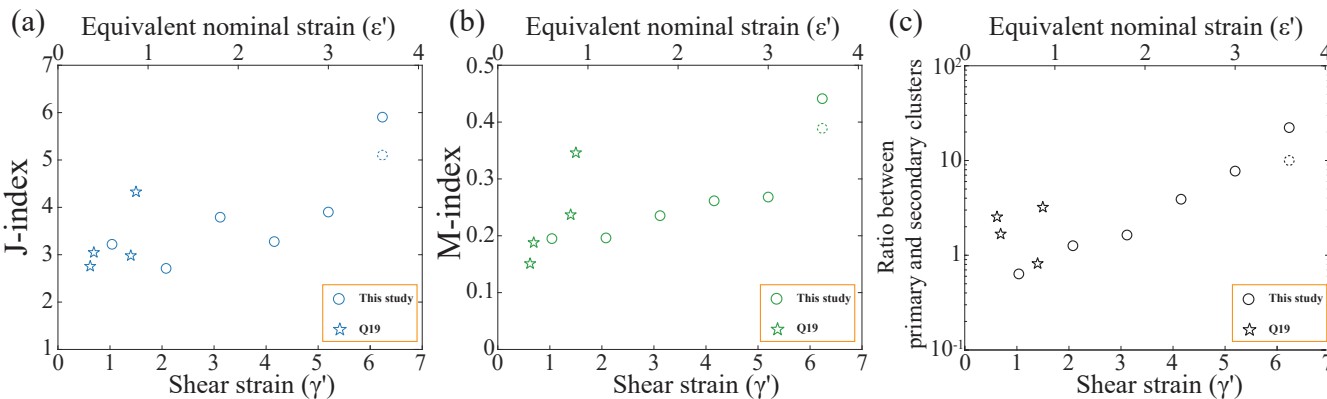

**Figure 8.** Crystallographic fabric strength as a function of strain. Fabric strength is quantified using (a) the J-index, (b) the M-index and (c) the ratio of the number of orientations in the peak of primary $c$-axis cluster over those of secondary cluster. Both values of shear strain and nominal equivalent strain are presented as x-axes. Circles are data from this study, and stars are the $-5°C$ data from Qi et al. (2019). Circle with dashed line represents the annealed sample. The number of orientations in the peaks of the clusters are calculated from Figure 7(c).

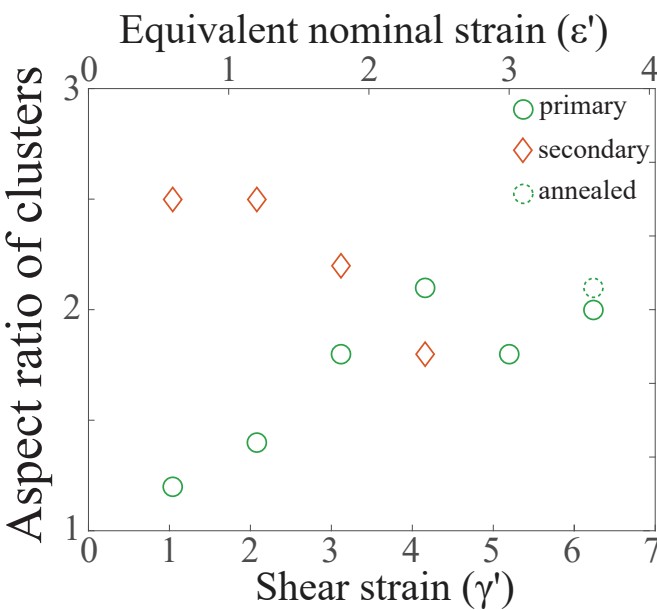

**Figure 9.** Aspect ratio of clusters as a function of strain. Both values of shear strain and nominal equivalent strain are presented as x-axes. Circles are for primary clusters, and diamonds are for secondary clusters. In the samples deformed to 5 and 6 passes, the secondary clusters become irregular and/or very weak, so that the aspect ratios were not calculated. The circle with dashed lines represents the annealed sample.

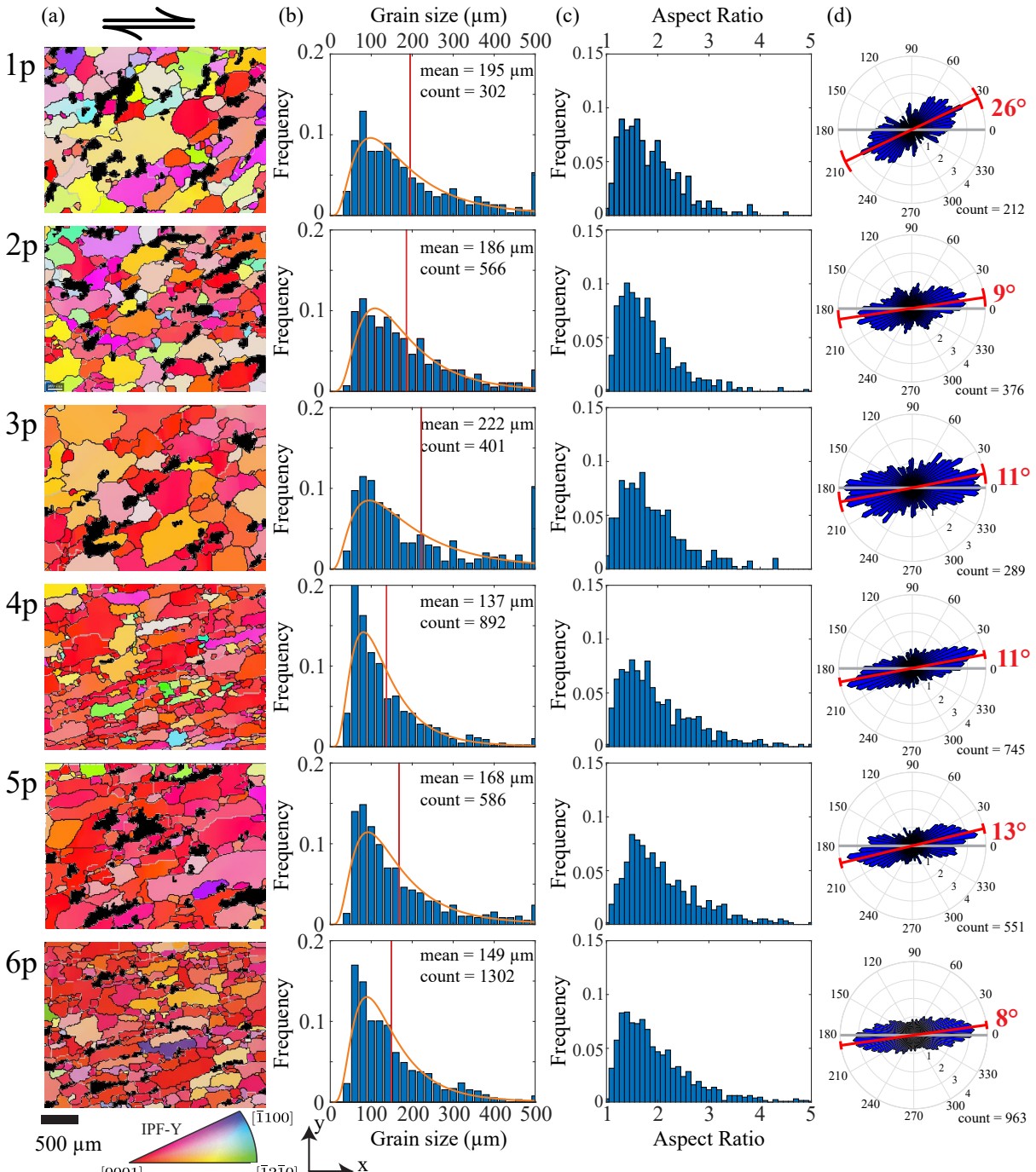

**Figure 10.** Microstructure results for all deformed samples. (a) Orientation maps colored by the crystallographic orientation normal to the shear plane. Graphite phase detected by EDS is black. From top to bottom are samples after 1 to 6 passes. The color map is the same as in Figure 5. In each map, grain boundaries, characterized by a misorientation of $\geqslant 10°$, are black, and subgrain boundaries, characterized by a misorientation of $< 10°$ and $\geqslant 2°$, are gray. Unindexed spots are white. Maps are sub-areas of the whole mapped area. (b) Histograms for grain-size distributions. In each histogram, the red line marks the arithmetic mean, and the orange curve is a log-normal fit for the distribution. (c) Histograms for distributions of aspect ratios of grains. (d) Rose diagrams presenting shape preferred orientations. The gray line represents the shear plane, while the red line and associated numbers indicate the average angles of the SPO, as calculated from the rose diagram. Only grains with an aspect ratio $> 1.5$ are counted in these diagrams.

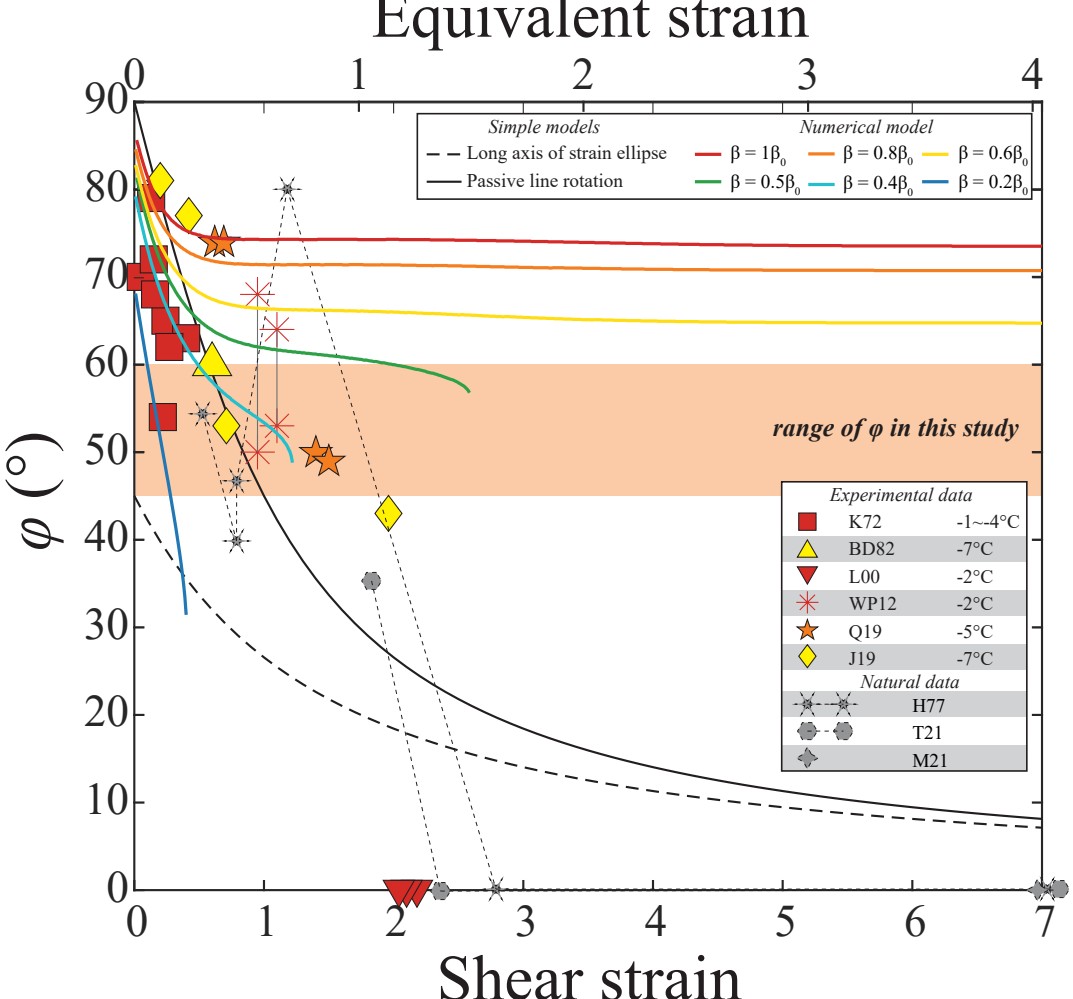

**Figure 11.** Plot of the angle between c-axis clusters, $\varphi$, for experiments from this study, experiments from the literature, results of simple models and results of the numerical model. Since the nominal shear strains in this study cannot be directly compared with those in previous experimental studies, a range of $\varphi$ is marked by a shaded box in the plot, independent of strain. Experimental data are from the following studies: K72: Kamb (1972); BD82: Bouchez and Duval (1982); L00: Li et al. (2000); WP12: Wilson and Peternell (2012); Q19: Qi et al. (2019); J19: Journaux et al. (2019). Only experimental data at $-5\pm4°C$ were illustrated here. Symbol colors broadly indicate deformation temperature, with red, orange and yellow colors indicating $> -5°C$, $-5°C$ and $< -5°C$, respectively. Data from natural ice are as follows: H77: Hudleston (1977); T21: Thomas et al. (2021); M21: Monz et al. (2021). Symbols tied by lines indicate they are from one sample. The maximum strains for the natural samples were estimated to be larger than the scale of our experiments, and thus, three markers (M21, H77 and T21) were placed at the right end of the x-axis suggesting that their shear strains are larger than 7. Outcomes from the numerical model based on Richards et al. (2021) are marked by colored thick lines. The parameter, $\beta$, represents the rate of migration recrystallization in the model. The termination of these curves suggests that a single-cluster fabric forms.

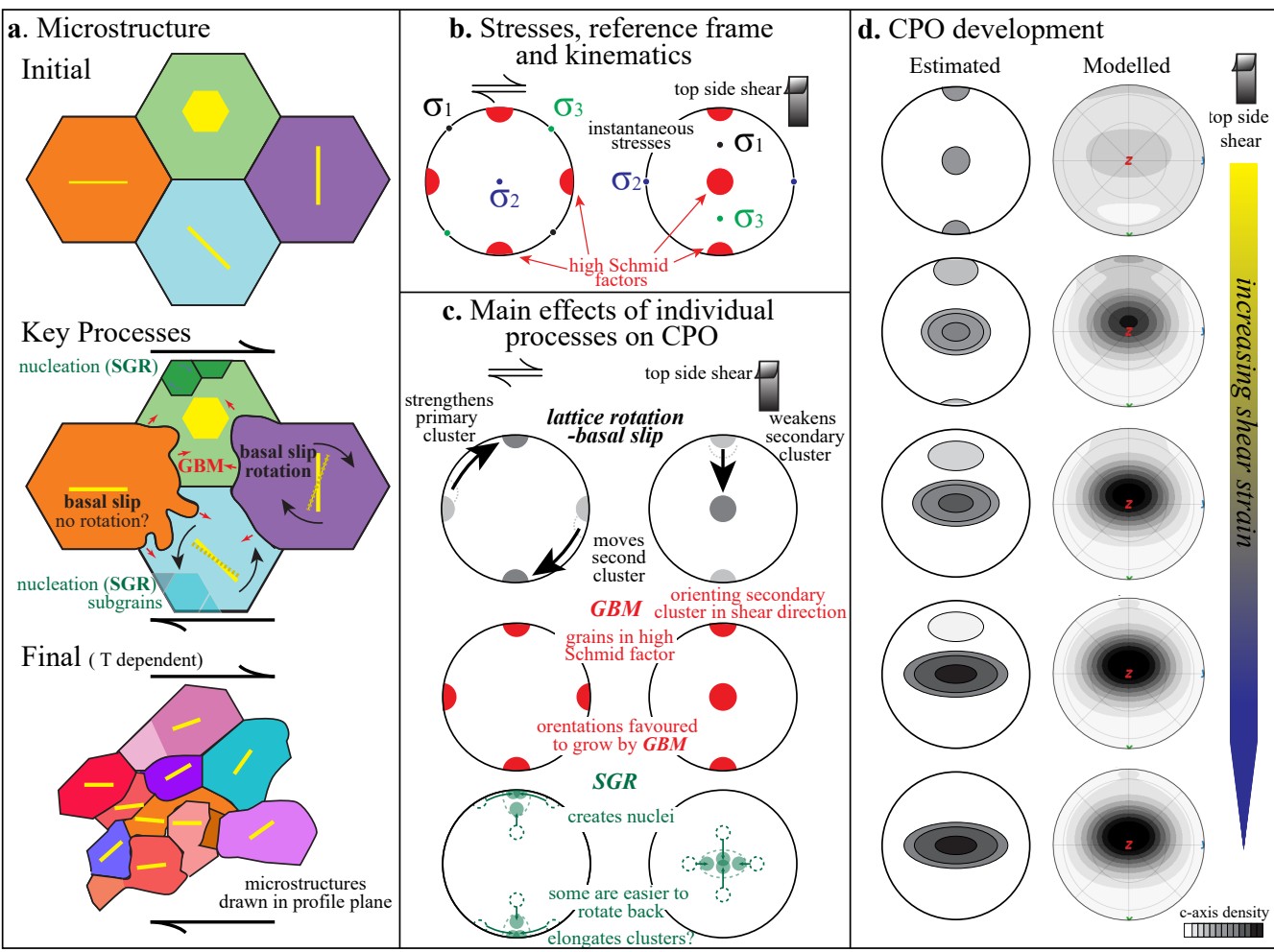

**Figure 12.** Schematic drawing for the development of CPOs in ice sheared in the laboratory, adapted from Qi et al. (2019). (a) The evolution of microstructure represented by four hexagonal grains with different initial orientation of basal planes. (b) Stress and kinematics at different reference frames. On the left, shear is top to the right; on the right, shear is to side up. Same for the next two panels. $\sigma_1$ and $\sigma_3$ are the maximum and minimum deviatoric stresses (compressive positive), respectively. (c) The main effects of CPO-formation mechanisms. In the SGR panel, longer arrows suggest faster rotation, and shorter suggest slower. (d) The development of CPOs with increasing strain. SGR: subgrain rotation. GBM: grain boundary migration.