# Peer review of "Evolution of crystallographic preferred orientations of ice sheared to high strains by equal-channel angular pressing"

_EGUsphere, 2024_

## Referee Comment (RC1)

**Review of *'Evolution of crystallographic preferred orientations of ice sheared to high strains by equal-channel angular pressing'**

**TC manuscript egusphere-2024-331**

**Summary**

This study adapts the method of equal-channel angular pressing (ECAP), previously used primarily to study the plastic deformation of metals and alloys, to understand the development of crystal-preferred orientation (CPO) in ice experiencing high strains. The authors generate artificial polycrystalline ice and perform deformation tests at -5 °C. Repeated deformation of the same ice samples allows them to reach higher shear strains, resulting in six analysis samples with equivalent strains ranging from 0.6 to 3.6 in 0.6 increments.

They analyze the samples with cryo-electron backscatter diffraction to obtain c-axes distributions, grain size, and shape. They find that all samples result in two clusters of c-axes, and while one strengthens with increasing strain, the other seems to disappear. At the same time, the angle between the clusters remains unaffected by strain. Based on these observations and a fabric evolution model, the authors suggest that the combined effects of lattice rotation due to dislocation slip and grain growth by strain-induced grain boundary migration can explain the CPO patterns. They conclude that grain-boundary migration becomes progressively less important with increasing strain and that single-cluster fabrics can be generated with high-strain experiments.

I believe this paper is a valuable and relevant contribution to the current understanding of CPO development in ice and other anisotropic materials on Earth and in the Solar System. The manuscript is of good quality and lies well within the scope of The Cryosphere. The text is well written and nicely illustrated with comprehensive figures and photos documenting the experimental process. Although I am not an expert in deformation experiments, I find the methods used sound and well explained for the general reader of The Cryosphere. I, therefore, recommend this article for publication with minor revisions.

This review is structured into general, specific line-by-line, and technical comments. Suggested text edits or quotes are marked in blue, and changes are tracked with strikethrough and italic font.
* * *
**General Comments**

**Structure & Text Fluidity:**

In general, the text is well written and well-supported by references. However, it seems as if different sections were written by different authors, resulting in some repetition and a lack of logical order. Long, complex sentences could be simplified and broken down to improve readability, particularly in

Sections 1, 2 and 4. The introduction could benefit from a more structured flow, guiding the reader through the background, problem statement, and research significance in a clearer sequence. Some sections could be more specific about the gaps in current research and how this study addresses them. The discussion could be condensed and sentences shortened/simplified. Additionally, it would be helpful to clearly distinguish between the results and conclusions of this study and what is known from previous work.

The structure of the manuscript in sections and subsections seems logical. However, in some areas, these sections mix together. For example, lines 62-70 in the introduction describe the ECAP method in detail, including a reference to Figure 1, which should be part of the methods (e.g., before section 2.1).

Lines 156-183 are difficult to read and repetitive. For example, the orientation data is mentioned three times as being analyzed with the MTEX toolbox: lines 158, 171, and 174. It is also described twice how individual graphite particles can't be detected with EDS: lines 168 and 170. Shortening and simplifying this section would improve readability. The text also jumps between analysis methods and datasets, which is confusing.

See my specific comments for suggested edits at specific lines.

**Processes Affecting CPO Development:**

In the introduction, I would have liked to see a paragraph focusing on the individual processes affecting CPO and under what conditions they are relevant. You describe (lines 39-41) that "the dominant mechanism for CPO formation changes from grain boundary migration (GBM) to lattice rotation and subgrain rotation (polygonization), with increasing stress, increasing strain or decreasing temperature (Qi et al., 2019; Fan et al., 2020)." However, you do not explain how these processes work. This is somewhat done in Section 4.7 (lines 405-408) and later in that section, as well as in Fig. 12a, but it would be very helpful to have an overview of these processes already in the introduction.

**SpecCAF Model:**

It is unclear what role the modeling actually plays here. In the methods, you spend some time explaining the SpecCAF model and how you "re-used" the model by Richards et al. (2021), but lack details of how exactly that modeling was done (i.e., parameterization, initial conditions, or what values of $\beta$ you used, etc.). This makes it seem as if you did simulations in the context of this study. However, no modeling results are shown or mentioned anywhere in the results. The only place modeling results are mentioned is in the discussion, and in Fig. 11 where "outcomes from a recently published numerical model by Richards et al. (2021) are plotted." You should be more clear about what you did in the context of this study and provide the details necessary to reproduce your results, and what is a result by Richards et al. (2021). For modeling done in the frame of this manuscript, its outcome should be shown before the discussion.

**Implications:**

In the introduction, you outline the broad relevance of CPO development for ice and mantle material on Earth as well as on other planetary bodies. However, the discussion and conclusion sections lack detail on how your work relates to these broader implications and in where this strong shear deformation at

relatively warm temperatures used in your experiments is likely to occur in nature. For example, what are the implications of this study for CPO developments in shear margins of ice streams? It would also be helpful to have a brief discussion on limitations of this kind of experiment in representing conditions found in natural ice deformation (e.g. different timescales).

**Axes Definition:**

Somewhere in the manuscript, you should define a-axes, m-planes, and c-axes with their corresponding Miller indices. It is somewhat defined in line 250, but this is rather late. c-axes are defined in the introduction, so it would make sense for the other two being described there as well.
* * *
**Specific comments**

line 4:  To examine the impact of strain *on the relative importance of these two mechanisms,* might be more clear.

line 20:  *water* ice or *Ice Ih*.

line 20:  '*the most common compound in the universe*' needs citation. Or leave out, I think this is redundant

line 21:  satellites, (comma)

line 25-26:  I find this sentence wordy and complex. Consider breaking it down to improve readability.

line 27:  than on other slip planes (Duval et al., 1983); 0001 is defined as c-axis in line 46?

line 28:  'to ice' is unnecessary and can be omitted.

line 30:  Omit 'subsequent'. Not clear what 'response' refers to in this sentence. I suggest rephrasing to something like:

  *When the stress field driving ice flow changes, the rate of ice deformation depends on the existing CPO and its evolution under the new stress configuration (Hudleston, 2015).*

line 31:  perhaps add gravitational forcing to the list?

line 34:  For CPOs in ice sheets on Earth, radio-echo sounding (RES) is another widely used technique to derive CPOs. You could consider citing e.g.:

  https://tc.copernicus.org/articles/16/1719/2022/
  https://ieeexplore.ieee.org/document/8755860
  https://tc.copernicus.org/articles/17/1097/2023/

| line 37-39: | What kind of 'transition' have been observed? change in the dominant mechanism for CPO deformation between different experiments? Or with time/increasing strain? Needs to be clarified. |
|---|---|
| line 42-45: | long sentence, consider breaking it down: |

*While uni-axial compression tests are commonly used to study ice microstructural evolution during deformation, high-strain deformation in ice sheets and glaciers is mainly simple shear (Cuffey and Paterson, 2010). Therefore, understanding CPO evolution under shear is critical.*

| line 47-49: | sentence seems a bit wordy. Suggestion: |
|---|---|

*The highest shear strain achieved in the lab, 2.6, was reported by Qi et al. (2019) at −30°C. Previous experiments at similar temperatures did not exceed shear strains of 0.12 (Wilson and Peternell, 2012).*

| line 50: | ... cluster. Instead ... (break sentence here). |
|---|---|
| line 50: | Not clear what gamma refers to. Please clarify. |
| line 49-52: | I think this part is a bit confusing. I would suggest first stating that natural ice samples often experience shear strains larger than 5, which leads to the development of a single cluster of [0001] axes. Then go on and explain that this could so far not be reproduced by lab experiments, which even under strains larger than 2 fail to produce such a strong single maximum fabric. |
| line 53: | single, primary cluster *evolving* at smaller... |
| line 55: | omit '*simply*' |
| line 56: | at higher strains |
| line 57: | are needed. |
| line 59: | move the alternative definitition one line up: *Equal-channel angular pressing (ECAP; also known as equal-channel angular extrusion) is a technique for generating severe plastic deformation, resulting in highly strained microstructures with ultra-fine grain sizes and strong fabric. ECAP was initially developed in the 1980s ...* |
| line 63: | intersect at an angle $\Phi$ |
| line 79: | it would be nice to have one sentence summarizing the most relevant findings and relevance of this study at the end of the introduction. |
| Fig. 1: | - panel a): arrows are hard to see. Consider a zoom window with a close-up of the shear plane. Photos could be arranged so the load arrow in panel c is not cut off. |
| | - panel b): TEC is not defined in caption. |

- panel c): LVDT is not defined. Processed sample (instead of Sample processed?)

- caption: Photos and dreawings *of* the ECAP apparatus.

- caption: both parts of *the* die are shown with a dummy sample in the channel (omit comma).

- caption: define abbreviations.

- caption: 'all passes are done without changing the orientation of the sample': how do you ensure the orientation is the same during the second deformation?

- panel c, d and caption: Thermistor misspelled (thermister)

line 85:    change 'has' to 'have'

line 86:    'for more details please refer to Table 1' or 'more details can be found in Table 1'.

Table 1:    I think it would be helpful to describe parameters $\varepsilon'$ and $\varphi$ in the caption.

line 90:    This first sentence is not very informative. Remove and change the second sentence to:

   *'The ECAP die consists of two symmmetrical stainless-steel parts, as shown in Fig. 1a. '*

line 92:    The diameters of the channel and ice samples is 25 mm .

line 92-93:    perhaps use dash for 'mirror-finished'. The quotation marks seem strange here, I would simply remove it. Is the soap coating refreshed between experiments? Please clarify.

line 93:    The channel *geometry* is defined by two angles

line 94:    remove '(equal to $120^o$ in Figure 1(a))' . Instead add '(see Fig. 1a)' at the end of the sentence.

Eq. (1) :    I am confused on why strain here is expressed as $\varepsilon$ and as $\varepsilon'$ in Table 1. I also think it is strange to use '=' for the definition of $\Phi$ and '$\approx$' for the definition of $\psi$. Can the angle $\psi$ be determined less accurately or why is that?  In addition to this equation, it might be helpful to state the relationship of the equivalent strain and the shear strain.

line 112-113:    Once  the sample temperature had equilibrated with the die *(typically after 20 min),* a load was applied to the sample using dead weights hung on the aluminum profile

line 115-117:    why is that? In some cases (ECAP_19, 21) the load of the second pass decreases by 5kg, while for others (ECAP_34, 38) it decreases by 10 kg and being stable for further passes. Please clarify.

line 116: for *a* single sample; (*for* more details

line 120-123: I think this sentence should be shortened/broken up and simplified.

Fig. 2: - caption: Drawings  *illustrating* the accumulation of shear straine in a sample deformed from the i-th  to the (i+2)-th pass *in side view*.

- general: In the case illustrated i=1, right? I am unsure of the generalization with *'i'* since the strain ellipse suggests it has not been deformed previously before pass *i*. I would suggest changing it to first, second, third pass and adding a comment in the bottom of the figure or the caption, stating that for further passes step 3 is repeated with adding additional ice to compensate the length.

line 124-133: This section sounds in parts as if it is a result, rather than experiment set-up or method. For example (line 125) ' indicating shearing' seems redundant when you are describing a shear deformation experiment. See my suggested edits below:

Upon passing through the channel, the sample's diameter  *remains* unchanged, but the head and tail surfaces  *are* no longer perpendicular to the cylindrical axis (see Figures 2 and 3). *In preparation for additional passes* , the head and tail surfaces were trimmed , restoring the sample's  *cylindrical* shape  *with* a reduced length *compared to the original sample* (Figure 2). The sample can be reinserted into the channel with the same orientation relative to the corner and deformed again.  *For* sample*s*  *becoming* shorter than 50 mm, an additional piece *of ice* was added  to achieve the required length *but this extra ice was not considered in the analysis* (Figure 2).  This process allows for multiple passes through the channel, accumulating high strains. After reaching the target number of passes, the *samples*  *were* wrapped in aluminum foil and stored in liquid nitrogen. Note that during an experiment, the head part of the sample  *is* exposed to air, while the tail part  *is* deformed in the corner, causing more sublimation in the head part.  *To avoid sublimation bias, we used the middle part near the tail for microstructural analysis.*

line 126-127: How do you ensure that the orientation is preserved? Is it marked by something? Please clarify.

line 128-129: Where is the additional piece of ice added? From Fig. 2 it looks as if it is added at the tail. Since the head part is more prone to be affected by sublimation, why not add this piece at the head? Maybe it is, but please clarify here in the text and mark head and tail in Fig. 2.

line 134-135: To investigate the effect of annealing on pre-existing CPO one sample was annealed at −3.5∘ C for 24 days *after ECAP deformation*. The annealing experiment was done using a similar apparatus described in Fan et al. (2023).

Can you state which of the samples in Table 1 that is?

Fig. 3:    - panel c & d): It would be good to have an approximate scale on these photos too. Abbreviation SEM in panel d) needs to be defined in caption.

- caption: Illustration for a sample deformed and prepared for microstructural analysis.  (a) Starting sample before deformation. (b) Sample deformed by ECAP. (c) Sample cut along profile plane. (d) Sample polished and mounted on a copper ingot for analysis. Note that the  *black color stems from graphite powder added to the ice samples.*

line 139:    why is sample storage so cold and for how long? Do temperature changes from -5 to -190/-120 to -10 degrees not affect your analysis?

line 140:    I think it would be helpful to be more specific here on how the cut is oriented relative to the shear plane. Alternatively, define what you mean with 'profile plane' and indicate in Fig. 1.

How thick is this 'section'?

line 145-146:

I don't think this sentence adds much information as it stands now.

line 152:    It would be helpful to state the approximate sample size here or somewhere else. Were they all cut to the same size?

line 154:    Subsequently, carbon element data were obtained from EDS  *for the* same *selected* regions with a step size ranging from 3.2 to 6.7 μm. (assuming this is only done for regions with step size of 15 micrometer. Else please clarify)

line 156:    I think the first sentence is redundant. Rather than describing what you didn't do, simply describe what you did.

Orientation data obtained from diffraction data with a step size of 30 μm  were used for analyzing the CPO patterns.

line 157-158:   Here you say that 15 micrometer data is analyzed with MTEX. In line 176 you mention that 30 micrometer data is analyzed with MTEX toolbox. Why not generalize this in one sentence in the beginning of the  paragraph in something like ' We used the MTEX MATLAB toolbox (citations) to process orientation data' or similar.

line 166-167:   To match the dimensions of the EBSD data, the pixel size of *the* EDS map was adjusted to match the step size of *the* EBSD. Then these pixels were attributed to *the* graphite phase in *the* EBSD data.

line 168:      *By combining* the EBSD data from ice and graphite, a data set with two phases was obtained (Figure 4(e)).

line 168-169:  Note that this method cannot identify individual graphite particles, but only  *reveals* graphite-rich regions. (mixed tense)

line 169-170:  Repeat of previous sentence.

line 171-172:  Not very clear what you're doing here. Are you really 'reconstructing grains? Do you mean you locate grain boundaries by looking for orientation changes between pixels of more than 10 degrees? Please clarify.

line 177-179:  consider splitting this sentence in two.

line 181-183:  We adopted the same method *for cluster identification* used previously in Qi et al. (2019)*, where t*he normalized counts of data per orientation in the profile plane were plotted on a histogram. φ was defined as  *the angular width* between the two peaks on the histogram.

line 183:     reference to Fig. 7d?

line 185:      *The observed* CPOs were compared to predictions from the *spectral continuum anisotropic fabric evolution (*SpecCAF*)* model (Richards et al.*,* 2021).

line 186:     The SpecCAF model cannot directly simulate microstructural *changes*  *like* other models, e.g., ELLE, (Jessell et al., 2001).

line 185-189:  I think these section is confusing and should be clarified by being more specific of how 'microstructural processes' are simulated differently between these models. 'The evolution of CPO' which 'SpecCAF simulates' could also be interpreted as 'simulating microstructural changes'.
              It could also be mentioned that in contrast to more complex models, such as ELLE, SpecCAF is computationally more effective and more suitable for large-scale CPO modeling.

line 192:     The numerical model was  *adapted from* Richards et al. (2021)*,* producing simulated pole figures *representing the distribution of c-axis orientation,* and the angle, φ*, describing the angular distance between clusters.*

line 193:     'φ was compared to model predictions with a variety of β = kβ0 values': I don't understand this. Do you mean experimentally observed φ were compared to modelled? Or do you mean φ was tuned to observations by adjusting β? Either way, needs clarification.

line 195:     How many, and in what increments?

Fig. 4:      - title of panel c): Combine*d* pixels - combined pixels of what? panel (a) and (b)?

- caption: The caption reads in part as instructions, rather than a description of what is shown. I find the description of panel e) especially confusing ((e) Knowing the coordinates of graphite from (c), create "EBSD" data of graphite. Combine the ice phase and the graphite phase.) Suggested edits:

The example  *shown* here is sample ECAP_38_6P.

(c) Combine*d* pixels so that the pixel size is the same as the *15 μm* step size of *the* EBSD data.

(e)  Create*d* "EBSD" data of graphite, *informed by the graphite coordinates in (c), and combined with*  the ice phase *in (d)*.

(f) Denoise*d*  data with grain boundaries tracked by the MTEX toolbox.  Note that by this method, the identified graphite phase represents the upper limit of the area fraction of the graphite.

line 198:     The ice microstructure of the starting materials was similar in character to that described in Qi et al., (2017).

Fig. 5:        - panel labels would be helpful.

- colorbar: I would suggest adjusting the colorbar for the stereoplots to range from 0 to 1.6 and add at least one more number between to indicate that it is linear (e.g. 0,0.8,1.6)

- it might be helpful to indicate that [0001] is the c-axis, and [1120] and [1010] are axes in the basal plane of the crystal here.

- the subgrain boundaries look almost white and are hard to see - use a darker gray for better visibility.

- labels of IPF-Y colorscale are very small

- y-axis in caption is undefined. Is it vertical to the sample cut? Please clarify.

- caption:  Microstructural analyses of an undeformed ice  *sample*.

'The grain-size data are calculated from a larger area' - larger than what? how large, and is it the same size for all samples?

',  791 grains *in this sample*.'

line 201:      (better say how much smaller)

line 208:     I think you mean the temperature is lower by 0.5°C. Better state -5.5°C to be more clear.

line 214-215:  I don't think this is true for ECAP_21 and ECAP_34, while we don't know for ECAP_33. Given that this is only true for 2 out of 5 samples, can you really say that the first pass is 'generally much slower' ?

line 215-227:  This section could be shortened and streamlined. Line 215-220 seems more relevant for the discussion, rather than results

Fig. 6:  - panel a): increase line thickness in the legend - it is hard to see the colors in such thin lines. A very minor detail: why are the colors in panel (a) in pastel/different from (b) and (c) when denoting the same thing? (same comments for Fig. A1).

- I would adjust the y-axis to min/max achieved temperatures to increase visibility in panel (a)

- green and red lines might be hard to be distinguished by colorblind people.

- panel (c), y-label: *Equivalent nominal strain, ε'*

Fig. 7:  - caption: The contours on the stereonets are colored by MUD, values of which  range from 0 to the maximum value indicated on top left of each stereonet.

- panel c): I think the maximum value of the colormap should also be stated here. Please state in the caption how the shear direction is now (from top to bottom? or left to right?). It seems as this has been tried to be indicated by the gray figure on top of panel c, but should be complemented by a unambiguous description.

Fig. 8:  - panel labels are missing.

- a legend in each panel would be helpful to see what the plots show faster.

Fig. 9:  - why not add the annealed sample to the legend too?

- additional math symbol in x-label would help for faster comparison with  e.g. fig. 7, Table 1 and text.

line 257-262:  I'm not sure this belongs to 'results'. Since you already have a section in the discussion (4.2) dedicated to the effect of graphite, I would suggest moving this to 4.2.

line 264:  more *frequent* in the samples

line 266:  deformed  *by* 1-3 passes

line 267:  deformed  *by* 4-6 passes

line 270:  The distribution of *grain* aspect ratios (I assume?  Should be clearly distinguished from cluster aspect ratio)

line 270:      deformed  *by* 2-6 passes  (as above. Notation doesn't matter, but should be consistent).

line 278:      multi-pass *samples*

line 280-282:  This overview of the discussion sections is very helpful. I suggest using section names to be more specific. I think you should also mention what section 4.3 and 4.7 is about

line 298-299:  I think this sentence is redundant:

line 310:      I believe en dashes are used for ranges

line 321-324:  repetitive

line 335-340:  I suggest mentioning the names and location of the glaciers for natural samples instead of 'a  glacier'. For example: Thomas et al. (2021) found a roughly round  c-axis cluster in the shear margin  *Priestly G*lacier *in Antarctica*.

line 357:

line 359:      two recent studies

line 362-363:  It is worth noting that under the same strain, the strength of the CPO in the sample deformed by ECAP is weaker than that in samples deformed continuously *due to effects of annealing discussed in Section 4.1.*

line 369:      in *Qi et al. (2019)*

line 375:      laboratory samples *from this study*.

line 385:      is in good agreement with laboratory data *from this study,*

line 385-386:  I'm not sure I understand this second part of the sentence. Do you mean:  *and is possibly because the secondary cluster weakens with increasing strain, rather than moving towards the primary cluster.* ?

line 390:      define FFT.

line 391-393:  The SpecCAF model, incorporating recrystallization, lattice rotation and grain rotation processes, yielded excellent quantitative agreement in the CPOs  with experimental observations  (Richards et al., 2021).

line395-396:  *see* Richards et al. ( 2021))

line 408:       have to deform through slip

line 408-409:  Is this really your hypothesis? In the introduction you state something similar in line

26-28: 'When deformed by dislocation glide, a single crystal of ice is several orders of magnitude weaker for slip on the basal plane, (0001), than on others (Duval et al., 1983).

need to be clear what is your results or hypothesis and what is known from earlier studies.

line 411: remove double parenthesis around Figure 12(c); check hyphenation in higher-Schmid and lower-Schmid factors

line 429-430: awkward sentence

line 435:  *Consequently,* ?

line 430: secondary *clusters*

line 461: We thank  Prof. Jianhua Rao for his help with designing the ECAP die.

Fig. 10: -panel a): I suggest adding the IPF-Y colorbar here as well, else readers have to jump between this Figure and Fig. 5. I would also indicate the shear direction on these figures

- do you have an explanation for why grain size increases in 3p compared to undeformed?

- panel d): legend for red and gray lines?

Fig. 11: - caption: unclear what 'simple models' and 'numerical models' are. What model has been used (reference). Are these both results done in this study or obtaine previously? It is also not clear if 'Model from R21' is a result you obtained in this study or if it was obtained by Richards et al., 2021. Please clarify.

- 'Since the nominal shear strains in this study cannot be directly compared with those in previous experimental studies, a range of φ is marked by a shaded box in the plot, independent of strain.' - I don't understand how this is more helpful than simply plotting your datapoints, and discuss why they differ from e.g. models in the text.

- 'Data *points* at  nominal equivalent strain of 3.6 can be treated as a single c-axis cluster, but a disappearing secondary cluster can still be identified, which gives a value of φ, marked by a shaded marker' - I don't understand this either. Where is this shaded marker? Is this still referring to the results from this study?

- Experimental data are from *the following studies* :

- three markers (M21, H77 and T21) were placed at the right end of the x-axis suggesting that their *shear* strains are larger than 7.

- Outcomes from a recent*ly* published numerical model by Richards et al. (2021) are marked by colored thick lines.

- The termination of these curves suggests that a single-cluster fabric forms. - unclear what this means. Are you saying that for lines ending before nominal strain of 4 was reached have already developed a single-cluster? Isn't it strange that e.g. for β=0.5βo two clusters develop, separated by 58 degrees and abruptly turn into a single cluster fabric?

Fig. 12:    - this figure is almost identical to Fig. 10 in Qi et al., (2019) and should be cited as 'adapted from Qi et al.,(2019)' or similar in the caption.

- panel labels are different from other figures. I think The Cryosphere asks for (a) instead of **a.**

- panel a is a very useful overview. In my opinion this could also be a figure for the introduction, explaining these individual processes and how they work. Whether you move this (part of the) figure or reference it in the introduction is up to you.

- panel d): again it is not clear where these model results come from and how they were obtained.

Fig. A1:    - typo in all panels for the temperature panel y-axis
- increase line thickness in legend for temperature plots and adjust y-axis for better visibility.
* * *
**Technical comments**

1. Most citations are lacking a doi.
2. Figure references are usually abbreviated as Fig. 1. Figure is used only in the beginning of sentences (see TC author guidelines https://www.the-cryosphere.net/submission.html)
3. You often use red-green lines in plots - beware colorblindness
4. I think the bullet points in the conclusion are redundant. You can just remove them and write it as continuous text.

---

## Author Comment (AC1)

Dear Reviewer,

We appreciate your helpful comments. Here we present our responses to the comments. Our responses are in black, while your comments are in blue. We will make necessary revisions to address the questions.

**Response to general comments**

**Structure & Text Fluidity:**

In general, the text is well written and well-supported by references. However, it seems as if different sections were written by different authors, resulting in some repetition and a lack of logical order.

Long, complex sentences could be simplified and broken down to improve readability, particularly in Sections 1, 2 and 4. The introduction could benefit from a more structured flow, guiding the reader through the background, problem statement, and research significance in a clearer sequence. Some sections could be more specific about the gaps in current research and how this study addresses them.

The discussion could be condensed and sentences shortened/simplified. Additionally, it would be helpful to clearly distinguish between the results and conclusions of this study and what is known from previous work.

The structure of the manuscript in sections and subsections seems logical. However, in some areas, these sections mix together. For example, lines 62-70 in the introduction describe the ECAP method in detail, including a reference to Figure 1, which should be part of the methods (e.g., before section 2.1).

Lines 156-183 are difficult to read and repetitive. For example, the orientation data is mentioned three times as being analyzed with the MTEX toolbox: lines 158, 171, and 174. It is also described twice how individual graphite particles can't be detected with EDS: lines 168 and 170. Shortening and simplifying this section would improve readability. The text also jumps between analysis methods and datasets, which is confusing.

See my specific comments for suggested edits at specific lines.

We thank the helpful suggestions. We have made some edits in corresponding sections.

Line 58-79:

[revised manuscript text omitted]

We add a Supplementary Figure S2.

"Supplementary Figure S2.

[Figure]

"Supplementary Figure S2. (a) Typical two-cluster distribution of *c* axes on stereonets within the shear plane reference frame. (b) A schematic illustration explaining the method used to quantify the distribution of *c* axes. (c) A histogram of *c* axes plotted in a histogram, illustrating the angle φ between the two clusters of *c* axes."

**Processes Affecting CPO Development:**

In the introduction, I would have liked to see a paragraph focusing on the individual processes affecting CPO and under what conditions they are relevant. You describe (lines 39-41) that "the dominant mechanism for CPO formation changes from grain boundary migration (GBM) to lattice rotation and subgrain rotation (polygonization), with increasing stress, increasing strain or decreasing temperature (Qi et al, 2019; Fan et al, 2020)." However, you do not explain how these processes work. This is somewhat done in Section 4.7 (lines 405-408) and later in that section, as well as in Fig. 12a, but it would be very helpful to have an overview of these processes already in the introduction.

We appreciate the suggestions. We have made the following changes.

Line 37-45: "Many experimental studies focusing on the evolution of CPO in ice have found a transition of the crystallographic fabric with changing stress, temperature and strain (e.g., Qi et al, 2017, 2019; Fan et al, 2020, 2021). The CPO of ice is typically characterized by the alignment of the ⟨0001⟩ axes (*c* axes), normal to the basal planes. The orientation of other crystal axes, such as the ⟨11-20⟩ axes (*a* axes) and the poles to {10-10} planes (poles to *m* planes), could also serve as important kinematic indicators (Schmid and Casey, 1986). This transition is attributed to a change in the dominant mechanism for CPO formation from grain boundary migration (GBM) to lattice rotation and subgrain rotation (polygonization) (Alley, 1992; Qi et al., 2017; Fan et al., 2020). The two mechanisms are also referred to as migration recrystallization and rotation recrystallization (Poirier, 1985, pp. 179–185). GBM is usually driven by the difference in dislocation density of grains on both sides of the grain boundary, causing the grain boundary to migrate from the low-density side to the high-density side, thereby consuming grains with high density of dislocations (Urai et al, 1986). The dislocation density accumulated within a grain is predominantly influenced by its orientation relative to the applied deviatoric stress. Consequently, as grains with lower dislocation densities grow at the expense of those with higher densities, there is a corresponding increase in the proportion of *c* axes oriented along a specific direction. For instance, in compression tests, grains with their basal planes oriented at 45° to the compression axis typically exhibit lower dislocation densities. When GBM dominates CPO formation, *c* axes are observed to present a conical distribution at 45° to the compression axis (Qi et al., 2017; Fan et

al., 2020). Lattice rotation gradually rotates the orientations of slip planes in a grain, to accommodate the bulk deformation. Subgrain rotation produces new grains with orientations similar to those of their parent grains, but with a deviation. So rotation recrystallization leads to a diffused concentration in the *c*-axis distribution (Alley, 1992; Halfpenny et al., 2006). For instance, in uniaxial compression, when rotational recrystallization is the dominant mechanism, the *c* axes tend to form a single cluster parallel to the applied compression axis. Since high-strain deformation in ice sheets and glaciers is dominantly simple shear (Cuffey and Paterson, 2010), understanding the CPO evolution under shear is critical."

Line 405-410: "GBM is typically driven by differences in dislocation density between grains on either side of the grain boundary, which is often associated with strain inhomogeneity across the grains (Urai et al., 1986). Grains with low resolved shear stresses, or Schmid factors, on the basal plane (also referred to as poorly oriented for easy slip), have to deform through slip on non-basal slip systems. For ice, these non-basal-slip dislocations are more difficult to glide (Duval et al., 1983), and there will be more than one interacting slip systems. Therefore, grains with basal planes poorly oriented for easy (basal) slip tend to exhibit higher internal distortion, leading to higher dislocation density (Vaughan et al, 2017)."

**SpecCAF Model:**

It is unclear what role the modeling actually plays here. In the methods, you spend some time explaining the SpecCAF model and how you "re-used" the model by Richards et al (2021), but lack details of how exactly that modeling was done (i.e., parameterization, initial conditions, or what values of β you used, etc.). This makes it seem as if you did simulations in the context of this study. However, no modeling results are shown or mentioned anywhere in the results. The only place modeling results are mentioned is in the discussion, and in Fig. 11 where "outcomes from a recently published numerical model by Richards et al (2021) are plotted." You should be more clear about what you did in the context of this study and provide the details necessary to reproduce your results, and what is a result by Richards et al (2021). For modeling done in the frame of this manuscript, its outcome should be shown before the discussion.

We thank the reviewer for the suggestion. We have added a new subsection in the Results section.

"Section 3.5 Numerical modeling

The SpecCAF model requires deformation, temperature, and initial CPO as inputs. The model was run in simple shear, as the ECAP mostly results in simple shear. The

model was also run at -5°C, same as the experimental condition. The initial condition for CPO was set to isotropic, similar to our initial samples. The model was also run with the parameter β controlling the effect of GBM, reduced to different fractions (k) of the value Richards et al. (2021) found for T = −5°C (β0). The output of the modeling was the predicted angle between the c-axis clusters, φ, plotted alongside the experimental results in Fig. 11. The value of φ decreases with increasing shear strain. At a given strain, the value of φ decreases with decreasing β. At higher values of β (0.6 --- 1β0), φ decreases from over 80° to ~70° as shear strains increases to 1 and stays roughly constant at larger strains. At lower values of β (0.2 --- 0.5β0), the curves of φ terminate at different strains. This termination means the model cannot identify a secondary c-axis cluster, and the CPO is characterized by a single cluster. At β = 0.5β0, φ rapidly decreases from ~80° to around 60° as shear strain increases to 1, and gradually decreases at larger strains until the secondary cluster disappears at a shear strain of ~2.6. At β = 0.4β0 and 0.2β0, φ rapidly decreases to ~50° and ~30°, when the secondary cluster disappears at shear strains of ~1.3 and ~0.5, respectively. The curve of φ at β = 0.6β0 was found to closely match the experimental results in this study. The modelled CPOs at this value of β are illustrated in Fig 12."

**Implications:** In the introduction, you outline the broad relevance of CPO development for ice and mantle material on Earth as well as on other planetary bodies. However, the discussion and conclusion sections lack detail on how your work relates to these broader implications and in where this strong shear deformation at relatively warm temperatures used in your experiments is likely to occur in nature. For example, what are the implications of this study for CPO developments in shear margins of ice streams? It would also be helpful to have a brief discussion on limitations of this kind of experiment in representing conditions found in natural ice deformation (e.g. different timescales)

We thank the suggestions. Now a subsection "4.8 Implications to natural ice" is added.

"As ice is a highly anisotropic material, once formed, the CPO has significant influences on the mechanical strength of ice, and thus, models of the flow of natural ice often rely on applying an anisotropy factor to the laboratory-derived flow laws (Pimienta et al., 1987). Although the models using an anisotropy factor obtained from historic observations generally predict the right magnitude of glacial flow rates, Azuma's CPO-only flow law (Azuma, 1994) best describes the strength evolution as strain increases (Fan et al., 2021). The anisotropic factor in Azuma's CPO-only flow law is not based on phenomenological data but calculated from the orientation data of

*c* axes. Thus, this study and many previous studies focusing on the CPO development in ice aim to understand the physical processes that controls the evolution of c-axis orientation during deformation, and thus, to better constrain the anisotropy factor and predict the glacial flow rates, especially for ice-stream margins, where shear deformation is severe.

One important result from the observations of this study is that the secondary *c*-axis cluster remains its orientation and weakens with increasing strain. This result changes our previous intuitive hypothesis and provides different values of anisotropy factors at strains when the *c*-axis fabric is evolving from double clusters to a single cluster. Such fabric evolution could occur at regions not too far away from the dome, where the shear deformation just starts, and possibly at the upstream regions of ice-stream margins, where the shear plane changes from horizontal to vertical, and the fabric has to evolve accordingly. However, it is necessary to note that the laboratory observed microstructures and their evolutions are obtained at strain rates and stresses larger than those in natural ice bodies. The contribution from rotation recrystallization in natural ice could be weaker, as stresses are smaller. Moreover, natural ice is usually impure. Insoluble particles and air bubbles could accumulate along grain boundaries and reduce grain boundary mobility and inhibit GBM, which possibly also occurred in the experiments of this study. The effects of the two mechanisms can only be qualitatively discussed for natural conditions. The microstructural processes observed in laboratory experiments provides good constrains on models and simulations (e.g., Richards et al., 2021; Hunter et al., 2022), which could extrapolate laboratory results to natural ice."

**Axes Definition**: Somewhere in the manuscript, you should define a-axes, m-planes, and c-axes with their corresponding Miller indices. It is somewhat defined in line 250, but this is rather late. c-axes are defined in the introduction, so it would make sense for the other two being described there as well.

This is a good suggestion. We have added sentences defining these axes early in the Introduction section. Please check our reply to the second major comment.

Updated sentence: "The CPO of ice is typically characterized by the alignment of the ⟨0001⟩ axes (*c* axes), normal to the basal planes. The orientation of other crystal axes, such as the ⟨11-20⟩ axes (*a* axes) and the poles to {10-10} planes (poles to *m* planes), could also serve as important kinematic indicators (Schmid and Casey, 1986)."

**Response to specific comments**

line 4: To examine the impact of strain *on the relative importance of these two*

*mechanisms*, might be more clear.

Thanks for your suggestions. The sentences are modified as follows:

line 4-6: "To examine the impact of strain on the relative importance of these two mechanisms, synthetic ice (doped with $\sim$1 vol.% graphite) was deformed using equal-channel angular pressing technique, enabling multiple passes to accumulate substantial shear strains."

line 20: water ice or *Ice Ih*.

line 20: '*the most common compound in the universe*' needs citation. Or leave out, I think this is redundant

line 21: satellites, (comma)

Thanks for your suggestions. The revised text addressing the above three points as follows:

line 20-21: "Ice Ih, constitutes glaciers and ice sheets on Earth, polar ice caps on Mars, icy shells on icy satellites, and a major part of many dwarf planets and asteroids in the solar system."

line 25-26: I find this sentence wordy and complex. Consider breaking it down to improve readability.

Thanks for your suggestions. The sentences are modified as follows:

line 25-26: "During plastic deformation of ice, crystallographic preferred orientations (CPOs) are induced. These CPOs translate the kinematics into anisotropy in the microstructure, as ice is a highly anisotropic material."

line 27: than on other slip planes (Duval et al, 1983); 0001 is defined as c-axis in line 46?

In crystallography, the notation (hkl) typically represents a crystal plane, while [hkl] denotes a crystallographic direction. For the hexagonal crystal system of ice Ih, the plane (0001) corresponds to the basal plane, and the direction [0001] indicates the orientation of the c-axis. Line 27 defines the (0001) plane; Line 46 defines the [0001] axis.

line 28: 'to ice' is unnecessary and can be omitted.

Thanks for your suggestions. The sentences are modified as follows:

line 28- 29: "Once a CPO is formed in polycrystalline ice, it can lead to elastic and

viscous anisotropies, making shear parallel to the aligned basal planes easier and deformation in other orientations more difficult (Azuma, 1995)."

line 30: Omit 'subsequent'. Not clear what 'response' refers to in this sentence. I suggest rephrasing to something like: When the stress field driving ice flow changes, the rate of ice deformation depends on the existing CPO and its evolution under the new stress configuration (Hudleston, 2015).

Thanks for your suggestions. We do mean the CPO evolution after the change. We think "subsequent" should be kept. The unclear usage of "response" is removed.

line 29-31: "When the stress field that driving ice flow changes, how fast the flow rate changes depends on the existing CPO and its subsequent evolution under the new stress configuration (Hudleston, 2015; Gerber et al., 2023)."

line 31: perhaps add gravitational forcing to the list?

Thanks for your suggestion. The sentences are modified as follows:

line 31-33: "Thus, the mechanical response of terrestrial and planetary ice bodies to climate, tidal, gravitational forcing and/or geological forcing depends, in part, on the evolution of their CPOs."

line 34: For CPOs in ice sheets on Earth, radio-echo sounding (RES) is another widely used technique to derive CPOs. You could consider citing e.g.:
https://tc.copernicus.org/articles/16/1719/2022/
https://ieeexplore.ieee.org/document/8755860
https://tc.copernicus.org/articles/17/1097/2023/

Thank you for recommending the radio-echo sounding (RES) methods and article. The sentences are modified as follows:

line 33-36: "Moreover, CPOs observed in natural ice samples (Jackson and Kamb, 1997; Jackson, 1999; Faria et al, 2014; Thomas et al, 2021) and derived from seismic data (e.g., Lutz et al, 2020, 2022) or radio-echo sounding (e.g., Jordan et al.,2019; Ershadi et al., 2022; Zeising et al., 2023) can provide valuable insights into the conditions and history of ice deformation. This is similar to how CPOs for quartz (Schmid and Casey, 1986; Law, 2014) and olivine (Karato et al., 2008) are used to understand deformation in the Earth's crust and mantle, respectively.

line 37-39: What kind of 'transition' have been observed? change in the dominant mechanism for CPO deformation between different experiments? Or with time/increasing strain? Needs to be clarified.

line 37-41: "Many experimental studies focusing on the evolution of CPO in ice have found a transition of the crystallographic fabric with changing stress, temperature and strain (e.g., Qi et al, 2017, 2019; Fan et al, 2020, 2021). The CPO of ice is typically characterized by the alignment of the $\langle 0001 \rangle$ axes ($c$ axes), normal to the basal planes. The orientation of other crystal axes, such as the $\langle 11\text{-}20 \rangle$ axes ($a$ axes) and the poles to {10-10} planes (poles to $m$ planes), could also serve as important kinematic indicators (Schmid and Casey, 1986). This transition is attributed to the transition to a change in the dominant mechanism for CPO formation from grain boundary migration (GBM) to lattice rotation and subgrain rotation (polygonization) (Alley, 1992; Qi et al., 2017; Fan et al., 2020). The two mechanisms are also referred to as migration recrystallization and rotation recrystallization (Poirier, 1985, pp. 179–185). "

line 42-45: long sentence, consider breaking it down:

*While uni-axial compression tests are commonly used to study ice microstructural evolution during deformation, high-strain deformation in ice sheets and glaciers is mainly simple shear (Cuffey and Paterson, 2010). Therefore, understanding CPO evolution under shear is critical.*

Thank you very much for your suggestion. The sentence is changed as follows:

line 42-45: "Since high-strain deformation in ice sheets and glaciers is dominantly simple shear (Cuffey and Paterson, 2010), understanding the CPO evolution under shear is critical."

line 47-49: sentence seems a bit wordy. Suggestion: The highest shear strain achieved in the lab, 2.6, was reported by Qi et al (2019) at −30°C. Previous experiments at similar temperatures did not exceed shear strains of 0.12 (Wilson and Peternell, 2012).

line 50: ... cluster. Instead ... (break sentence here).

line 50: Not clear what gamma refers to. Please clarify.

line 49-52: I think this part is a bit confusing. I would suggest first stating that natural ice samples often experience shear strains larger than 5, which leads to the development of a single cluster of [0001] axes. Then go on and explain that this could so far not be reproduced by lab experiments, which even under strains larger than 2 fail to produce such a strong single maximum fabric.

Thank you very much for your suggestions in these comments; we would like to accept them. The sentences are modified as follows:

line 47-52: "The highest shear strain achieved in the lab, $\gamma = 2.6$, was reported by Qi et al. (2019) at -30°C. Previous experiments at similar temperatures did not exceed shear strains of 0.12 (Wilson and Peternell, 2012). Natural ice samples are characterized by a double cluster of $c$ axes at low strains and typically exhibit a sharp, concentrated single cluster of $c$ axes at high shear strains of $\gamma > 5$ (Hudleston, 1977; Jackson, 1999; Thomas et al., 2021). Shear experiments on polycrystalline ice have found that the fabric transition from double-clustered to single-clustered $c$ axes occurs with increasing strain (Kamb, 1972; Bouchez and Duval, 1982; Li et al., 2000; Wilson and Peternell, 2012; Qi et al., 2019). However, even at shear strains of $\gamma > 2$, the $c$-axis fabric has not evolved to an absolute single cluster. Instead, the $c$ axes form a diffused cluster at $\gamma = 2.6$ and $-30$°C, and a very weak secondary cluster at $\gamma = 2.2$ and -20°C (Qi et al, 2019)."

line 53: single, primary cluster *evolving* at smaller...

Thank you for the suggestion. Yes, adding a word there would be helpful. We think "occurring" could be a better choice for the meaning of the sentence. The sentences are modified as follows:

line 53-54: "Meanwhile, numerical simulations incorporating dynamic recrystallization processes have found a single, primary cluster occurring at smaller shear strains than experiments (Llorens et al, 2017; Piazolo et al, 2019; Richards et al, 2021)."

line 55: omit '*simply*'

line 54-56: "This discrepancy leads to an uncertainty on to whether the CPO in natural ice can be explained by the formation mechanisms proposed from experimental observations."

line 56: at higher strains

line 57: are needed.

Thank you very much for your suggestions. "Deformation experiments to high strain" is the common writing. Experiments cannot be carried out "at" a strain. An experiment starts at strain of 0 and terminates once the sample is deformed "to" a target strain. We hope to keep using "to".

line 56-57: "Thus, new laboratory experiments to higher strains, closer to those found in naturally deformed ice, are needed."

line 59: move the alternative definitition one line up: *Equal-channel angular pressing*

*(ECAP; also known as equal-channel angular extrusion) is a technique for generating severe plastic deformation, resulting in highly strained microstructures with ultra-fine grain sizes and strong fabric. ECAP was initially developed in the 1980s ...*

Thanks for your suggestions. Combining the first major comment, the sentences are modified as follows:

line 58-60: "In this contribution, we adapt the equal-channel angular pressing (ECAP) method to polycrystalline ice. ECAP, also known as equal-channel angular extrusion, is a technique for generating severe plastic deformation, resulting in highly strained microstructures with ultra-fine grain sizes and strong fabric through repeated deformation of the sample. ECAP was initially developed in the 1980s by V.M. Segal and colleagues (Segal et al, 1981)."

line 63: intersect at an angle Φ

Thanks for your suggestions. As suggested in the first major comment, these sentences are moved to Methods section.

Section 2.2: "The channel angle Φ represents the angle at which the two channels of equal cross-sectional area intersect and is a critical factor influencing the shear strain."

line 79: it would be nice to have one sentence summarizing the most relevant findings and relevance of this study at the end of the introduction.

Thank you for the suggestion. The last two sentences of introduction are modified.

"The objective of this paper is to explore the effects of shear strain on the CPOs and microstructures of ice, especially when ice is deformed to shear strains higher than any previous experiments. The results could help understanding the physical processes that control the development of CPO in highly deformed natural ice."

Fig. 1:

Note: To make the responses more organized, I have arranged your comments according to the order of the panels in the Figure 1.

- caption: Photos and dreawings *of* the ECAP apparatus.

- panel a): arrows are hard to see. Consider a zoom window with a close-up of the shear plane. Photos could be arranged so the load arrow in panel c is not cut off.

- caption: both parts of the die are shown with a dummy sample in the channel (omit

comma).

- panel b): TEC is not defined in caption.

- panel c): LVDT is not defined.

Processed sample (instead of Sample processed?)

- caption: define abbreviations.

- caption: 'all passes are done without changing the orientation of the sample': how do you ensure the orientation is the same during the second deformation?

- panel c, d and caption: Thermistor misspelled (thermister)

Thank you for your suggestions. We have made changes accordingly. we have updated Figure 1 and its caption:

[Figure]

"Figure 1. Photos and drawings of the ECAP apparatus. (a) Photos focusing on the channel of the ECAP die. In the top photo, the channel angle, $\Phi$, and the curvature angle, $\Psi$, are illustrated. The bottom image is a zoomed-in view of the red box in the top image, providing a close-up view of the shear plane and shear direction. (b) Photo of the apparatus running an experiment. TEC: thermoelectric coolers. (c) and (d) Drawings of the ECAP apparatus from front and side views. Thermistors are stuck to the die on the outer surface. The plunger cannot bend at the corner, such that a spare piece of ice is added to the tail of sample to push the sample through the corner. Then the spare piece is removed. All passes are done without changing the orientation of the sample. LVDT: linear variable displacement transducer."

Meanwhile, we have changed "Peltier coolers" to "thermoelectric coolers" in the text.

Line 104: "Four thermoelectric coolers (TEC) were attached to the flat surface of the ECAP die, with the cooling side stuck to the die and the heating side stuck to a water box."

Line 108: "The TEC controller regulates the power of the TEC via proportional-integral-derivative method based on the target temperature, which is set to -5℃ for all experiments."

On the comments about how to ensure the orientation of a sample between passes. Because the ECAP die is made from two parts, as a sample passes through the die, a trace is formed on the outer surface of the sample, corresponding to the divide line of the two halves. When the sample is inserted in the device again, the trace is kept aligned with the divide, then ensuring consistent orientation during each deformation process. We have made corresponding changes in this paragraph, as in the subsequent reply for "line 124-133".

"Because the ECAP die is made of two halves, as a sample passes through, the channel leaves a trace on the outer surface of the sample, corresponding to the divide of the two halves. Using the trace, the sample can be reinserted into the channel with the same orientation relative to the corner and deformed again."

line 85: change 'has' to 'have'

line 86: 'for more details please refer to Table 1' or 'more details can be found in Table 1'.

Thanks for your suggestions. The updated text for above two points as follows:

Line 85-86: "Samples fabricated this way have a graphite fraction of 1.8–3.6 wt.%

(corresponding to 0.8—1.5 vol.%, for more details please refer to Table 1).”

Table 1: I think it would be helpful to describe parameters ε' and φ in the caption.

We have added the following in the caption: “ε' is nominal equivalent strain, φ is the angle between the two clusters”.

Table 1. Summary of experiments. ε' is nominal equivalent strain, φ is the angle between the two clusters.

| Sample | Graphite fraction | Passes | Load (Kg) | Length (mm) | ε' | Part for analysis | points of EBSD data | area(mm*mm) | φ |
|---|---|---|---|---|---|---|---|---|---|
| ECAP_33 | 2.1wt.% (0.9 vol.%) | 1 | 42.5 | 110 | 0.6 | ECAP_33_1P | 144045 | 14.85*8.73 | 50° |
| ECAP_19 | 3.6wt.% (1.5 vol.%) | 1 | 42.5 | 98 | 0.6 | | | | |
| | | 2 | 37.5 | 100* | 1.2 | ECAP_19_2P | 99099 | 12.87*6.93 | 55° |
| | | 3 | 37.5 | 50 | 1.8 | ECAP_19_3P | 104550 | 12.75*7.38 | 50° |
| ECAP_21 | 1.8wt.% (0.8 vol.%) | 1 | 42.5 | 110 | 0.6 | | | | |
| | | 2 | 37.5 | 105 | 1.2 | | | | |
| | | 3 | 37.5 | 93 | 1.8 | | | | |
| | | 4 | 32.5 | 85 | 2.4 | ECAP_21_4P | 108035 | 15.81*6.15 | 45° |
| ECAP_34 | 2.6wt.% (1.1 vol.%) | 1 | 42.5 | 107 | 0.6 | | | | |
| | | 2 | 32.5 | 99 | 1.2 | | | | |
| | | 3 | 32.5 | 95 | 1.8 | | | | |
| | | 4 | 32.5 | 85 | 2.4 | | | | |
| | | 5 | 32.5 | 65 | 3 | ECAP_34_5P | 112892 | 13.86*7.97 | 60° |
| ECAP_38 | 2.2wt.% (0.9 vol.%) | 1 | 42.5 | 105 | 0.6 | | | | |
| | | 2 | 32.5 | 100 | 1.2 | | | | |
| | | 3 | 32.5 | 95 | 1.8 | | | | |
| | | 4 | 32.5 | 95 | 2.4 | | | | |
| | | 5 | 32.5 | 92 | 3 | | | | |
| | | 6 | 32.5 | 94 | 3.6 | ECAP_38_6P | 161100 | 16.11*9 | 55° |

line 90: This first sentence is not very informative. Remove and change the second sentence to: 'The ECAP die consists of two symmetrical stainless-steel parts, as shown in Fig. 1a. '

Thanks for your suggestions. Since we have modified Figure 1(a), we will also update the corresponding description. The sentences are modified as follows:

line 90: “The die consists of two symmetrical stainless-steel parts, each as half of the

channel."

line 92: The diameters of the channel and ice samples is 25 mm .

line 92: "The diameters of both the channel and the ice samples are 25 mm."

line 92-93: perhaps use dash for 'mirror-finished'. The quotation marks seem strange here, I would simply remove it. Is the soap coating refreshed between experiments? Please clarify.

line 92-93: "The channels are mirror-finished, and a layer of solid soap is coated before each experiment to minimize friction during the experimental runs."

line 93: The channel *geometry* is defined by two angles;

line 94: remove '(equal to 120° in Figure 1(a))'. Instead add '(see Fig. 1a)' at the end of the sentence.

Thanks for your suggestions. The updated text for above two points are as follows:

line 93-94: "The channel geometry is defined by two angles: the channel angle $\Phi$ represents the angle at which the two channels intersect and is a critical factor influencing the shear strain, and the curvature angle $\Psi$ defines the angle at the outer curvature arc where the two segments intersect (see Fig. 1a)."

Eq. (1) : I am confused on why strain here is expressed as $\varepsilon$ and as $\varepsilon'$ in Table 1. I also think it is strange to use '=' for the definition of $\Phi$ and '$\approx$' for the definition of $\psi$. **Can the angle $\psi$ be determined less accurately or why is that?** In addition to this equation, it might be helpful to state the relationship of the equivalent strain and the shear strain.

We agree with your comment. As mentioned in lines 225-235, due to the differences between ECAP experiments and traditional deformation tests, we use $\varepsilon'$ to denote the nominal equivalent strain to avoid any potential misunderstanding by readers, distinguishing it from true equivalent strain $\varepsilon$. And for the shears in ECAP, we should use $\varepsilon'$, as its is different from the equivalent strain from traditional compression or shear experiments. We also move the definition of nominal strain from section 3.3 to this section.

"It can be shown from first principles that the nominal equivalent strain after N passes, $\varepsilon'_N$, is given by a relationship of the form (Iwahashi et al., 1996)

$$\varepsilon'_N = \frac{N}{\sqrt{3}} \left( 2 \cot \left( \frac{\Phi}{2} + \frac{\Psi}{2} \right) + \Psi \operatorname{cosec} \left( \frac{\Phi}{2} + \frac{\Psi}{2} \right) \right).$$

For our design, $\Phi = 120°$ and $\Psi = 60°$, theoretically resulting a nominal equivalent strain of $\varepsilon' \approx 0.6$ (with the nominal shear strain being $\gamma' = \sqrt{3}\varepsilon' \approx 1.0$) in the sample per pass. We note that the ECAP samples are deformed differently from traditional experiments in which the whole sample deformed simultaneously (e.g., Kamb, 1972; Qi et al., 2019). Thus, we refer the calculated strain as the nominal strain, denoted by a prime. ε' is nominal equivalent strain and γ' is nominal shear strain."

line 112-113: Once  the sample temperature had equilibrated with the die (typically after 20 min), a load was applied to the sample using dead weights hung on the aluminum profile

Changed as suggested.

"Once the sample temperature had equilibrated with the die (typically after 20 min), a load was applied to the sample using dead weights hung on the aluminum profile."

line 115-117: why is that? In some cases (ECAP_19, 21) the load of the second pass decreases by 5kg, while for others (ECAP_34, 38) it decreases by 10 kg and being stable for further passes. Please clarify.

When the sample is deformed using the ECAP device, can be regarded as an environment without confining pressure. Consequently, if the sample passes through the ECAP device too quickly, it may crack. Typically, the rate at which the sample passes through the ECAP for the (n+1)th passes is faster than for the nth passes. That is due to the formation of a CPO that weakens the sample in the shear direction (that is where the CPO-induced anisotropy comes in). To prevent cracking due to excessive speed, we gradually reduce the load with each pass to slow the rate and ensure the sample's integrity.

line 116: for *a* single sample; (*for* more details

"It is worth noting that as number of passes through the channel increases for a single sample, the maximum load applied decreases slightly (for more details please refer to Table 1 and Supplementary Fig. S1)."

Note that Supplementary Fig. S1 is the appendix Figure A1 in the current manuscript. Since we are going to add several supplementary figures, we want to move this one

into supplementary as well.

line 120-123: I think this sentence should be shortened/broken up and simplified.

"When the sample tail reaches the corner of the channel, the plunger can't push it further. So, we insert a spare piece of ice (not for testing) to push the sample through the corner, completing a 1-pass experiment."

Fig. 2: - caption: Drawings describing illustrating the accumulation of shear straine in a sample deformed from the i-th pass to the (i+2)-th pass in side view.

- general: In the case illustrated i=1, right? I am unsure of the generalization with 'i' since the strain ellipse suggests it has not been deformed previously before pass i. I would suggest changing it to first, second, third pass and adding a comment in the bottom of the figure or the caption, stating that for further passes step 3 is repeated with adding additional ice to compensate the length.

Thank you for your suggestion. Very reasonable. We made changes in the figure accordingly and caption addressing the above two points.

[Figure]

"Figure 2. Drawings describing the accumulation of shear strain in a sample deformed from the first pass to the third pass in side view. For the preparation of each pass, both ends of the sample must be cut flat and perpendicular to the cylindrical axis. For subsequent passes, the process is repeated. It is important to note that the addition of an extra section is not exclusive to the third pass; rather, an additional section is added to the sample to make up the length, once the sample length falls below 50 mm. The circle in the sample is a strain ellipse, describing the theoretical strain accumulated by each pass."

line 124-133: This section sounds in parts as if it is a result, rather than experiment set-up or method. For example (line 125) ' indicating shearing' seems redundant when you are describing a shear deformation experiment. See my suggested edits below:

Upon passing through the channel, the sample's diameter  *remains* unchanged, but the head and tail surfaces  are no longer perpendicular to the cylindrical axis,  (see Figures 2 and 3). In *preparation for additional passes* , the head and tail surfaces were , restoring the sample's  cylindrical shape  *with* a reduced length *compared to the original sample* (Figure 2). The sample can be reinserted into the channel with the same orientation relative to the corner and deformed again.  *For* samples  *becoming* shorter than 50 mm, an additional piece of *ice*  was added  to achieve the required length *but this extra ice was not considered in the analysis* (Figure 2). This process allows for multiple passes through the channel, accumulating high strains. After reaching the target number of passes, the sample*s*  *were* wrapped in aluminum foil and stored in liquid nitrogen. Note that during an experiment, the head part of the sample  is exposed to air, while the tail part  *is* deformed in the corner, causing more sublimation in the head part.  Instead, the middle part near the tail was used for analysis in the next subsection. *To avoid sublimation bias, we used the middle part near the tail for microstructural analysis.*

Thank you very much for the suggestion. We have changed the paragraph accordingly.

"Upon passing through the channel, the sample's diameter remains unchanged, but the head and tail surfaces are no longer perpendicular to the cylindrical axis (see Figures 2 and 3). In preparation for additional passes, the head and tail surfaces were restored to the sample's cylindrical shape with a reduced length compared to the original sample (Figure 2). Because the ECAP die is made of two halves, as a sample passes through, the channel leaves a trace on the outer surface of the sample, corresponding to the divide of the two halves. Using the trace as a marker, the sample can be reinserted into the channel with the same orientation relative to the corner and deformed again. For samples becoming shorter than 50 mm, an additional piece of ice was added to achieve the required length, but this extra ice was not considered in the analysis (Figure 2). This process allows for multiple passes through the channel, accumulating high strains. After reaching the target number of passes, the samples were wrapped in aluminum foil and stored in liquid nitrogen. Note that during an experiment, the head part of the sample is exposed to air, while the tail part is

deformed in the corner, causing more sublimation in the head part. To avoid sublimation bias, we used the middle part near the tail for microstructural analysis."

line 126-127: How do you ensure that the orientation is preserved? Is it marked by something? Please clarify.

Because the ECAP die is made from two parts, as a sample passes through the die, a trace is formed on the outer surface of the sample, corresponding to the divide line of the two halves. When the sample is inserted in the device again, the trace is kept aligned with the divide, then ensuring consistent orientation during each deformation process. We have made corresponding changes in this paragraph, as in the previous reply.

line 128-129: Where is the additional piece of ice added? From Fig. 2 it looks as if it is added at the tail. Since the head part is more prone to be affected by sublimation, why not add this piece at the head? Maybe it is, but please clarify here in the text and mark head and tail in Fig. 2.

The spare piece of ice is added at the tail end. As we explained in lines 120-123, the main purpose of the extra ice we added to push the sample out of the channel, rather than to prevent sublimation. We need the extra material so that all of the sample has passed the corner, leaving the extra piece in the corner. Tail and head are marked in Figure 2 as suggested.

line 134-135: To investigate the effect of annealing on pre-existing CPO,  one sample was annealed at −3.5 °C for 24 days after ECAP deformation. The annealing experiment was done using a similar apparatus described in Fan et al (2023). Can you state which of the samples in Table 1 that is?

Changed as suggested.

"To investigate the effect of annealing on pre-existing CPO, one sample (ECAP_38_6) was annealed at -3.5 °C for 24 days after ECAP deformation. The annealing experiment was done using a similar apparatus described in Fan et al (2023)."

Fig. 3: - panel c & d): It would be good to have an approximate scale on these photos too.

Abbreviation SEM in panel d) needs to be defined in caption.

- caption: Illustration for a sample deformed and prepared for microstructural analysis.  (a) Starting

sample before deformation. (b) Sample deformed by ECAP. (c) Sample cut along profile plane. (d) Sample polished and mounted on a copper ingot for analysis. Note that the  black color stems from graphite powder added to the ice samples

Thank you for the suggestions. We added rough scale markers in the two panels and updated caption of panel (c) and (d), and changed the caption as suggested.

[Figure]

"Figure 3. Illustration for a sample deformed and prepared for microstructural analysis. (a) Starting sample before deformation. (b) Sample deformed by ECAP. (c) Sample cut along profile plane. (d) Sample polished and mounted on a copper ingot for analysis. Note that the black color stems from graphite powder added to the ice samples. SEM: Scanning Electron Microscope."

line 139: why is sample storage so cold and for how long? Do temperature changes from -5 to -190/-120 to -10 degrees not affect your analysis?

The sample is cooled rapidly from -5°C to low temperature. Just like the quench process for a rock sample in high-temperature experiments. At low temperatures (-190/-120°C), defect activity within the crystal is minimal, and the dislocation recovery rate of ice is also very low. Therefore, at this temperature, the microstructure of the sample will not change over time and can be preserved.

Samples transfer from the low-temperature Dewar ($\leq -190$°C) or the heat transfer box ($\leq -120$°C) to the $-10$°C cold room involves a heating process. The cutting should be completed swiftly in the $-10$°C environment, including the time for heating and cutting, which typically takes about 15 minutes. After cutting, the samples are quickly re-cooled to $-190$°C or $-120$°C. While some thermal effects may occur during this process, but they are considered acceptable. You may find the standard procedures for doing EBSD on ice in Prior et al. (2015), which is a summary of more than ten years of trials, including determining what temperatures have minimal effect on the analysis.

 I think it would be helpful to be more specific here on how the cut is oriented relative to the shear plane. Alternatively, define what you mean with 'profile plane' and indicate in Fig. 1. How thick is this 'section'?

"The sample was first cut in half along the profile plane, which is parallel to the cylinder axis and perpendicular to the shear plane, and then cut again parallel to the initial cut to obtain a 5-mm-thick section of the profile plane (see Figure 3 ). This was done using a band saw in a cold room maintained at -10°C."

line 145-146: ""

I don't think this sentence adds much information as it stands now.

Thank you for your suggestion, we deleted this sentence

line 152: It would be helpful to state the approximate sample size here or somewhere else. Were they all cut to the same size?

They were all cut to similar sizes. For the orientation data with 30-µm step size, the collected area ranged from 0.9 to 1.4 cm$^2$. This information is added to Table 1.

line 154: Subsequently, carbon element data were obtained from EDS in the for the same selected regions with a step size ranging from 3.2 to 6.7 µm. (assuming this is only done for regions with step size of 15 micrometer. Else please clarify)

Your understanding is correct, this is only done for regions with step size of 15 micrometer. We adjusted the word order to make it easier for readers to understand.

line 152-155:

"For each sample, several smaller orientation maps with a step size of 15 µm were collected from selected regions of the section. Carbon elemental data were then obtained from EDS in these same regions, with step sizes ranging from 3.2 to 6.7 µm. Subsequently, an orientation map with a step size of 30 µm was acquired for the majority of the section."

line 156: I think the first sentence is redundant. Rather than describing what you didn't do, simply describe what you did.

Orientation data obtained from diffraction data with a step size of 30 µm  were used for analyzing the CPO patterns.

This is a good suggestion. We have made the changes, and also move this sentence to a new subsection, as replied to the first major comment.

"Orientation distributions were generated from the complete set of raw EBSD data with 30 μm step size using the MTEX toolbox in MATLAB (Bachmann et al, 2010; Mainprice et al, 2015)."

line 157-158: Here you say that 15 micrometer data is analyzed with MTEX. In line 176 you mention that 30 micrometer data is analyzed with MTEX toolbox. Why not generalize this in one sentence in the beginning of the paragraph in something like ' We used the MTEX MATLAB toolbox (citations) to process orientation data' or similar.

The two data sets are used for different purposes. As suggested, we separated them into two subsections. Please refer to the reply to the first major comment .

line 166-167: To match the dimensions of the EBSD data, the pixel size of *the* EDS map was adjusted to match the step size of *the* EBSD. Then these pixels were attributed to *the* graphite phase in *the* EBSD data.

Changed as suggested.

"To match the dimensions of the EBSD data, the pixel size of the EDS map was adjusted to match the step size of the EBSD. Then these pixels were attributed to the graphite phase in the EBSD data."

line 168:  *By combining* the EBSD data from ice and graphite, a data set with two phases was obtained (Figure 4(e)).

Changed as suggested.

"By combining the EBSD data from ice and graphite, a data set with two phases was obtained (Figure 4(e))."

line 168-169: Note that this method cannot identify individual graphite particles, but only  *reveals* graphite-rich regions. (mixed tense)

line 169-170: Repeat of previous sentence.

Thank you for the suggestions. We combine the two comments. The sentences are changed to as follows:

"Note that due to the relatively low spatial resolution of EDS, individual graphite particles cannot be identified, and only graphite-rich areas are revealed. Consequently,

the data processed by this method typically indicate a higher graphite fraction."

line 171-172: Not very clear what you're doing here. Are you really 'reconstructing grains? Do you mean you locate grain boundaries by looking for orientation changes between pixels of more than 10 degrees? Please clarify.

Yes, we basically did what you said, and this process is called "reconstructing grains" in MTEX. The name is given by Prof. David Mainprice and co. (they made the MTEX toolbox), and now is widely accepted. Based on the above explanation, we update the statement as follows.

line 171-172: "The ice grains were digitally 'reconstructed' from the processed data using the MTEX algorithm (Bachmann et al., 2010). In this process, misorientation between neighbouring pixels exceeding 10° were used to identify grain boundaries, while misorientation exceeding 2° were used to identify subgrain boundaries."

line 177-179: consider splitting this sentence in two.

"The J-index, based on a calculated orientation distribution function, increases from 1 (random) to infinity (single crystal). The M-index, which is based on the distribution of random-pair misorientation axes, increases from 0 (random) to 1 (single crystal)."

line 181-183: We adopted the same method *for* cluster identification used previously in Qi et al (2019), *where t*he-normalized counts of data per orientation in the profile plane were plotted on a histogram. φ was defined as  the angular width between the two peaks on the histogram.

line 183: reference to Fig. 7d?

Based on both comments, we have reorganized the language in this section to improve clarity and added a more detailed description of Fig. 7d as follows.

"We adopted a method similar to that of Fig. 2 in Qi et al. (2019). As illustrated in Supplementary Fig. S2(b), pole figures were generated using a lower hemisphere equal-area projection, with the shear plane (green circle) oriented perpendicular to the page. In the stereonets, angles ranging from 0 to 180° were defined on the shear plane. At a given angle, two semicircles with 5∘ between them (orange circle) were drawn perpendicular to the page. The number of data points falling between these semicircles was counted, normalized, and plotted as the frequency for each angle in a histogram. The angle φ was defined as the angle between the two peaks in the histogram (Supplementary Fig. S2(c)).

Please refer to the figure in Page 4 of this file.

line 185:  *The observed* CPOs were compared to predictions from the *spectral continuum anisotropic fabric evolution* (SpecCAF) model (Richards et al, 2021).

Changed as suggested.

"The observed CPOs were compared to predictions from the spectral continuum anisotropic fabric evolution (SpecCAF) model (Richards et al, 2021)."

line 186: The SpecCAF model cannot directly simulate microstructural *changes*  *like* other models, e.g., ELLE, (Jessell et al, 2001).

line 185-189: I think these section is confusing and should be clarified by being more specific of how 'microstructural processes' are simulated differently between these models. 'The evolution of CPO' which 'SpecCAF simulates' could also be interpreted as 'simulating microstructural changes'.

It could also be mentioned that in contrast to more complex models, such as ELLE, SpecCAF is computationally more effective and more suitable for large-scale CPO modeling.

Considering both comments, we changed the first two sentences as follows:

"The observed CPOs were compared to predictions from the spectral continuum anisotropic fabric evolution (SpecCAF) model (Richards et al., 2021). The SpecCAF model cannot directly simulate some microstructural changes, such as grain-scale deformation, like other models, e.g., ELLE, (Jessell et al 2001)."

line 192: The numerical model  *adapted from* Richards et al (2021), producing simulated pole figures *representing the distribution of c-axis orientation*, and the angle, φ, *describing the angular distance between clusters*.

line 193: 'φ was compared to model predictions with a variety of β = kβ0 values': I don't understand this. Do you mean experimentally observed φ were compared to modelled? Or do you mean φ was tuned to observations by adjusting β? Either way, needs clarification.

By adjusting β, we get different values of φ in the model. We have modified the sentences based on both comments.

"The numerical model adapted from Richards et al (2021), producing simulated pole figures representing the distribution of *c*-axis orientation, and the angle, φ, describing the angular distance between clusters. Different values of φ were simulated based on

the numerical model with a variety of $\beta$. $\beta = k\beta_0$, where $\beta$ controls the magnitude of migration recrystallization effects on the modeled CPO, with $0 \leq k < 1$, and $\beta_0$ represents the value for natural ice at T = -5 °C."

line 195: How many, and in what increments?

We tested k = 0.2, 0.4, 0.5, 0.6, 0.8 and 1. This sentence is changed to include this information.

"Different values of k (0.2, 0.4, 0.5, 0.6, 0.8 and 1) were used to assess the relative contributions of GBM and lattice rotation to CPO."

Fig. 4:

  - title of panel c): Combined pixels - combined pixels of what? panel (a) and (b)?

- caption: The caption reads in part as instructions, rather than a description of what is shown. I find the description of panel e) especially confusing ((e) Knowing the coordinates of graphite from (c), create "EBSD" data of graphite. Combine the ice phase and the graphite phase.)

Suggested edits: The example used shown here is sample ECAP_38_6P.

(c) Combined pixels so that the pixel size is the same as the 15 μm step size of the EBSD data., which is 15 μm.

(e) Knowing the coordinates of graphite from (c), Created "EBSD" data of graphite, informed by the graphite coordinates in (c), and combined with Combine the ice phase in (d). and the graphite phase.

(f) Denoised the data with grain boundaries tracked by the MTEX toolbox. and reconstruct grains with MTEX toolbox. Note that by this method, the identified graphite phase represents the upper limit of the area fraction of the graphite.

We really appreciate your suggestions. Very helpful! We have made the changes accordingly. The pixel size in the EDS map in panel (b) is 6.3 microns, which is smaller than the step size (pixel size) of the orientation map in panel (d). In order to put both phase data together, we did pixel binning to make them the pixels in both panels the same. We have changed the caption to make it clear.

[Figure]

"Figure 4. Illustration for the combination of EDS and EBSD data to locate the graphite phase. The example shown here is sample ECAP_38_6P. (a) Raw data for carbon obtained from EDS. (b) Data with signal strength ⩾ 2. (c) Pixel binning process so that the pixel size is the same as the 15-μm step size of the EBSD data. (e) Created 'EBSD' data of graphite, informed by the graphite coordinates in (c), and combined with the ice phase in (d). (f) Denoised data with grain boundaries tracked by the MTEX toolbox. Note that by this method, the identified graphite phase represents the upper limit of the area fraction of the graphite."

line 198: The ice microstructure of the starting materials was similar in character to that described in (Qi et al, (2017).

Sorry for the typo.

"The ice microstructure of the starting materials was similar in character to that described in Qi et al, (2017)."

Fig. 5: - panel labels would be helpful.

- colorbar: I would suggest adjusting the colorbar for the stereoplots to range from 0 to 1.6 and add at least one more number between to indicate that it is linear (e.g. 0,0.8,1.6) - it might be helpful to indicate that [0001] is the c-axis, and [1120] and [1010] are axes in the basal plane of the crystal here.

- the subgrain boundaries look almost white and are hard to see - use a darker gray for better visibility.

- labels of IPF-Y colorscale are very small - y-axis in caption is undefined. Is it vertical to the sample cut? Please clarify.

- caption: Microstructural analyses of an undeformed ice  *sample*.

'The grain-size data are calculated from a larger area' - larger than what? how large, and is it the same size for all samples?   ',  791 grains *in this sample*.'

Thank you for the suggestions. We have added panel labels. We added a colorbar ranging from 0 to 2. We hope that is helpful. Subgrain boundaries are changed to darker gray. We have enlarged the labels in the IPF-Y color map, and added the x-y coordinates. Sorry for the typo. We have corrected that.

[Figure]

"Figure 5. Microstructural analyses of an undeformed ice sample. The orientation map on the left is colored by IPF-Y, which uses the color map to indicate the specific crystallographic axis that is parallel to the y-axis. The step size is 15 µm. Grain boundaries, characterized by a misorientation of $\geqslant 10°$, are in black, and sub-grain boundaries, characterized by a misorientation of $\geqslant 2°$, are in gray. Graphite is in black. The stereonets on the top right are for distributions of [0001], [11-20] and [10-10] axes. Data are based on all orientation data and colored by multiples of uniform distribution (MUD), as shown in the color bar. All stereonets are equal-area lower-hemisphere projections. The histogram on the bottom-right illustrates the grain-size distribution. The grain-size data are calculated from a larger area (approximately four times the area of panel (a)), 1199 grains in the entire map."

line 201: rather (better say how much smaller)

The caption is changed as follows:

"The mean ice grain size is 154 μm, which is approximately 80 μm smaller than that of pure polycrystalline ice made in a similar method (e.g., Qi et al, 2017)."

line 208: I think you mean the temperature is lower by 0.5°C. Better state -5.5°C to be more clear.

Good suggestion. The sentence is modified to "but the 6th pass shows a larger perturbation and a slightly lower average temperature (∼-5.5°C), likely due to variations in laboratory temperature and humidity."

line 214-215: I don't think this is true for ECAP_21 and ECAP_34, while we don't know for ECAP_33. Given that this is only true for 2 out of 5 samples, can you really say that the first pass is 'generally much slower' ?

You're right. Also, the first pass is under the largest load. It is hard to compare. Since we did not analyze the slope of the LVDT-time curve, this conclusion is somewhat arbitrary. We have decided to delete this sentence.

line 215-227: This section could be shortened and streamlined.

Line 215-220 seems more relevant for the discussion, rather than results

Thanks for your suggestions. The revised text addressing the above two points is as follows:

"Since load and displacement data from ECAP experiments are not well-suited for rigorous analysis of mechanical behavior (see Valiev and Langdon, 2006, for a review), this paper will not include an in-depth analysis of the mechanical properties, consistent with the limitations of most ECAP deformation studies. However, as ECAP is a relatively new technique for ice, the deformation pathway the sample experiences during pressing is described here. For each pass, the deformation starts at the head of the sample, with the stress in the head gradually rising to a maximum, resulting in a nominal equivalent strain of approximately 0.6 in the deformed part. Meanwhile, the rest of the sample is annealed at ≲-5°C. Subsequently, the deformed head slides through the outlet tube, with the stress to drop to zero. Simultaneously, the ice behind the head arrives the corner, leading to a gradual increase in stress to its maximum value and obtaining a nominal equivalent strain of approximately 0.6, while the rest, including both deformed head and undeformed parts, continues to anneal at ≲-5°C. This sequence is repeated until the tail of the sample passes the corner."

Fig. 6: - panel a): increase line thickness in the legend - it is hard to see the colors in

such thin lines. A very minor detail: why are the colors in panel (a) in pastel/different from (b) and (c) when denoting the same thing? (same comments for Fig. A1).

- I would adjust the y-axis to min/max achieved temperatures to increase visibility in panel (a) - green and red lines might be hard to be distinguished by colorblind people.

- panel (c), y-label: Equivalent nominal strain, ε'

Panel (a), lines are made thicker in the legend. They are in the same color as in (b) and (c). They were just too thin. Now they look the same. We tried to adjust the min/max in (a), it was not helpful. Actually we just want to show basically all experiments their temperature curves lying on top of each other (except for the 6th pass). Panels (c) is changed as suggested.

[Figure]

Fig. 7: - caption: The contours on the stereonets are colored by MUD, values of which  range from 0 to the maximum value indicated on top left of each stereonet.

- panel c): I think the maximum value of the colormap should also be stated here. Please state in the caption how the shear direction is now (from top to bottom? or left to right?). It seems as this has been tried to be indicated by the gray figure on top of panel c, but should be complemented by a unambiguous description.

Changed as suggested. Max values are added in panel (c).

Panel (a): "The contours on the stereonets are colored by MUD, values of which range from 0 to the maximum value indicated on top left of each stereonet."

Panel (c): "Distributions of orientations of [0001] axes. The shear plane rotated to be parallel to the paper, and the shear direction is topside up."

[Figure]

**-5°C**

| | (a) 1000 points [0001] | (b) All orientation data [0001] [11\(\bar{2}\)0] [10\(\bar{1}\)0] | (c) All orientation data [0001] | (d) Distribution of c axes |
|---|---|---|---|---|

ECAP_33_1P
$\varepsilon' \approx 0.6$
J = 3.22
M = 0.195

ECAP_19_2P
$\varepsilon' \approx 1.2$
J = 2.71
M = 0.196

ECAP_19_3P
$\varepsilon' \approx 1.8$
J = 3.79
M = 0.235

ECAP_21_4P
$\varepsilon' \approx 2.4$
J = 3.28
M = 0.26

ECAP_34_5P
$\varepsilon' \approx 3.0$
J = 3.90
M = 0.268

ECAP_38_6P
$\varepsilon' \approx 3.6$
J = 5.90
M = 0.441

ECAP_38_6P
**Annealed**
J = 5.15
M = 0.387

Fig. 8: - panel labels are missing.

- a legend in each panel would be helpful to see what the plots show faster.

Thank you for your reminder. We have added the panel labels and legend accordingly.

Fig. 9: - why not add the annealed sample to the legend too?

- additional math symbol in x-label would help for faster comparison with e.g. fig. 7, Table 1 and text.

Changed as suggested.

[Figure]

line 257-262: I'm not sure this belongs to 'results'. Since you already have a section in the discussion (4.2) dedicated to the effect of graphite, I would suggest moving this to 4.2.

Thank you for the suggestion. We propose that these sentences should be categorized under "Results". While it does encompass aspects related to the influence of graphite on the sample, this influence pertains primarily to the SEM technique rather than the microstructural outcomes. They are really not related to Section 4.2, where the physical influences of graphite on microstructural evolution and CPO development were discussed.

line 264: more *frequent* in the samples

Changed as suggested.

"Subgrain boundaries are observed in all samples, and are more frequent in the samples with higher strains."

line 266: deformed  *by* 1- 3 passes

Changed as suggested.

"In samples deformed by 1-3 passes ($\varepsilon' \leqslant 1.8$), grain sizes vary greatly, with many grains larger than 500 μm and many smaller than 100 μm."

line 267: deformed  *by* 4-6 passes

Changed as suggested.

"In samples deformed by 4-6 passes ($\varepsilon' \geqslant 2.4$), the deviation in grain size is smaller, the grainsize distribution is better fit by a log-normal distribution (oranges in Figure 10(b)), and the mean grain size is similar to the starting grain size."

line 270: The distribution of *grain* aspect ratios (I assume? Should be clearly distinguished from cluster aspect ratio)

line 270: deformed  *by* 2- 6 passes (as above. Notation doesn't matter, but should be consistent).

Both are changed as suggested.

"The distribution of grain aspect ratios varied very little in samples deformed by 2-6 passes."

line 278: multi-pass samples

Changed as suggested.

"In our experiments, apart from the single-pass sample (ECAP_33_1p), which underwent a single deformation and annealing, the multi-pass samples experienced cyclic deformation and annealing."

line 280-282: This overview of the discussion sections is very helpful. I suggest using section names to be more specific. I think you should also mention what section 4.3 and 4.7 is about.

Changed as suggested.

"We will focus on the influences of annealing and graphite in Sections 4.1 and 4.2, and analyze the elongated c-axis clusters in Sections 4.3. A careful comparison of the CPO observed in laboratory experiments, field samples, and models will be performed in Sections 4.4, 4.5, and 4.6, respectively. Finally, based on these analyses, we will refine the model of CPO development under simple shear deformation in Sections 4.7 and discuss implications for natural ice in Sections 4.8."

line 298-299: I think this sentence is redundant:

Thank you for your suggestion, we will delete this sentence

line 310: I believe en dashes are used for ranges

You are correct. We fixed those.

"Cyprych et al (2016) and Wilson et al (2019) conducted axial compression experiments on pure $D_2O$ ice and $D_2O$ ice doped with 20 or 40 vol.% graphite (<150 or 150–355 μm in diameter), or 20 or 40 vol.% calcite (< 150 or 150–355 μm in diameter) at temperatures 10°C below the melting point (actual at -7°C) and ambient pressures."

line 321-324: repetitive

We have adjusted the sentences and move the following two sentences to the beginning of the Subsetion 4.2.

"Due to rapid grain growth, we could not perform an ECAP experiment on particle-free ice without fracturing it. So a small fraction of graphite (~1 vol.%) was added to inhibit grain growth."

We will delete the following sentence.

"Based on the discussion above, we think the CPOs in our samples are comparable with the CPOs in previous studies."

line 335-340: I suggest mentioning the names and location of the glaciers for natural samples instead of 'a glacier'. For example: Thomas et al (2021) found a roughly round c-axis cluster in the shear margin  *Priestly* Glacier in Antarctica.

Changed as suggested.

"The CPOs reported in Hudleston (1977) for a marginal shear zone of Barnes Ice Cap are all characterized by round c-axis clusters. The CPOs reported in Jackson (1999) and Jackson and Kamb (1997) for a marginal shear zones of Ice Stream B, however, have elongated c-axis clusters perpendicular to the shear direction. Recently, studies employing modern EBSD technique provide much more data for CPOs in natural ice samples. Monz et al. (2021) found an elongated primary c-axis cluster in a shear-dominated region of Storglaciären Glacier; while Thomas et al. (2021) found a roughly round primary c-axis cluster in the shear margin of Priestly Glacier, Antarctica."

line 357:

Changed as suggested.

"However, Wilson and Peternell (2012) present more complex patterns of CPOs, which are characterized by double-cluster patterns at a shear strain of ∼1 at -2 ∘C, despite conducting their experiments using the same apparatus and kinematic constraints as Li et al (2000)."

line 359: two recent studies

Changed as suggested.

"Two recent studies both reported double-cluster CPOs at warm temperatures (Qi et al, 2019; Journaux et al, 2019)."

line 362-363: It is worth noting that under the same strain, the strength of the CPO in the sample deformed by ECAP is weaker than that in samples deformed continuously *due to effects of annealing discussed in Section 4.1.*

Changed as suggested.

"It is worth noting that under the same strain, the strength of the CPO in the sample deformed by ECAP is weaker than that in samples deformed continuously due to effects of annealing discussed in Section 4.1."

line 369: in *Qi et al (2019)*

Changed as suggested.

"The model of CPO development in Qi et al (2019) needs to be refined."

line 375: laboratory samples *from this study*.

Changed as suggested.

"Here, we review these with new results from natural (Monz et al, 2021; Thomas et al, 2021) and laboratory samples from this study."

line 385: is in good agreement with laboratory data *from this study,*

line 385-386: I'm not sure I understand this second part of the sentence. Do you mean: *and is possibly because the secondary cluster weakens with increasing strain, rather than moving towards the primary cluster.* ?

You are correct, we updated the text as follows based on both comments.

"The absence of low angles (<40°) between clusters in natural samples is in good agreement with laboratory data from this study, and it is possibly because the

secondary cluster weakens with increasing strain, rather than moving towards the primary cluster."

line 390: define FFT.

Changed as suggested.

"The evolution of c-axis clusters was compared between CPOs from experiments and a viscoplastic fast-Fourier-transform (VPFFT) model (Llorens et al., 2017), as detailed in Qi et al. (2019)."

line 391-393: The SpecCAF model, incorporating recrystallization, lattice rotation and grain rotation processes, yielded excellent quantitative agreement in the CPOs from with experimental observations  (Richards et al, 2021).

Changed as suggested.

"The SpecCAF model, incorporating recrystallization, lattice rotation and grain rotation processes, yielded excellent quantitative agreement in the CPOs from with experimental observations (Richards et al, 2021)."

Line 395-396: *see* Richards et al ( 2021))

Changed as suggested.

"Applying this model to simple shear yields roughly constant values of $\varphi$ with increasing strain at shear strains > 1, when the effect of GBM is similar to or smaller than the value estimated for natural ice ($\varphi \approx 74°$, $71°$, $65°$ and $60°$ for $\beta = 1$, $0.8$, $0.6$ and $0.5\beta_0$, respectively; see Richards et al (2021))"

line 408:  have to deform through slip

Sorry for the typo. Changed as suggested.

"Grains with low resolved shear stresses, or Schmid factors, on the basal plane (also referred to as poorly oriented for easy basal slip), have to deform through slip on non-basal slip systems."

line 408-409: Is this really your hypothesis? In the introduction you state something similar in line 26-28: 'When deformed by dislocation glide, a single crystal of ice is several orders of magnitude weaker for slip on the basal plane, (0001), than on others (Duval et al, 1983).' need to be clear what is your results or hypothesis and what is known from earlier studies.

We apologize for the incorrect statement. This sentence was modified incorrectly in a

previous review. Here is the corrected text.

"For ice, these non-basal-slip dislocations are more difficult to glide (Duval et al., 1983), and there will be more than one interacting slip system."

line 411: remove double parenthesis around Figure 12(c);

check hyphenation in higher-Schmid and lower-Schmid factors

Thanks for your suggestions. The sentences are modified as follows:

"Grains with higher-Schmid factors on the basal plane tend to grow by consuming grains with lower-Schmid factors (Figure 12(c))".

line 429-430: awkward sentence

line 429-430: Sorry for the typo. Updated sentence as follows:

"Subgrain rotation recrystallization plays a critical role in CPO formation by supplying grains with varied orientations compared to the primary cluster, thus providing the necessary grains for GBM."

line 435:  *Consequently,* ?

Changed as suggested.

"Consequently, there are fewer grains in low-Schmid-factor orientations, which will reduce the number of grains with high dislocation density, effectively reducing the driving force for GBM."

line 430: secondary *clusters*

Changed as suggested.

"The orientation of secondary clusters is a result of the competition between lattice rotation and GBM."

line 461: We thank  Prof. Jianhua Rao for his help with designing the ECAP die.

Changed as suggested.

"We thank Jianhua Rao for his help with designing the ECAP die."

Fig. 10:

-panel a): I suggest adding the IPF-Y colorbar here as well, else readers have to jump between this Figure and Fig. 5. I would also indicate the shear direction on these figures

Thank you for your suggestion. Very reasonable. We made changes in the figure accordingly.

[Figure]

- do you have an explanation for why grain size increases in 3p compared to undeformed?

The 3pass and 2pass samples both come from sample number 19. After undergoing ECAP twice, the tail of sample 19 was retained for the 2pass analysis, while the head was used for deformation in the 3pass. Due to some sublimation during the first two passes, the diameter of the sample likely changed. Consequently, during the third

pass, the shear deformation might have been insufficient, potentially resulting in a larger grain size.

- panel d): legend for red and gray lines?

Thank you for your suggestion. However, we believe it would be more helpful to include a description of the red and gray lines in the caption of panel (d). The room is small in those plots.

"The gray line represents the shear plane, while the red line and associated numbers indicate the average angles of the SPO, as calculated from the rose diagram."

Fig. 11:

- caption: unclear what 'simple models' and 'numerical models' are. What model has been used (reference). Are these both results done in this study or obtained previously?

The simple models based on the evolution of the angle between the long axis of the strain ellipse and the shear direction or the passive rotation of a rigid line originally perpendicular to the shear plane (Etchecopar, 1977). The results from these two models are based on simple math, pertain solely to the geometric model of strain, and do not reflect the findings of this study.

It is also not clear if 'Model from R21' is a result you obtained in this study or if it was obtained by Richards et al, 2021. Please clarify.

The numerical model was the same as those reported in Richards et al (2021), producing simulated pole figures and the angle $\varphi_0$ at -5 °C. But we processed it again for this study. We changed it to "numerical model".

- 'Since the nominal shear strains in this study cannot be directly compared with those in previous experimental studies, a range of $\varphi$ is marked by a shaded box in the plot, independent of strain.' - I don't understand how this is more helpful than simply plotting your datapoints, and discuss why they differ from e.g. models in the text.

Due to the differences between the shear strains obtained in our experiments and those reported in previous laboratory studies and natural samples, we have opted to present our results as strain ranges rather than discrete data points. This approach aims to mitigate potential misunderstandings or misinterpretations by readers and to provide a clearer context for the variability in our findings.

- 'Data points at an nominal equivalent strain of 3.6 can be treated as a single c-axis

cluster, but a disappearing secondary cluster can still be identified, which gives a value of φ, marked by a shaded marker' - I don't understand this either. Where is this shaded marker? Is this still referring to the results from this study?

We are very sorry that this description belongs to a previous version and should have been deleted. This sentence is now deleted.

- Experimental data are from the *following studies* :

Changed as suggested.

"Experimental data are from following studies: $K_{72}$: Kamb (1972); $BD_{82}$: Bouchez and Duval (1982); $L_{00}$: Li et al (2000); $WP_{12}$: Wilson and Peternell (2012); $Q_{19}$: Qi et al (2019); $J_{19}$: Journaux et al (2019)."

- three markers (M21, H77 and T21) were placed at the right end of the x-axis suggesting that their *shear* strains are larger than 7.

Changed as suggested.

"The maximum strains for the natural samples were estimated to be larger than the scale of our experiments, and thus, three markers (M21, H77 and T21) were placed at the right end of the x-axis suggesting that their shear strains are larger than 7."

- Outcomes from a recent*ly* published numerical model by Richards et al (2021) are marked by colored thick lines.

Changed as suggested.

"Outcomes from a recently published numerical model by Richards et al (2021) are marked by colored thick lines."

- The termination of these curves suggests that a single-cluster fabric forms. - unclear what this means. Are you saying that for lines ending before nominal strain of 4 was reached have already developed a single-cluster? Isn't it strange that e.g. for $\beta = 0.5\beta_0$ two clusters develop, separated by 58 degrees and abruptly turn into a single cluster fabric?

Yes. Your understanding is correct. While achieving perfect replication of CPO results from nature or laboratory conditions through numerical models remains challenging, current models can still provide partial validation of CPO evolution theories. For example, the abrupt termination of weaker GBM curves ($\beta = 0.5$, 0.4, and 0.2 $\beta_0$) shows the transformation of double clusters into single cluster pattern. This result corresponds with our experimental finding that secondary clusters retain their

orientations but gradually weaken and disappear with increasing strain due to the diminishing effect of GBM. In the numerical model, the curves only show the evolution of the angle, but not the strength of the secondary cluster. Consider when there is only one last grain in the secondary cluster, the curve still has a value for $\varphi$. But in the next strain step, the orientation of this grain changes, and there is no secondary cluster. The curve terminates.

Fig. 12:

- this figure is almost identical to Fig. 10 in Qi et al, (2019) and should be cited as 'adapted from Qi et al, (2019)' or similar in the caption.

Changed as suggested.

"Schematic drawing for the development of CPOs in ice sheared in the laboratory, adapted from Qi et al, (2019)."

- panel labels are different from other figures. I think The Cryosphere asks for (a) instead of a.

Thanks for your suggestion, we have corrected it in the figure.12.

- panel a is a very useful overview. In my opinion this could also be a figure for the introduction, explaining these individual processes and how they work. Whether you move this (part of the) figure or reference it in the introduction is up to you.

Thank you for the valuable suggestion. Although we did not move panel (a) of Figure 12, we have added relevant background information in the Introduction section in response to the second major comment.

- panel d): again it is not clear where these model results come from and how they were obtained.

The numerical model was the same as those reported in Richards et al (2021), producing simulated pole figures and the angle $\varphi_0$ at -5°C. But we processed it again for this study. We changed it to "numerical model".

Fig. A1: - typo in all panels for the temperature panel y-axis

- increase line thickness in legend for temperature plots and adjust y-axis for better visibility.

Sorry for the typo. Changed as suggested.

[Figure]

**Technical comments**

1. Most citations are lacking a doi.

Thanks for your suggestion, we will add the doi.

2. Figure references are usually abbreviated as Fig. 1. Figure is used only in the beginning of sentences (see TC author guidelines https://www.the-cryosphere.net/submission.html)

Thanks for your suggestion, we will modify all the figure references.

3. You often use red-green lines in plots - beware colorblindness

Thank you for bringing this up. We updated the color scheme with a color-blindness friendly color scheme.

4. I think the bullet points in the conclusion are redundant. You can just remove them and write it as continuous text.

Changed as suggested.

Conclusion: "Utilizing the ECAP technique, we achieved a nominal equivalent strain of ~3.6 (a nominal shear strain of ~6.2) in polycrystalline ice doped with ~1 vol.% graphite deformed at −5°C in roughly simple shear. The cyclic annealing introduced by ECAP deformation, along with the presence of a small amount of graphite, may reduce the strength of the CPO, but will not alter the patterns of the CPO. All samples develop a primary c-axis cluster perpendicular to the shear plane and a secondary c-axis cluster in the profile plane antithetic to the imposed shear direction. The orientation of the primary cluster does not change as a function of strain. The secondary cluster roughly remains its orientation but weakens as strain increases. Annealing of the 6-pass sample at −3.5°C for 24 days revealed the same CPO patterns as before annealing. The strength of this CPO became slightly weaker and the secondary cluster became slightly stronger after annealing. A combination of our data and published literature data, and comparisons with numerical models reveal the key processes that control the evolution of CPOs in ice during shear. The CPO patterns results from a balance of two competing mechanisms: lattice rotation due to dislocation slip, strengthening the primary cluster and rotating and weakening the secondary one, and growth of grains by strain-induced GBM, strengthening both clusters and rotating the secondary cluster back. GBM contributes less as shear strain increases."

---

## Author Comment (AC2)

Dear Prof. Gerbi,

We appreciate your helpful comments. Here we present our responses to the comments. Our responses are in black, while your comments are in blue. We will make necessary revisions to address the questions.

This is a very valuable study to provide additional constraints on the crystallographic development of Ice Ih. As the authors note, the ice fabric plays a significant role in glacier and ice sheet mechanics, so being able to predict and explain fabric development provides a much stronger grounding for describing and modeling ice flow.

I particularly appreciate the authors explaining their experimental steps in such detail – it makes it easy for the reader to follow and understand the strengths of their approach. In addition, the primary conclusion of this study, namely the persistence of a secondary, albeit weak, c-axis cluster even at high shear strain, appears quite robust. I offer my suggestions below in the spirit of making the analysis more transparent, and thus easier to compare with other work.

Sensitivity. Line 175 and following suggest that all calculations related to the fabric use all orientation data. However, determining which pixels are labeled as ice vs graphite seems to have been a non-trivial exercise. Did the authors perform any sensitivity to evaluate how their processing algorithm may affect the final orientation or other datasets?

This is a good suggestion. To clarify, the data used to calculate the CPO for all orientation with a 30-μm step size do not involve extrapolation of unindexed points. We have restructured the section, so that the analysis of CPO (using 30-μm step size) and grain size (using 15-μm step size) are separated. For the data with a 15-μm step size, we did extrapolate unindexed points, incorporating EDS data in the process. In this extrapolation, the intensity of the graphite signal (greater than 1) is a crucial parameter. We performed sensitivity tests using different thresholds for the graphite signal intensity (1, 2, 4) to assess the impact of this extrapolation on microstructural features, such as grain size. The table below shows the effect of different thresholds for the graphite signal intensity on shape preferred orientation (SPO) and grain size after extrapolation.

The table demonstrates that the choice of signal intensity threshold has a slightly effect on the microstructural analysis. As the threshold increases, the SPO angle

slightly decreases, while grain size slightly increases. However, the magnitude of these changes is subtle, which may be attributed to differences in the index rate. Since unindexed areas are assigned to surrounding grains during the grain reconstruction process, the index rate can influence the final results after reconstruction. Overall, the threshold selection follows a consistent trend across the samples. As long as all samples were processed using the same threshold, our data are comparable between each other.

Table S1. The effect of different thresholds for EDS data on SPO and grain size.

| | Threshold of signal | undeformed | 33_1p | 19_2p | 19_3p | 21_4p | 34_5p | 38_6p |
|---|---|---|---|---|---|---|---|---|
| SPO (°) | 1 | | 32 | 12 | 14 | 11 | 17 | 9 |
| | 2 | | 26 | 10 | 12 | 10 | 14 | 8 |
| | 4 | | 22 | 6 | 10 | | 13 | 8 |
| grain size (µm) | 1 | 140 | 180 | 167 | 211 | 139 | 155 | 143 |
| | 2 | 154 | 195 | 186 | 222 | 137 | 168 | 149 |
| | 4 | 162 | 207 | 203 | 234 | | 180 | 153 |

As part of this, I would like to see an explanation of why the EDS and EBSD data were collected separately at different step sizes, as I would have thought that the hardware and software would allow for simultaneous collection.

Theoretically, the Aztech software allows simultaneous collection of EDS and EBSD data. However, we meet a technical challenge that EDS and EBSD require different acceleration voltages. Ice EBSD requires 30kV, but to get good quality EDS data for graphite, lower voltages (~15kV) are needed to EDS. Consequently, we need to scan twice using different voltages for EBSD and EDS. Meanwhile, since EDS was done separately, we tried to collect EDS data at a higher resolution, that is why the pixel of EDS is smaller than the step size of EBSD.

With some work, I think I can understand which figures and interpretations rely on the 15um vs 30um step size EBSD data. At the same time, I think that could be more clearly explained in the text.

Thank you for pointing this out. We have restructured the section, so that the analysis of CPO (using 30-µm step size) and grain size (using 15-µm step size) are in two subsections now.

line 157-158:

2.4 Analysis of microstructure

"Data with a 15-µm step size were combined with EDS data to identify unindexed

points, which were then used to analyze grain size, aspect ratio, and shape preferred orientations."

Section 2.5 Analysis of crystallographic orientations

"Orientation distributions were generated from the complete set of raw EBSD data with 30 µm step size using the MTEX toolbox in MATLAB (Bachmann et al, 2010; Mainprice et al, 2015). To quantify the strength of the CPOs, both the J-index (Bunge, 1982) and the M-index (Skemer et al, 2005) were used."

Additionally, we also added the information on step size in the caption.

"Figure 5. Microstructural analyses of an undeformed ice samples, using EBSD data with a 15-µm step size."

"Figure 8. Crystallographic fabric strength as a function of strain, based on EBSD data with a 30-µm step size."

"Figure 9. Aspect ratio of clusters as a function of strain, based on EBSD data with a 30-µm step size."

"Figure 10. Microstructure results for all deformed samples, based on EBSD data with a 15-µm step size."

Number of data points. I may have missed it, but I didn't see a total of the number of datapoints used in the orientation data analysis. I suggest adding that value to perhaps Figure 7 or Table 1.

Reply: Thank you for the suggestions. We have added it to Table 1.

Table 1. Summary of experiments. $\varepsilon'$ is nominal equivalent strain, $\varphi$ is the angle between the two clusters.

| Sample | Graphite fraction | Passes | Load (Kg) | Length (mm) | $\varepsilon'$ | Part for analysis | points of EBSD data | area(mm*mm) | $\varphi$ |
|---|---|---|---|---|---|---|---|---|---|
| ECAP_33 | 2.1wt.% (0.9 vol.%) | 1 | 42.5 | 110 | 0.6 | ECAP_33_1P | 144045 | 14.85*8.73 | 50° |
| ECAP_19 | 3.6wt.% (1.5 vol.%) | 1 | 42.5 | 98 | 0.6 | | | | |
| | | 2 | 37.5 | 100* | 1.2 | ECAP_19_2P | 99099 | 12.87*6.93 | 55° |
| | | 3 | 37.5 | 50 | 1.8 | ECAP_19_3P | 104550 | 12.75*7.38 | 50° |
| ECAP_21 | 1.8wt.% (0.8 vol.%) | 1 | 42.5 | 110 | 0.6 | | | | |
| | | 2 | 37.5 | 105 | 1.2 | | | | |
| | | 3 | 37.5 | 93 | 1.8 | | | | |
| | | 4 | 32.5 | 85 | 2.4 | ECAP_21_4P | 108035 | 15.81*6.15 | 45° |
| ECAP_34 | 2.6wt.% (1.1 vol.%) | 1 | 42.5 | 107 | 0.6 | | | | |

| | | 2 | 32.5 | 99 | 1.2 | | | | |
|---|---|---|---|---|---|---|---|---|---|
| | | 3 | 32.5 | 95 | 1.8 | | | | |
| | | 4 | 32.5 | 85 | 2.4 | | | | |
| | | 5 | 32.5 | 65 | 3 | ECAP_34_5P | 112892 | 13.86*7.97 | 60∘ |
| ECAP_38 | 2.2wt.% (0.9 vol.%) | 1 | 42.5 | 105 | 0.6 | | | | |
| | | 2 | 32.5 | 100 | 1.2 | | | | |
| | | 3 | 32.5 | 95 | 1.8 | | | | |
| | | 4 | 32.5 | 95 | 2.4 | | | | |
| | | 5 | 32.5 | 92 | 3 | | | | |
| | | 6 | 32.5 | 94 | 3.6 | ECAP_38_6P | 161100 | 16.11*9 | 55∘ |

Line 219: when discussing stress, presumably you mean differential, deviatoric, or shear stress? Please clarify.

Thank you for pointing this out. In most cases, we mean shear stress. For deviatoric stress and equivalent stress, we always write them out. Based on comments from Reviewer 1, this sentence has been deleted. And we have changed "stress" to "shear stress" in the next paragraph.

Line 257 refers to larger analysis areas used than presented in the paper. I would appreciate seeing the full maps (in appendix), as they help the reader evaluate heterogeneity. Similarly, for Figure 10, I would find it useful to indicate that the histograms and rose diagrams use larger datasets than shown if that is accurate.

Reply: Thank you very much for your suggestion, we have added the complete map in the supplementary. The supplementary figures are illustrated as follows.

ECAP_33_1p

[Figure]

ECAP_19_2p

[Figure]

ECAP_19_3p

[Figure]

ECAP_21_4p

ECAP_34_5p

[Figure]

ECAP_38_6p

For the reader to best appreciate the comparison of the experimental data with the modeled data, I suggest adding a section to Results to present the SpecCAF calculations. That way, the model results can stand somewhat independently for the later comparison.

We thank the reviewer for the suggestion. We have added a new subsection in the Results section.

"Section 3.5 Numerical modeling

The SpecCAF model requires deformation, temperature, and initial CPO as inputs. The model was run in simple shear, as the ECAP mostly results in simple shear. The

model was also run at -5°C, same as the experimental condition. The initial condition for CPO was set to isotropic, similar to our initial samples. The model was also run with the parameter β controlling the effect of GBM, reduced to different fractions (k) of the value Richards et al. (2021) found for T = −5°C (β0). The output of the modeling was the predicted angle between the c-axis clusters, φ, plotted alongside the experimental results in Fig. 11. The value of φ decreases with increasing shear strain. At a given strain, the value of φ decreases with decreasing β. At higher values of β (0.6 --- 1β0), φ decreases from over 80° to ~70° as shear strains increases to 1 and stays roughly constant at larger strains. At lower values of β (0.2 --- 0.5β0), the curves of φ terminate at different strains. This termination means the model cannot identify a secondary c-axis cluster, and the CPO is characterized by a single cluster. At β = 0.5β0, φ rapidly decreases from ~80° to around 60° as shear strain increases to 1, and gradually decreases at larger strains until the secondary cluster disappears at a shear strain of ~2.6. At β = 0.4β0 and 0.2β0, φ rapidly decreases to ~50° and ~30°, when the secondary cluster disappears at shear strains of ~1.3 and ~0.5, respectively. The curve of φ at β = 0.6β0 was found to closely match the experimental results in this study. The modelled CPOs at this value of β are illustrated in Fig 12."

Lines 182-3 and Figure 7d. It isn't clear to me how the data that lie off the profile line are used in the production of the histogram. Are all data projected onto the profile line? Or does the histogram include only a subset within a certain angular distance from the profile line? This may be explained in Qi et al. (2019), but a short review here (could also be in the appendix) would be helpful.

Reply: "We adopted a method similar to that of Fig. 2 in Qi et al. (2019). As illustrated in Supplementary Fig. S2(b), pole figures were generated using a lower hemisphere equal-area projection, with the shear plane (green circle) oriented perpendicular to the page. In the stereonets, angles ranging from 0 to 180° were defined on the shear plane. At a given angle, two semicircles with 5◦ between them (orange circle) were drawn perpendicular to the page. The number of data points falling between these semicircles was counted, normalized, and plotted as the frequency for each angle in a histogram. The angle φ was defined as the angle between the two peaks in the histogram (Supplementary Fig. S2(c))."

[Figure]

(a) Shear plane ref. frame

(b) Example data from PIL94 in Qi et al., (2019)

(c) Distribution

Lower hemisphere

"Supplementary Figure S2. (a) Typical two-cluster distribution of $c$ axes on stereonets within the shear plane reference frame. (b) A schematic illustration explaining the method used to quantify the distribution of $c$ axes. (c) A histogram of $c$ axes plotted in a histogram, illustrating the angle φ between the two clusters of $c$ axes."

In a similar vein, how do the authors define the boundaries of the clusters to calculate the ratio plotted in Figure 8c?

Reply: The ratio between the two clusters shown in Fig. 8c was determined by comparing the peak values of the distribution frequencies for each cluster, specifically at the 5° interval. We have added a sentence explaining this in the paragraph on φ. See the reply to the previous comment.

Section 4.5. Another natural dataset for comparison is from the temperate Jarvis Glacier in Alaska [Gerbi C et al. (2021). Microstructures in a shear margin: Jarvis Glacier, Alaska. Journal of Glaciology 67(266), 1163–1176. https://doi.org/10.1017/jog.2021.62]. The results paint a different picture than the experimental results here, but the conditions are also quite different, thus providing some assessment of the applicability of the present work.

We apologize for overlooking this research. We have now included the relevant information in Section 4.5.

"Gerbi et al., (2021) investigated the microstructure of the lateral shear margins of a temperate glacier, Jarvis Glacier. Despite the fabric being relatively weak due to high water content, and short flow distance, c axes are slightly more concentrated in regions closer to the margin where strain is larger. However, without azimuth angles, it is not possible to determine the orientation of these samples relative to the shear

plane."

Line 435: The authors suggest that the changing intensity of orientations clusters may relate to the number of grains in particular orientations. Could it instead (or also) be that the sizes of the grains change such that instead of having fewer grains in low-Schmid-factor orientations, it is just that these grains are smaller? This comment is based on the presumption that the data presented are all pixels as suggested in Figure 7, rather than one-point-per-grain.

This is a reasonable concern. First, let me clarify that our CPO analysis is based on EBSD data with a 30-µm step size. Our goal is to cover a large area in limited time, so that the data is more representative for the sample. Considering the grain size, there are probably 4-6 points across a grain. Grain reconstruction based on such data did not give us good quality.

Second, due to the influence of graphite on the EBSD data, the graphite-rich regions are not indexed. Without combining EDS+EBSD data together, we found the noise reduction process in channel5 will artificially enlarge some grains, and MTEX are not able to process grains for that data. So, we cannot reconstruct grains to create a pole figure for each individual grain.

However, based on our orientation results and grain sizes reconstructed with a 15 µm step size (Figures 10a and 10b in the article), grain-size distributions in 4P–6P samples can be well fit by a log-normal distribution. This suggests that there are not a group of small grains.

Additionally, Qi et al. (2019) reported that the pole figure from full orientation data (Figure 4c in Qi et al. (2019)) exhibits stronger secondary clustering compared to the one-point-per-grain pole figure (Figure 4b in Qi et al. (2019)). This observation suggests that low-Schmid-factor grains are fewer in number but often have larger grain areas.

In the discussion, I suggest adding two subsections. One is for the limitations of these experiments: that is, under which natural conditions do the authors think these experiments apply? The second relates to rheological implications. The introduction opens with reference to the value of this work for rheology. I would find it quite valuable for the authors to reflect on how their work impacts the evolution of the mechanical properties of sheared ice.

We thank the suggestions. Now a subsection "4.8 Implications to natural ice" is added.

"As ice is a highly anisotropic material, once formed, the CPO has significant influences on the mechanical strength of ice, and thus, models of the flow of natural ice often rely on applying an anisotropy factor to the laboratory-derived flow laws (Pimienta et al., 1987). Although the models using an anisotropy factor obtained from historic observations generally predict the right magnitude of glacial flow rates, Azuma's CPO-only flow law (Azuma, 1994) best describes the strength evolution as strain increases (Fan et al., 2021). The anisotropic factor in Azuma's CPO-only flow law is not based on phenomenological data but calculated from the orientation data of $c$ axes. Thus, this study and many previous studies focusing on the CPO development in ice aim to understand the physical processes that controls the evolution of c-axis orientation during deformation, and thus, to better constrain the anisotropy factor and predict the glacial flow rates, especially for ice-stream margins, where shear deformation is severe.

One important result from the observations of this study is that the secondary $c$-axis cluster remains its orientation and weakens with increasing strain. This result changes our previous intuitive hypothesis and provides different values of anisotropy factors at strains when the $c$-axis fabric is evolving from double clusters to a single cluster. Such fabric evolution could occur at regions not too far away from the dome, where the shear deformation just starts, and possibly at the upstream regions of ice-stream margins, where the shear plane changes from horizontal to vertical, and the fabric has to evolve accordingly. However, it is necessary to note that the laboratory observed microstructures and their evolutions are obtained at strain rates and stresses larger than those in natural ice bodies. The contribution from rotation recrystallization in natural ice could be weaker, as stresses are smaller. Moreover, natural ice is usually impure. Insoluble particles and air bubbles could accumulate along grain boundaries and reduce grain boundary mobility and inhibit GBM, which possibly also occurred in the experiments of this study. The effects of the two mechanisms can only be qualitatively discussed for natural conditions. The microstructural processes observed in laboratory experiments provides good constrains on models and simulations (e.g., Richards et al., 2021; Hunter et al., 2022), which could extrapolate laboratory results to natural ice."

---

## Author Response (AR2)

Dear Dr. Kaitlin Keegan,

We sincerely appreciate your thoughtful comments that will help improve our manuscript. Here we present our responses to the comments. Editorial comments are in blue, our responses are in black, extracts from the revised manuscript are in italics.

L5: add 'the' before 'equal-'

Changed as suggested.

L13: should be 'suggests' here

Yes. Changed as suggested.

L29: should be '⋯flow changes, the rate of change of the flow rate depends on⋯'

Yes. Accurate. Thank you. Changed as suggested.

L31: remove 'forcing' after 'gravitational' because it is redundant

Changed as suggested.

L58: either add 'the' before 'laboratory' or use the phrase 'in a laboratory setting'

Changed to "achieved in a laboratory setting".

L67: should be 'This discrepancy leas to an uncertainty as to whether⋯'

Changed as suggested.

L80-81: 'The results could help our understanding of the⋯'

Changed as suggested.

L86: remove 'for' here

Changed as suggested.

L93: within the text, spell out Figure here (and other places). When referencing the figure in parentheses within the text, use the abbreviation 'Fig.'

Thank you. Figure references within the text are changed to Figure, in 21 places.

L109: add 'in' before 'a nominal' here

Changed as suggested.

L112: add 'the' before 'nominal' in both places in the last sentence here.

Changed as suggested.

L116: is 'attached to' a better way to describe the setup than 'stuck to'?

Yes. Changed as suggested.

L148: add 'the' before 'ECAP' here

Changed as suggested.

L157: within 15 minutes of what?

We separate this sentence.

Now in Line 156-157: "*This process was conducted using a band saw in a cold room maintained at −10℃. The cutting was done within 15 minutes, and the thermal effects……*"

L169: you can remove 'reason' from the sentence here. 'The EDS data were collected separately from the EBSD because the operating voltage for EDS (below 15 kV) needs to be lower than that used for EBSD (30 kV) to obtain high-quality data for graphite.'

Changed as suggested.

L179-180: add a dash between 'X-rays'

Changed as suggested.

L215: should this be 'produces' here, instead of 'producing'?

Yes. Sorry for this. Changed as suggested.

L244: do you mean 'where the shear stress drops to zero.' here?

Yes. Changed to "*where the shear stress drops to zero*".

L245: add 'at' before 'the corner'

Changed as suggested.

L253: lower case 'we' here

Changed as suggested.

L254: it might be clearer to say 'our samples' here, instead of 'the samples'

Yes. Changed as suggested.

L272: should be a comma instead of a semicolon here

Changed as suggested.

L288: 'oranges' should be 'orange curves'

Changed as suggested.

L294: should be '..required by this model⋯'

Changed as suggested.

L299: 'most' instead of 'mostly'

Changed as suggested.

L303: 'Before moving on to discussion,⋯' is a little confusing. Consider changing this phrase to 'Here, we note⋯'

Changed as suggested.

L304: should be '⋯due to the fact that EBSD⋯' here

Changed as suggested.

L305: need 'a' between 'in' and 'limited', and between 'such' and 'large'

Changed as suggested.

L306: add 'which is' before 'insufficient' here

Changed as suggested.

L308: '⋯dominated by a few very large grains.' might convey what you mean better

Changed as suggested.

L309: '⋯that pole figures drawn using all orientation data represent the CPOs in these samples well.'

Changed as suggested.

L310: "In this discussion,⋯'

Changed as suggested.

L316: make 'c' italics here

Sorry for this. Changed as suggested.

L345: add a comma after 'So'

Changed as suggested.

L356: I think you mean "Extrapolating from their results,⋯' here

Yes. You are right. Changed as suggested.

L371: 'a' should be 'the' and 'zones' should be singular

Changed as suggested. We have also replaced the "a" to "the" in line 370.

L373: 'techniques'

Changed as suggested.

L374: the semicolon should be a comma

Changed as suggested.

L384: add 'the fact' before 'that the stress⋯' here

Changed as suggested.

L399: add 'the' before 'effects'

Changed as suggested.

L405: I think you mean 'retains' instead of 'remains' here; consider adding a transition phrase to begin the second sentence here, like 'Thus,⋯'

Yes, retains is a better choice. And "Thus," is added to the second sentence.

L411: make 'c' italics here

Changed as suggested.

L428: add 'it' before 'disappears'

Changed as suggested. And the "remains" is changed to "retains" in the line above.

L436: I think that you mean 'retains' instead of 'remains' here

Changed as suggested.

L440: I think you mean "The resulting angle…" here

Changed as suggested.

L443: add a comma after 'Here'

Changed as suggested.

L449: 'systems' should be singular here

We think it should be plural. There are several non-basal slip systems (for example, prismatic and pyramidal slip systems). During deformation, slip may occur on more than one slip systems.

L479: '…the CPO has a significant influence…'

Changed as suggested.

L482: 'correct' might be a better word than 'right' here

Yes. Changed as suggested.

L484: add 'rather is' before 'calculated'

Changed as suggested.

L485: 'controls' should be singular; consider changing this second use of 'thus' to something else like 'ultimately'

Yes. Thank you. Changed as suggested.

L488: again, I think you mean 'retains' here

Changed as suggested.

L498: should be 'constraints' here

Changed as suggested.

L499: 'which could allow for the extrapolation of laboratory…'

Changed as suggested.

L506: 'retains'

Changed as suggested.

Figure 1 caption: should 'stuck to' be 'attached to'?

Yes. Changed to "*attached to*".

Figure 4: title above panel (f) should say 'reconstructed'

Changed as suggested.

Figure 10 caption: 'sub-areas'

Changed as suggested.

Figure 11 caption: '···from the following studies:'

Changed as suggested.

Figure S2 caption: 'plane normal to the shear plane···'

Changed as suggested.

We also made a few corrections and modifications in words and citations. We think these can clarify some parts in the manuscript. Line numbers are with respect to the new version.

Line 3: "*The development of CPOs is governed by two pivotal mechanisms: recrystallization dominated by subgrain/lattice rotation and by strain-induced grain boundary migration (GBM)*" is changed to

"*The development of CPOs is governed by the relative importance of two pivotal recrystallization mechanisms: subgrain/lattice rotation and strain-induced grain boundary migration (GBM)*".

Line 33: Added citation of Monz et al., 2021.

Line 40: We want to make it clear that we mean the c axes are normal to the basal planes.

"*... characterized by the alignment of the ⟨0001⟩ axes (c axes), normal to the basal planes*" is changed to "*...characterized by the alignment of the ⟨0001⟩ axes (c axes, which are normal to the basal planes)*".

Line 50: This is what we inferred. So we changed "*typically exhibit*" to "*are inferred to have*".

Line 52: Changed citations to Jacka and Maccagnan, 1984; Jacka and Li, 2000; Qi et al., 2017.

Line 54: Added citations of Guillope and Poirier, 1979; Poirier, 1985; Urai et al., 1986.

Line 60: We want to make our descriptions more accurate for citations.

The sentence is changed to "*Natural ice samples from shear zones have strong single (Thomas et al., 2021) or double c-axis clusters (Jackson and Kamb, 1997; Jackson, 1999; Thomas et al., 2021) with the primary cluster perpendicular to the shear plane. Where shear strain is quantified, the double clusters switch to a single cluster at γ> 5*

*(Hudleston, 1977)."*

Line 97: The sentence is changed to "*The channels are mirror-finished, and are coated with a layer of solid soap…*"

Line 160: "*Sample mounting to copper ingots to go into the SEM was done…*"

Line 161: The sentence "*A small piece of the sample…*" is now moved to the end of the paragraph.

Line 175: The original sentence sounds like the unindexed points were used to analyze. We separate the sentence into two.

"*Orientation data from EBSD with a 15-µm step size were combined with element data from EDS to identify unindexed points. The combined data were then used to analyze…*"

Line 180: The sentence "*Note that due to the relatively low…*" is moved from originally line 184 to here. And the sentence "*Consequently, the data…*" (used to go after sentence "*Note that…*") can be removed.

Line 202: "*…a method similar to that of Figure 2 in Qi et al. (2019), based on an approach used by Bouchez and Duval (1982).*"

Line 269: "*the strength of…*" is removed, as weaker means we are talking about strength.

Line 272: "*The logarithm of the ratio of data-point number in the primary and the secondary clusters increases with increasing strain*" is changed to

"*The ratio of number of data points in the primary and the secondary clusters increases with increasing strain*".

Line 295: "*of ~10°*" changed to "*at ~10°*".

Line 369: "*…for which CPOs with substantial measured grain have been reported*" changed to

"*…for which CPOs with a large number of grains have been measured*".

Line 453: "*tend to exhibit*" changed to "*are inferred to have*".

Figure 11 captions: in the first sentence, we removed citation of Etchecopar, 1977, as his simple model has been removed in the previous version.